# More Supervision, Less Computation: Statistical-Computational Tradeoffs in Weakly Supervised Learning

**Xinyang Yi**[†*]   **Zhaoran Wang**[‡*]   **Zhuoran Yang**[‡*]   **Constantine Caramanis**[†]   **Han Liu**[‡]
[†]The University of Texas at Austin      [‡]Princeton University
[†]{yixy,constantine}@utexas.edu    [‡]{zhaoran,zy6,hanliu}@princeton.edu
{∗: equal contribution}

## Abstract

We consider the weakly supervised binary classification problem where the labels are randomly flipped with probability $1 - \alpha$. Although there exist numerous algorithms for this problem, it remains theoretically unexplored how the statistical accuracies and computational efficiency of these algorithms depend on the degree of supervision, which is quantified by $\alpha$. In this paper, we characterize the effect of $\alpha$ by establishing the information-theoretic and computational boundaries, namely, the minimax-optimal statistical accuracy that can be achieved by all algorithms, and polynomial-time algorithms under an oracle computational model. For small $\alpha$, our result shows a gap between these two boundaries, which represents the computational price of achieving the information-theoretic boundary due to the lack of supervision. Interestingly, we also show that this gap narrows as $\alpha$ increases. In other words, having more supervision, i.e., more correct labels, not only improves the optimal statistical accuracy as expected, but also enhances the computational efficiency for achieving such accuracy.

## 1   Introduction

Practical classification problems usually involve corrupted labels. Specifically, let $\{(\mathbf{x}_i, z_i)\}_{i=1}^n$ be $n$ independent data points, where $\mathbf{x}_i \in \mathbb{R}^d$ is the covariate vector and $z_i \in \{0, 1\}$ is the uncorrupted label. Instead of observing $\{(\mathbf{x}_i, z_i)\}_{i=1}^n$, we observe $\{(\mathbf{x}_i, y_i)\}_{i=1}^n$ in which $y_i$ is the corrupted label. In detail, with probability $(1 - \alpha)$, $y_i$ is chosen uniformly at random over $\{0, 1\}$, and with probability $\alpha$, $y_i = z_i$. Here $\alpha \in [0, 1]$ quantifies the degree of supervision: a larger $\alpha$ indicates more supervision since we have more uncorrupted labels in this case. In this paper, we are particularly interested in the effect of $\alpha$ on the statistical accuracy and computational efficiency for parameter estimation in this problem, particularly in the high dimensional settings where the dimension $d$ is much larger than the sample size $n$.

There exists a vast body of literature on binary classification problems with corrupted labels. In particular, the study of randomly perturbed labels dates back to [1] in the context of random classification noise model. See, e.g., [12, 20] for a survey. Also, classification problems with missing labels are also extensively studied in the context of semi-supervised or weakly supervised learning by [14, 17, 21], among others. Despite the extensive study on this problem, its information-theoretic and computational boundaries remain unexplored in terms of theory. In a nutshell, the information-theoretic boundary refers to the optimal statistical accuracy achievable by any algorithms, while the computational boundary refers to the optimal statistical accuracy achievable by the algorithms under a computational budget that has a polynomial dependence on the problem scale $(d, n)$. Moreover, it remains unclear how these two boundaries vary along with $\alpha$. One interesting question to ask is

how the degree of supervision affects the fundamental statistical and computational difficulties of this problem, especially in the high dimensional regime.

In this paper, we sharply characterize both the information-theoretic and computational boundaries of the weakly supervised binary classification problems under the minimax framework. Specifically, we consider the Gaussian generative model where $X|Z = z \sim \mathcal{N}(\boldsymbol{\mu}_z, \boldsymbol{\Sigma})$ and $z \in \{0, 1\}$ is the true label. Suppose $\{(\mathbf{x}_i, z_i)\}_{i=1}^n$ are $n$ independent samples of $(X, Z)$. We assume that $\{y_i\}_{i=1}^n$ are generated from $\{z_i\}_{i=1}^n$ in the aforementioned manner. We focus on the high dimensional regime, where $d \gg n$ and $\boldsymbol{\mu}_1 - \boldsymbol{\mu}_0$ is $s$-sparse, i.e., $\boldsymbol{\mu}_1 - \boldsymbol{\mu}_0$ has $s$ nonzero entires. We are interested in estimating $\boldsymbol{\mu}_1 - \boldsymbol{\mu}_0$ from the observed samples $\{(\mathbf{x}_i, y_i)\}_{i=1}^n$. By a standard reduction argument [24], the fundamental limits of this estimation task are captured by a hypothesis testing problem, namely, $H_0 : \boldsymbol{\mu}_1 - \boldsymbol{\mu}_0 = \mathbf{0}$ versus $H_1 : \boldsymbol{\mu}_1 - \boldsymbol{\mu}_0$ is $s$-sparse and

$$(\boldsymbol{\mu}_1 - \boldsymbol{\mu}_0)^\top \boldsymbol{\Sigma}^{-1}(\boldsymbol{\mu}_1 - \boldsymbol{\mu}_0) := \gamma_n > 0, \tag{1.1}$$

where $\gamma_n$ denotes the signal strength that scales with $n$. Consequently, we focus on studying the fundamental limits of $\gamma_n$ for solving this hypothesis testing problem.

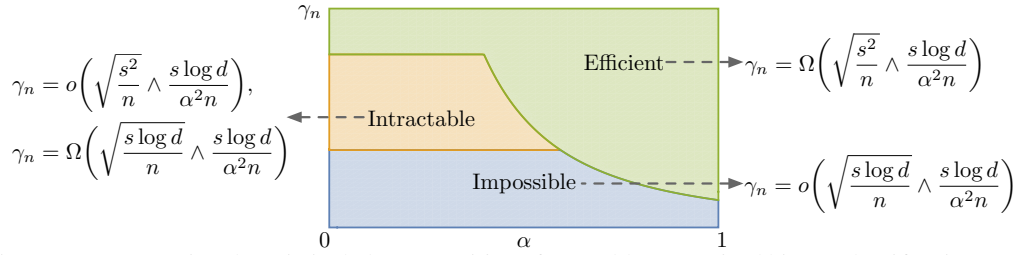

Figure 1: Computational-statistical phase transitions for weakly supervised binary classification. Here $\alpha$ denotes the degree of supervision, i.e., the label is corrupted to be uniformly random with probability $1 - \alpha$, and $\gamma_n$ is the signal strength, which is defined in (1.1). Here $a \wedge b$ denotes $\min\{a, b\}$.

Our main results are illustrated in Figure 1. Specifically, we identify the impossible, intractable, and efficient regimes for the statistical-computational phase transitions under certain regularity conditions.

(i) For $\gamma_n = o[\sqrt{s \log d/n} \wedge (1/\alpha^2 \cdot s \log d/n)]$, any algorithm is asymptotically powerless in solving the hypothesis testing problem.

(ii) For $\gamma_n = \Omega[\sqrt{s \log d/n} \wedge (1/\alpha^2 \cdot s \log d/n)]$ and $\gamma_n = o[\sqrt{s^2/n} \wedge (1/\alpha^2 \cdot s \log d/n)]$, any tractable algorithm that has a polynomial oracle complexity under an extension of the statistical query model [18] is asymptotically powerless. We will rigorously define the computational model in §2.

(iii) For $\gamma_n = \Omega[\sqrt{s^2/n} \wedge (1/\alpha^2 \cdot s \log d/n)]$, there is an efficient algorithm with a polynomial oracle complexity that is asymptotically powerful in solving the testing problem.

Here $\sqrt{s \log d/n} \wedge (1/\alpha^2 \cdot s \log d/n)$ gives the information-theoretic boundary, while $\sqrt{s^2/n} \wedge (1/\alpha^2 \cdot s \log d/n)$ gives the computational boundary. Moreover, by a reduction from the estimation problem to the testing problem, these boundaries for testing imply the ones for estimating $\boldsymbol{\mu}_2 - \boldsymbol{\mu}_1$ as well.

Consequently, there exists a significant gap between the computational and information-theoretic boundaries for small $\alpha$. In other word, to achieve the information-theoretic boundary, one has to pay the price of intractable computation. As $\alpha$ tends to one, this gap between computational and information-theoretic boundaries narrows and eventually vanishes. This indicates that, having more supervision not only improves the statistical accuracy, as shown by the decay of information-theoretic boundary in Figure 1, but more importantly, enhances the computational efficiency by reducing the computational price for attaining information-theoretic optimality. This phenomenon — "more supervision, less computation" — is observed for the first time in this paper.

## 1.1 More Related Work, Our Contribution, and Notation

Besides the aforementioned literature on weakly supervised learning and label corruption, our work is also connected to a recent line of work on statistical-computational tradeoffs [2–5, 8, 13, 15, 19, 26–28]. In comparison, we quantify the statistical-computational tradeoffs for weakly supervised learning for the first time. Furthermore, our results are built on an oracle computational model

in [8] that slightly extends the statistical query model [18], and hence do not hinge on unproven conjectures on computational hardness like planted clique. Compared with our work, [8] focuses on the computational hardness of learning heterogeneous models, whereas we consider the interplay between supervision and statistical-computational tradeoffs. A similar computational model is used in [27] to study structural normal mean model and principal component analysis, which exhibit different statistical-computational phase transitions. In addition, our work is related to sparse linear discriminant analysis and two-sample testing of sparse means, which correspond to our special cases of $\alpha = 1$ and $\alpha = 0$, respectively. See, e.g., [7, 23] for details. In contrast with their results, our results capture the effects of $\alpha$ on statistical and computational tradeoffs.

In summary, the contribution of our work is two-fold:

(i) We characterize the computational and statistical boundaries of the weakly supervised binary classification problem for the first time. Compared with existing results for other models, our results do not rely on unproven conjectures.

(ii) Based on our theoretical characterization, we propose the "more supervision, less computation" phenomenon, which is observed for the first time.

**Notation.** We denote the $\chi^2$-divergence between two distributions $\mathbb{P}, \mathbb{Q}$ by $D_{\chi^2}(\mathbb{P}, \mathbb{Q})$. For two nonnegative sequences $a_n, b_n$ indexed by $n$, we use $a_n = o(b_n)$ as a shorthand for $\lim_{n \to \infty} a_n/b_n = 0$. We say $a_n = \Omega(b_n)$ if $a_n/b_n \geq c$ for some absolute constant $c > 0$ when $n$ is sufficiently large. We use $a \vee b$ and $a \wedge b$ to denote $\max\{a, b\}$ and $\min\{a, b\}$, respectively. For any positive integer $k$, we denote $\{1, 2, \ldots, k\}$ by $[k]$. For $\mathbf{v} \in \mathbb{R}^d$, we denote by $\|\mathbf{v}\|_p$ the $\ell_p$-norm of $\mathbf{v}$. In addition, we denote the operator norm of a matrix $\mathbf{A}$ by $\|\mathbf{A}\|_2$.

## 2 Background

In this section, we formally define the statistical model for weakly supervised binary classification. Then we follow it with the statistical query model that connects computational complexity and statistical optimality.

### 2.1 Problem Setup

Consider the following Gaussian generative model for binary classification. For a random vector $\mathbf{X} \in \mathbb{R}^d$ and a binary random variable $Z \in \{0, 1\}$, we assume

$$\mathbf{X}|Z = 0 \sim \mathcal{N}(\boldsymbol{\mu}_0, \boldsymbol{\Sigma}), \quad \mathbf{X}|Z = 1 \sim \mathcal{N}(\boldsymbol{\mu}_1, \boldsymbol{\Sigma}), \tag{2.1}$$

where $\mathbb{P}(Z = 0) = \mathbb{P}(Z = 1) = 1/2$. Under this model, the optimal classifier by Bayes rule corresponds to the Fisher's linear discriminative analysis (LDA) classifier. In this paper, we focus on the noisy label setting where true label $Z$ is replaced by a uniformly random label in $\{0, 1\}$ with probability $1 - \alpha$. Hence, $\alpha$ characterizes the degree of supervision in the model. In specific, if $\alpha = 0$, we observe the true label $Z$, thus the problem belongs to supervised learning. Whereas if $\alpha = 1$, the observed label is completely random, which contains no information of the model in (2.1). This setting is thus equivalent to learning a Gaussian mixture model, which is an unsupervised problem. In the general setting with noisy labels, we denote the observed label by $Y$, which is linked to the true label $Z$ via

$$\mathbb{P}(Y = Z) = (1 + \alpha)/2, \; \mathbb{P}(Y = 1 - Z) = (1 - \alpha)/2. \tag{2.2}$$

We consider the hypothesis testing problem of detecting whether $\boldsymbol{\mu}_0 \neq \boldsymbol{\mu}_1$ given $n$ i.i.d. samples $\{y_i, \mathbf{x}_i\}_{i=1}^n$ of $(Y, \mathbf{X})$, namely

$$H_0 : \boldsymbol{\mu}_0 = \boldsymbol{\mu}_1 \text{ versus } H_1 : \boldsymbol{\mu}_0 \neq \boldsymbol{\mu}_1. \tag{2.3}$$

We focus on the high dimensional and sparse regime, where $d \gg n$ and $\boldsymbol{\mu}_0 - \boldsymbol{\mu}_1$ is $s$-sparse, i.e., $\boldsymbol{\mu}_0 - \boldsymbol{\mu}_1 \in \mathcal{B}_0(s)$, where $\mathcal{B}_0(s) := \{\boldsymbol{\mu} \in \mathbb{R}^d : \|\boldsymbol{\mu}\|_0 \leq s\}$. Throughout this paper, use the sample size $n$ to drive the asymptotics. We introduce a shorthand notation $\boldsymbol{\theta} := (\boldsymbol{\mu}_0, \boldsymbol{\mu}_1, \boldsymbol{\Sigma}, \alpha)$ to represent the parameters of the aforementioned model. Let $\mathbb{P}_{\boldsymbol{\theta}}$ be the joint distribution of $(Y, \mathbf{X})$ under our statistical model with parameter $\boldsymbol{\theta}$, and $\mathbb{P}_{\boldsymbol{\theta}}^n$ be the product distribution of $n$ i.i.d. samples accordingly. We denote the parameter spaces of the null and alternative hypotheses by $\mathcal{G}_0$ and $\mathcal{G}_1$ respectively. For any test function $\phi : \{(y_i, \mathbf{x}_i)\}_{i=1}^n \to \{0, 1\}$, the classical testing risk is defined as the summation of

type-I and type-II errors, namely

$$R_n(\phi; \mathcal{G}_0, \mathcal{G}_1) := \sup_{\boldsymbol{\theta} \in \mathcal{G}_0} \mathbb{P}_{\boldsymbol{\theta}}^n(\phi = 1) + \sup_{\boldsymbol{\theta} \in \mathcal{G}_1} \mathbb{P}_{\boldsymbol{\theta}}^n(\phi = 0).$$

The minimax risk is defined as the smallest testing risk of all possible test functions, that is,

$$R_n^*(\mathcal{G}_0, \mathcal{G}_1) := \inf_\phi R_n(\phi; \mathcal{G}_0, \mathcal{G}_1), \tag{2.4}$$

where the infimum is taken over all measurable test functions.

Intuitively, the separation between two Gaussian components under $H_1$ and the covariance matrix $\boldsymbol{\Sigma}$ together determine the hardness of detection. To characterize such dependence, we define the signal-to-noise ratio (SNR) as $\rho(\boldsymbol{\theta}) := (\boldsymbol{\mu}_0 - \boldsymbol{\mu}_1)^\top \boldsymbol{\Sigma}^{-1} (\boldsymbol{\mu}_0 - \boldsymbol{\mu}_1)$. For any nonnegative sequence $\{\gamma_n\}_{n \geq 1}$, let $\mathcal{G}_1(\gamma_n) := \{\boldsymbol{\theta} : \rho(\boldsymbol{\theta}) \geq \gamma_n\}$ be a sequence of alternative parameter spaces with minimum separation $\gamma_n$. The following minimax rate characterizes the information-theoretic limits of the detection problem.

**Definition 2.1** (Minimax rate). *We say a sequence $\{\gamma_n^*\}_{n \geq 1}$ is a minimax rate if*
- *For any sequence $\{\gamma_n\}_{n \geq 1}$ satisfying $\gamma_n = o(\gamma_n^*)$, we have $\lim_{n \to \infty} R_n^*[\mathcal{G}_0, \mathcal{G}_1(\gamma_n)] = 1$;*
- *For any sequence $\{\gamma_n\}_{n \geq 1}$ satisfying $\gamma_n = \Omega(\gamma_n^*)$, we have $\lim_{n \to \infty} R_n^*[\mathcal{G}_0, \mathcal{G}_1(\gamma_n)] = 0$.*

The minimax rate in Definition 2.1 characterizes the statistical difficulty of the testing problem. However, it fails to shed light on the computational efficiency of possible testing algorithms. The reason is that this concept does not make any computational restriction on the test functions. The minimax risk in (2.4) might be attained only by test functions that have exponential computational complexities. This limitation of Definition 2.1 motivates us to study statistical limits under computational constraints.

## 2.2 Computational Model

Statistical query models [8–11, 18, 27] capture computational complexity by characterizing the total number of rounds an algorithm interacts with data. In this paper, we consider the following statistical query model, which admits bounded query functions but allows the responses of query functions to be unbounded.

**Definition 2.2** (Statistical query model). *In the statistical query model, an algorithm $\mathscr{A}$ is allowed to query an oracle $T$ rounds, but not to access data $\{(y_i, \mathbf{x}_i)\}_{i=1}^n$ directly. At each round, $\mathscr{A}$ queries the oracle $r$ with a query function $q \in \mathcal{Q}_{\mathscr{A}}$, in which $\mathcal{Q}_{\mathscr{A}} \subseteq \{q : \{0,1\} \times \mathbb{R}^d \to [-M, M]\}$ denotes the query space of $\mathscr{A}$. The oracle $r$ outputs a realization of a random variable $Z_q \in \mathbb{R}$ satisfying*

$$\mathbb{P}\left( \bigcap_{q \in \mathcal{Q}_{\mathscr{A}}} \{|Z_q - \mathbb{E}[q(Y, \boldsymbol{X})]| \leq \tau_q\} \right) \geq 1 - 2\xi, \ \text{where}$$

$$\tau_q = [\eta(\mathcal{Q}_{\mathscr{A}}) + \log(1/\xi)] \cdot M/n \bigvee \sqrt{2[\eta(\mathcal{Q}_{\mathscr{A}}) + \log(1/\xi)] \cdot (M^2 - \{\mathbb{E}[q(Y, \boldsymbol{X})]\}^2)/n}. \tag{2.5}$$

*Here $\tau_q > 0$ is the tolerance parameter and $\xi \in [0, 1)$ is the tail probability. The quantity $\eta(\mathcal{Q}_{\mathscr{A}}) \geq 0$ in $\tau_q$ measures the capacity of $\mathcal{Q}_{\mathscr{A}}$ in logarithmic scale, e.g., for countable $\mathcal{Q}_{\mathscr{A}}$, $\eta(\mathcal{Q}_{\mathscr{A}}) = \log(|\mathcal{Q}_{\mathscr{A}}|)$. The number $T$ is defined as the oracle complexity. We denote by $\mathcal{R}[\xi, n, T, \eta(\mathcal{Q}_{\mathscr{A}})]$ the set of oracles satisfying (2.5), and by $\mathcal{A}(T)$ the family of algorithms that queries an oracle no more than $T$ rounds.*

This version of statistical query model is used in [8], and reduces to the VSTAT model proposed in [9–11] by the transformation $\widetilde{q}(y, \mathbf{x}) = q(y, \mathbf{x})/(2M) + 1/2$ for any $q \in \mathcal{Q}_{\mathscr{A}}$. The computational model in Definition 2.2 enables us to handle query functions that are bounded by an unknown and fixed number $M$. Note that that by incorporating the tail probability $\xi$, the response $Z_q$ is allowed to be unbounded. To understand the intuition behind Definition 2.2, we remark that (2.5) resembles the Bernstein's inequality for bounded random variables [25]

$$\mathbb{P}\left\{ \left| \frac{1}{n} \sum_{i=1}^n q(Y_i, \boldsymbol{X}_i) - \mathbb{E}[q(Y, \boldsymbol{X})] \right| \geq t \right\} \leq 2 \exp\left\{ \frac{t^2}{2\mathrm{Var}[q(Y, \boldsymbol{X})] + Mt} \right\}. \tag{2.6}$$

We first replace $\mathrm{Var}[q(Y, \boldsymbol{X})]$ by its upper bound $M^2 - \{\mathbb{E}[q(Y, \boldsymbol{X})]\}^2$, which is tight when $q$ takes values in $\{-M, M\}$. Then inequality (2.5) is obtained by replacing $n^{-1} \sum_{i=1}^n q(Y_i, \boldsymbol{X}_i)$ in (2.6) by $Z_q$ and then bounding the suprema over the query space $\mathcal{Q}_{\mathscr{A}}$. In the definition of $\tau_q$ in (2.5), we

incorporate the effect of uniform concentration over the query space $\mathcal{Q}_{\mathscr{A}}$ by adding the quantity $\eta(\mathcal{Q}_{\mathscr{A}})$, which measures the capacity of $\mathcal{Q}_{\mathscr{A}}$. In addition, under the Definition 2.2, the algorithm $\mathscr{A}$ does not interact directly with data. Such an restriction characterizes the fact that in statistical problems, the effectiveness of an algorithm only depends on the global statistical properties, not the information of individual data points. For instance, algorithms that only rely on the convergence of the empirical distribution to the population distribution are contained in the statistical query model; whereas algorithms that hinge on the first data point $(y_1, \mathbf{x}_1)$ is not allowed. This restriction captures a vast family of algorithms in statistics and machine learning, including applying gradient method to maximize likelihood function, matrix factorization algorithms, expectation-maximization algorithms, and sampling algorithms [9].

Based on the statistical query model, we study the minimax risk under oracle complexity constraints. For the testing problem (2.3), let $\mathcal{A}(T_n)$ be a class of testing algorithms under the statistical query model with query complexity no more than $T_n$, with $\{T_n\}_{n \geq 1}$ being a sequence of positive integers depending on the sample size $n$. For any $\mathscr{A} \in \mathcal{A}(T_n)$ and any oracle $r \in \mathcal{R}[\xi, n, T_n, \eta(\mathcal{Q}_{\mathscr{A}})]$ that responds to $\mathscr{A}$, let $\mathcal{H}(\mathscr{A}, r)$ be the set of test functions that deterministically depend on $\mathscr{A}$'s queries to the oracle $r$ and the corresponding responses. We use $\overline{\mathbb{P}}_{\boldsymbol{\theta}}$ to denote the distribution of the random variables returned by oracle $r$ when the model parameter is $\boldsymbol{\theta}$.

For a general hypothesis testing problem, namely, $H_0 \colon \boldsymbol{\theta} \in \mathcal{G}_0$ versus $H_1 \colon \boldsymbol{\theta} \in \mathcal{G}_1$, the minimax testing risk with respect to an algorithm $\mathscr{A}$ and a statistical oracle $r \in \mathcal{R}[\xi, n, T_n, \eta(\mathcal{Q}_{\mathscr{A}})]$ is defined as

$$\overline{R}_n^*(\mathcal{G}_0, \mathcal{G}_1; \mathscr{A}, r) := \inf_{\phi \in \mathcal{H}(\mathscr{A}, r)} \left[ \sup_{\boldsymbol{\theta} \in \mathcal{G}_0} \overline{\mathbb{P}}_{\boldsymbol{\theta}}(\phi = 1) + \sup_{\boldsymbol{\theta} \in \mathcal{G}_1} \overline{\mathbb{P}}_{\boldsymbol{\theta}}(\phi = 0) \right]. \tag{2.7}$$

Compared with the classical minimax risk in (2.4), the new notion in (2.7) incorporates the computational budgets via oracle complexity. In specific, we only consider the test functions obtained by an algorithm with at most $T_n$ queries to a statistical oracle. If $T_n$ is a polynomial of the dimensionality $d$, (2.7) characterizes the statistical optimality of computational efficient algorithms. This motivates us to define the computationally tractable minimax rate, which contrasts with Definition 2.1.

**Definition 2.3** (Computationally tractable minimax rate). *Let* $\mathcal{G}_1(\gamma_n) := \{\boldsymbol{\theta} : \rho(\boldsymbol{\theta}) \geq \gamma_n\}$ *be a sequence of model spaces with minimum separation* $\gamma_n$, *where* $\rho(\boldsymbol{\theta})$ *is the SNR. A sequence* $\{\overline{\gamma}_n^*\}_{n \geq 1}$ *is called a computationally tractable minimax rate if*

*• For any sequence* $\{\gamma_n\}_{n \geq 1}$ *satisfying* $\gamma_n = o(\overline{\gamma}_n^*)$, *any constant* $\eta > 0$, *and any* $\mathscr{A} \in \mathcal{A}(d^\eta)$, *there exists an oracle* $r \in \mathcal{R}[\xi, n, T_n, \eta(\mathcal{Q}_{\mathscr{A}})]$ *such that* $\lim_{n \to \infty} \overline{R}_n^*[\mathcal{G}_0, \mathcal{G}_1(\gamma_n); \mathscr{A}, r] = 1$;
*• For any sequence* $\{\gamma_n\}_{n \geq 1}$ *satisfying* $\gamma_n = \Omega(\overline{\gamma}_n^*)$, *there exist a constant* $\eta > 0$ *and an algorithm* $\mathscr{A} \in \mathcal{A}(d^\eta)$ *such that, for any* $r \in \mathcal{R}[\xi, n, T_n, \eta(\mathcal{Q}_{\mathscr{A}})]$, *we have* $\lim_{n \to \infty} \overline{R}_n^*[\mathcal{G}_0, \mathcal{G}_1(\gamma_n); \mathscr{A}, r] = 0$.

## 3 Main Results

Throughout this paper, we assume that the covariance matrix $\boldsymbol{\Sigma}$ in (2.1) is known. Specifically, for some positive definite $\boldsymbol{\Sigma} \in \mathbb{R}^{d \times d}$, the parameter spaces of the null and alternative hypotheses are defined as

$$\mathcal{G}_0(\boldsymbol{\Sigma}) := \{\boldsymbol{\theta} = (\boldsymbol{\mu}, \boldsymbol{\mu}, \boldsymbol{\Sigma}, \alpha) : \boldsymbol{\mu} \in \mathbb{R}^d\}, \tag{3.1}$$

$$\mathcal{G}_1(\boldsymbol{\Sigma}; \gamma_n) := \{\boldsymbol{\theta} = (\boldsymbol{\mu}_0, \boldsymbol{\mu}_1, \boldsymbol{\Sigma}, \alpha) : \boldsymbol{\mu}_0, \boldsymbol{\mu}_1 \in \mathbb{R}^d, \boldsymbol{\mu}_0 - \boldsymbol{\mu}_1 \in \mathcal{B}_0(s), \rho(\boldsymbol{\theta}) \geq \gamma_n\}. \tag{3.2}$$

Accordingly, the testing problem of detecting whether $\boldsymbol{\mu}_0 \neq \boldsymbol{\mu}_1$ is to distinguish

$$H_0 \colon \boldsymbol{\theta} \in \mathcal{G}_0(\boldsymbol{\Sigma}) \quad \text{versus} \quad H_1 \colon \boldsymbol{\theta} \in \mathcal{G}_1(\boldsymbol{\Sigma}; \gamma_n). \tag{3.3}$$

In §3.1, we present the minimax rate of the detection problem from an information-theoretic perspective. In §3.2, under the statistical query model introduced in §2.2, we provide a computational lower bound and a nearly matching upper bound that is achieved by an efficient testing algorithm.

### 3.1 Information-theoretic Limits

Now we turn to characterize the minimax rate given in Definition 2.1. For parameter spaces (3.1) and (3.2) with known $\boldsymbol{\Sigma}$, we show that in highly sparse setting where $s = o(\sqrt{d})$, we have

$$\gamma_n^* = \sqrt{s \log d / n} \wedge (1/\alpha^2 \cdot s \log d / n), \tag{3.4}$$

To prove (3.4), we first present a lower bound which shows that the hypothesis testing problem in (3.3) is impossible if $\gamma_n = o(\gamma_n^*)$.

**Theorem 3.1.** *For the hypothesis testing problem in (3.3) with known $\boldsymbol{\Sigma}$, we assume that there exists a small constant $\delta > 0$ such that $s = o(d^{1/2-\delta})$. Let $\gamma_n^*$ be defined in (3.4). For any sequence $\{\gamma_n\}_{n\geq 1}$ such that $\gamma_n = o(\gamma_n^*)$, any hypothesis test is asymptotically powerless, namely,*

$$\lim_{n\to\infty} \sup_{\boldsymbol{\Sigma}} R_n^*[\mathcal{G}_0(\boldsymbol{\Sigma}), \mathcal{G}_1(\boldsymbol{\Sigma}; \gamma_n)] = 1.$$

By Theorem 3.1, we observe a phase transition in the necessary SNR for powerful detection when $\alpha$ decreases from one to zero. Starting with rate $s\log d/n$ in the supervised setting where $\alpha = 1$, the required SNR gradually increases as label qualities decrease. Finally, when $\alpha$ reaches zero, which corresponds to the unsupervised setting, powerful detection requires the SNR to be $\Omega(\sqrt{s\log d/n})$. It is worth noting that when $\alpha = (s\log d/n)^{1/4}$, we still have $(n^3 s\log d)^{1/4}$ uncorrupted labels. However, our lower bound (along with the upper bound shown in Theorem 3.2) indicates that the information contained in these uncorrupted labels are buried in the noise, and cannot essentially improve the detection quality compared with the unsupervised setting.

Next we establish a matching upper bound for the detection problem in (3.3). We denote the condition number of the covariance matrix $\boldsymbol{\Sigma}$ by $\kappa$, i.e., $\kappa := \lambda_{\max}(\boldsymbol{\Sigma})/\lambda_{\min}(\boldsymbol{\Sigma})$, where $\lambda_{\max}(\boldsymbol{\Sigma})$ and $\lambda_{\min}(\boldsymbol{\Sigma})$ are the largest and smallest eigenvalues of $\boldsymbol{\Sigma}$, repectively. Note that marginally $Y$ is uniformly distributed over $\{0, 1\}$. For ease of presentation, we assume that the sample size is $2n$ and each class contains exactly $n$ data points. Note that we can always discard some samples in the larger class to make the sample sizes of both classes to be equal. Due to the law of large numbers, this trick will not affect the analysis of sample complexity in the sense of order wise.

Given $2n$ i.i.d. samples $\{(y_i, \mathbf{x}_i)\}_{i=1}^{2n}$ of $(Y, \boldsymbol{X}) \in \{0, 1\} \times \mathbb{R}^d$, we define

$$\mathbf{w}_i = \boldsymbol{\Sigma}^{-1/2}(\mathbf{x}_{2i} - \mathbf{x}_{2i-1}), \text{ for all } i \in [n]. \tag{3.5}$$

In addition, we split the dataset $\{(y_i, \mathbf{x}_i)\}_{i=1}^{2n}$ into two disjoint parts $\{(0, \mathbf{x}_i^{(0)})\}_{i=1}^n$ and $\{(1, \mathbf{x}_i^{(1)})\}_{i=1}^n$, and define

$$\mathbf{u}_i = \mathbf{x}_i^{(1)} - \mathbf{x}_i^{(0)}, \text{ for all } i \in [n]. \tag{3.6}$$

We note that computing sample differences in (3.5) and (3.6) is critical for our problem because we focus on detecting the difference between $\boldsymbol{\mu}_0$ and $\boldsymbol{\mu}_1$, and computing differences can avoid estimating $\mathbb{E}_{\mathbb{P}_{\boldsymbol{\theta}}}(\boldsymbol{X})$ that might be dense. For any integer $s \in [d]$, we define $\mathcal{B}_2(s) := \mathcal{B}_0(s) \cap \mathbb{S}^{d-1}$ as the set of $s$-sparse vectors on the unit sphere in $\mathbb{R}^d$. With $\{\mathbf{w}_i\}_{i=1}^n$ and $\{\mathbf{u}_i\}_{i=1}^n$, we introduce two test functions

$$\phi_1 := \mathbb{1}\left\{ \sup_{\mathbf{v}\in\mathcal{B}_2(s)} \frac{1}{n}\sum_{i=1}^n \frac{(\mathbf{v}^\top\boldsymbol{\Sigma}^{-1}\mathbf{w}_i)^2}{2\mathbf{v}^\top\boldsymbol{\Sigma}^{-1}\mathbf{v}} \geq 1 + \tau_1 \right\}, \tag{3.7}$$

$$\phi_2 := \mathbb{1}\left\{ \sup_{\mathbf{v}\in\mathcal{B}_2(1)} \frac{1}{n}\sum_{i=1}^n \langle \mathbf{v}, \text{diag}(\boldsymbol{\Sigma})^{-1/2}\mathbf{u}_i \rangle \geq \tau_2 \right\}, \tag{3.8}$$

where $\tau_1, \tau_2 > 0$ are algorithmic parameters that will be specified later. To provide some intuitions, we consider the case where $\boldsymbol{\Sigma} = \mathbf{I}$. Test function $\phi_1$ seeks a sparse direction that explains the most variance of $\mathbf{w}_i$. Therefore, such a test is closely related to the sparse principal component detection problem [3]. Test function $\phi_2$ simply selects the coordinate of $n^{-1}\sum_{i=1}^n \mathbf{u}_i$ that has the largest magnitude and compares it with $\tau_2$. This test is closely related to detecting sparse normal mean in high dimensions [16]. Based on these two ingredients, we construct our final testing function $\phi$ as $\phi = \phi_1 \vee \phi_2$, i.e., if any of $\phi_1$ and $\phi_2$ is true, then $\phi$ rejects the null. The following theorem establishes a sufficient condition for test function $\phi$ to be asymptotically powerful.

**Theorem 3.2.** *Consider the testing problem (3.3) where $\boldsymbol{\Sigma}$ is known and has condition number $\kappa$. For test functions $\phi_1$ and $\phi_2$ defined in (3.7) and (3.8) with parameters $\tau_1$ and $\tau_2$ given by*

$$\tau_1 = \kappa\sqrt{s\log(ed/s)/n}, \quad \tau_2 = \sqrt{8\log d/n}.$$

*We define the ultimate test function as $\phi = \phi_1 \vee \phi_2$. We assume that $s \leq C \cdot d$ for some absolute constant $Cs$ and $n \geq 64 \cdot s\log(ed/s)$. Then if*

$$\gamma_n \geq C'\kappa \cdot [\sqrt{s\log(ed/s)/n} \wedge (1/\alpha^2 \cdot s\log d/n)], \tag{3.9}$$

*where $C'$ is an absolute constant, then test function $\phi$ is asymptotically powerful. In specific, we have*

$$\sup_{\boldsymbol{\theta} \in \mathcal{G}_0(\boldsymbol{\Sigma})} \mathbb{P}^n_{\boldsymbol{\theta}}(\phi = 1) + \sup_{\boldsymbol{\theta} \in \mathcal{G}_1(\boldsymbol{\Sigma}; \gamma_n)} \mathbb{P}^n_{\boldsymbol{\theta}}(\phi = 0) \leq 20/d. \tag{3.10}$$

Theorem 3.2 provides a non-asymptotic guarantee. When $n$ goes to infinity, (3.10) implies that the test function $\phi$ is asymptotically powerful. When $s = o(\sqrt{d})$ and $\kappa$ is a constant, (3.9) yields $\gamma_n = \Omega[\sqrt{s \log d/n} \wedge (1/\alpha^2 \cdot s \log d/n)]$, which matches the lower bound given in Theorem 3.1. Thus we conclude that $\gamma_n^*$ defined in (3.4) is the minimax rate of testing problem in (3.3). We remark that when $s = \Omega(d)$, $\alpha = 1$, i.e., the standard (low-dimensional) setting of two sample testing, the bound provided in (3.9) is sub-optimal as [22] shows that SNR rate $\sqrt{d}/n$ is sufficient for asymptotically powerful detection when $n = \Omega(\sqrt{d})$. It is thus worth noting that we focus on the highly sparse setting $s = o(\sqrt{d})$ and provided sharp minimax rate for this regime. In the definition of $\phi_1$ in (3.7), we search over the set $\mathcal{B}_2(s)$. Since $\mathcal{B}_2(s)$ contains $\binom{d}{s}$ distinct sets of supports, computing $\phi_1$ requires exponential running time.

## 3.2 Computational Limits

In this section, we characterize the computationally tractable minimax rate $\overline{\gamma}_n^*$ given in Definition 2.3. Moreover, we focus on the setting where $\boldsymbol{\Sigma}$ is known a priori and the parameter spaces for the null and alternative hypotheses are defined in (3.1) and (3.2), respectively. The main result is that, in highly sparse setting where $s = o(\sqrt{d})$, we have

$$\overline{\gamma}_n^* = \sqrt{s^2/n} \wedge (1/\alpha^2 \cdot s \log d/n). \tag{3.11}$$

We first present the lower bound in the next result.

**Theorem 3.3.** *For the testing problem in* (3.3) *with* $\boldsymbol{\Sigma}$ *known a priori, we make the same assumptions as in Theorem 3.1. For any sequence* $\{\gamma_n\}_{n \geq 1}$ *such that*

$$\gamma_n = o\left\{\gamma_n^* \vee \left[\sqrt{s^2/n} \wedge (1/\alpha^2 \cdot s/n)\right]\right\}, \tag{3.12}$$

*where* $\gamma_n^*$ *is defined in* (3.4)*, any computationally tractable test is asymptotically powerless under the statistical query model. That is, for any constant* $\eta > 0$ *and any* $\mathscr{A} \in \mathcal{A}(d^\eta)$*, there exists an oracle* $r \in \mathcal{R}[\xi, n, T_n, \eta(\mathcal{Q}_{\mathscr{A}})]$ *such that* $\lim_{n \to \infty} \overline{R}_n^*[\mathcal{G}_0(\boldsymbol{\Sigma}), \mathcal{G}_1(\boldsymbol{\Sigma}, \gamma_n); \mathscr{A}, r] = 1$.

We remark that the lower bound in (3.12) differs from $\gamma_n^*$ in (3.11) by a logarithmic term when $\sqrt{1/n} \leq \alpha^2 \leq \sqrt{s \log d/n}$. We expect this gap to be eliminated by more delicate analysis under the statistical query model.

Now putting Theorems 3.1 and 3.3 together, we describe the "more supervision, less computation" phenomenon as follows.

(i) When $0 \leq \alpha \leq (\log^2 d/n)^{1/4}$, the computational lower bound implies that the uncorrupted labels are unable to improve the quality of computationally tractable detection compared with the unsupervised setting. In addition, in this region, the gap between $\gamma_n^*$ and $\overline{\gamma}_n^*$ remains the same.

(ii) When $(\log^2 d/n)^{1/4} < \alpha \leq (s \log d/n)^{1/4}$, the information-theoretic lower bound shows that the uncorrupted labels cannot improve the quality of detection compared with unsupervised setting. However, more uncorrupted labels improve the statistical performances of hypothesis tests that are computationally tractable by shrinking the gap between $\gamma_n^*$ and $\overline{\gamma}_n^*$.

(iii) When $(s \log d/n)^{1/4} < \alpha \leq 1$, having more uncorrupted labels improves both statistical optimality and the computational efficiency. In specific, in this case, the gap between $\gamma_n^*$ and $\overline{\gamma}_n^*$ vanishes and we have $\gamma_n^* = \overline{\gamma}_n^* = 1/\alpha^2 \cdot s \log d/n$.

Now we derive a nearly matching upper bound under the statistical query model, which establishes the computationally tractable minimax rate together with Theorem 3.3. We construct a computationally efficient testing procedure that combines two test functions which yields the two parts in $\overline{\gamma}_n^*$ respectively. Similar to $\phi_1$ defined in (3.7), the first test function discards the information of labels, which works for the purely unsupervised setting where $\alpha = 0$. For $j \in [d]$, we denote by $\sigma_j$ the $j$-th diagonal element of $\boldsymbol{\Sigma}$. Under the statistical query model, we consider the $2d$ query functions

$$q_j(y, \mathbf{x}) := x_j/\sqrt{\sigma_j} \cdot \mathbb{1}\{|x_j/\sqrt{\sigma_j}| \leq R \cdot \sqrt{\log d}\}, \tag{3.13}$$

$$\widetilde{q}_j(y, \mathbf{x}) := (x_j^2/\sigma_j - 1) \cdot \mathbb{1}\{|x_j/\sqrt{\sigma_j}| \leq R \cdot \sqrt{\log d}\}, \text{ for all } j \in [d], \tag{3.14}$$

where $R > 0$ is an absolute constant. Here we apply truncation to the query functions to obtain bounded queries, which is specified by the statistical query model in Definition 2.2. We denote by $z_{q_j}$ and $z_{\widetilde{q}_j}$ the realizations of the random variables output by the statistical oracle for query functions $q_j$ and $\widetilde{q}_j$, respectively. As for the second test function, similar to (3.8), we consider

$$\overline{q}_{\mathbf{v}}(y, \mathbf{x}) = (2y - 1) \cdot \mathbf{v}^\top \mathrm{diag}(\mathbf{\Sigma})^{-1/2} \mathbf{x} \cdot \mathbb{1}\left\{ |\mathbf{v}^\top \mathrm{diag}(\mathbf{\Sigma})^{-1/2} \mathbf{x}| \leq R \cdot \sqrt{\log d} \right\} \qquad (3.15)$$

for all $\mathbf{v} \in \mathcal{B}_2(1)$. We denote by $Z_{\overline{q}_{\mathbf{v}}}$ the output of the statistical oracle corresponding to query function $\overline{q}_{\mathbf{v}}$. With these $4d$ query functions, we introduce test functions

$$\overline{\phi}_1 := \mathbb{1}\left\{ \sup_{j \in [d]} (z_{\widetilde{q}_j} - z_{q_j}^2) \geq C\overline{\tau}_1 \right\}, \quad \overline{\phi}_2 := \mathbb{1}\left\{ \sup_{\mathbf{v} \in \mathcal{B}_2(1)} z_{\overline{q}_{\mathbf{v}}} \geq 2\overline{\tau}_2 \right\}, \qquad (3.16)$$

where $\overline{\tau}_1$ and $\tau_2$ are positive parameters that will be specified later and $C$ is an absolute constant.

**Theorem 3.4.** *For the test functions $\overline{\phi}_1$ and $\overline{\phi}_2$ defined in (3.16), we define the ultimate test function as $\overline{\phi} = \overline{\phi}_1 \vee \overline{\phi}_2$. We set*

$$\overline{\tau}_1 = R^2 \log d \cdot \sqrt{\log(4d/\xi)/n}, \quad \overline{\tau}_2 = R\sqrt{\log d} \cdot \sqrt{\log(4d/\xi)/n}, \qquad (3.17)$$

*where $\xi = o(1)$. For the hypothesis testing problem in (3.3), we further assume that $\|\boldsymbol{\mu}_0\|_\infty \vee \|\boldsymbol{\mu}_1\|_\infty \leq C_0$ for some constant $C_0 > 0$. Under the assumption that*

$$\sup_{j \in [d]} (\mu_{0,j} - \mu_{1,j})^2 / \sigma_j = \Omega\left\{ \left[ 1/\alpha^2 \cdot \log^2 d \cdot \log(d/\xi)/n \right] \wedge \log d \cdot \sqrt{\log(d/\xi)/n} \right\}, \qquad (3.18)$$

*the risk of $\overline{\phi}$ satisfies that $\overline{R}_n^*(\overline{\phi}) = \sup_{\boldsymbol{\theta} \in \mathcal{G}_0(\mathbf{\Sigma})} \overline{\mathbb{P}}_{\boldsymbol{\theta}}(\overline{\phi} = 1) + \sup_{\boldsymbol{\theta} \in \mathcal{G}_1(\mathbf{\Sigma}, \gamma_n)} \overline{\mathbb{P}}_{\boldsymbol{\theta}}(\overline{\phi} = 0) \leq 5\xi$. Here we denote by $\mu_{0,j}$ and $\mu_{1,j}$ the $j$-th entry of $\boldsymbol{\mu}_0$ and $\boldsymbol{\mu}_1$, respectively.*

If we set the tail probability of the statistical query model to be $\xi = 1/d$, (3.18) shows that $\overline{\phi}$ is asymptotically powerful if $\sup_{j \in [d]} (\mu_{0,j} - \mu_{1,j})^2 / \sigma_j = \Omega[(1/\alpha^2 \cdot \log^3 d/n) \wedge (\log^3 d/n)^{1/2}]$. When the energy of $\boldsymbol{\mu}_0 - \boldsymbol{\mu}_1$ is spread over its support, $\|\boldsymbol{\mu}_0 - \boldsymbol{\mu}_1\|_\infty$ and $\|\boldsymbol{\mu}_0 - \boldsymbol{\mu}_1\|_2/\sqrt{s}$ are close. Under the assumption that the condition number $\kappa$ of $\mathbf{\Sigma}$ is a constant, (3.18) is implied by

$$\gamma_n \gtrsim (s^2 \log^3 d/n)^{1/2} \wedge (1/\alpha^2 \cdot s \log^3 d/n).$$

Compared with Theorem 3.3, the above upper bound matches the computational lower bound up to a logarithmic factor and $\overline{\gamma}_n^*$ is between $\sqrt{s^2/n} \wedge (1/\alpha^2 \cdot s \log d/n)$ and $(s^2 \log^3 d/n)^{1/2} \wedge (1/\alpha^2 \cdot s \log^3 d/n)$. Note that the truncation on query functions in (3.13) and (3.14) yields an additional logarithmic term, which could be reduced to $(s^2 \log d/n)^{1/2} \wedge (1/\alpha^2 \cdot s \log d/n)$ using more delicate analysis. Moreover, the test function $\overline{\phi}_1$ is essentially based on a diagonal thresholding algorithm performed on the covariance matrix of $\boldsymbol{X}$. The work in [6] provides a more delicate analysis of this algorithm which establishes the $\sqrt{s^2/n}$ rate. Their algorithm can also be formulated into the statistical query model; we use the simpler version in (3.16) for ease of presentation. Therefore, with more sophicated proof techinque, it can be shown that $\sqrt{s^2/n} \wedge (1/\alpha^2 \cdot s \log d/n)$ is the critical threshold for asymptotically powerful detection with computational efficiency.

### 3.3 Implication for Estimation

Our aforementioned phase transition in the detection problems directly implies the statistical and computational trade-offs in the problem of estimation. We consider the problem of estimating the parameter $\Delta\boldsymbol{\mu} = \boldsymbol{\mu}_0 - \boldsymbol{\mu}_1$ of the binary classification model in (2.1) and (2.2), where $\Delta\boldsymbol{\mu}$ is $s$-sparse and $\mathbf{\Sigma}$ is known a priori. We assume that the signal to noise ratio is $\rho(\boldsymbol{\theta}) = \Delta\boldsymbol{\mu}^\top \mathbf{\Sigma}^{-1} \Delta\boldsymbol{\mu} \geq \gamma_n = o(\overline{\gamma}_n^*)$. For any constant $\eta > 0$ and any $\mathscr{A} \in \mathcal{A}(T)$ with $T = O(d^\eta)$, suppose we obtain an estimator $\Delta\widehat{\boldsymbol{\mu}}$ of $\Delta\boldsymbol{\mu}$ by algorithm $\mathscr{A}$ under the statistical query model. If $\Delta\widehat{\boldsymbol{\mu}}$ converges to $\Delta\boldsymbol{\mu}$ in the sense that

$$(\Delta\widehat{\boldsymbol{\mu}} - \Delta\boldsymbol{\mu})^\top \mathbf{\Sigma}^{-1} (\Delta\widehat{\boldsymbol{\mu}} - \Delta\boldsymbol{\mu}) = o[\gamma_n^2/\rho(\boldsymbol{\theta})],$$

we have $|\Delta\widehat{\boldsymbol{\mu}}^\top \mathbf{\Sigma}^{-1} \Delta\widehat{\boldsymbol{\mu}} - \Delta\boldsymbol{\mu}^\top \mathbf{\Sigma}^{-1} \Delta\boldsymbol{\mu}| = o(\gamma_n)$. Thus the test function $\phi = \mathbb{1}\{\Delta\widehat{\boldsymbol{\mu}}^\top \mathbf{\Sigma}\Delta\widehat{\boldsymbol{\mu}} \geq \gamma_n/2\}$ is asymptotically powerful, which contradicts the computational lower bound in Theorem 3.3. Therefore, there exists a constant $C$ such that $(\Delta\widehat{\boldsymbol{\mu}} - \Delta\boldsymbol{\mu})^\top \mathbf{\Sigma}^{-1} (\Delta\widehat{\boldsymbol{\mu}} - \Delta\boldsymbol{\mu}) \geq C\gamma_n^2/\rho(\boldsymbol{\theta})$ for any estimator $\Delta\widehat{\boldsymbol{\mu}}$ constructed from polynomial number of queries.

### Acknowledgments

We would like to thank Vitaly Feldman for valuable discussions.

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
