[Supplementary Material 1]

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

# A  Proofs of the Main Results

## A.1  Proof of Theorem 3.1

In this section, we prove the information-theoretic lower bound. In specific, we focus on the restricted testing problem

$$H_0 : \boldsymbol{\theta} = (\mathbf{0}, \mathbf{0}, \mathbf{I}, \alpha) \text{ versus. } H_1 : \boldsymbol{\theta} = (-\mathbf{v}/2, \mathbf{v}/2, \mathbf{I}, \alpha), \tag{A.1}$$

where

$$\mathbf{v} \in \mathcal{H}(s) := \{\mathbf{u} \in \{0, \beta\}^d : \|\mathbf{u}\|_0 = s\}.$$

Here we set $s\beta^2 = \gamma_n$ to ensure that $(-\mathbf{v}/2, \mathbf{v}/2, \mathbf{I}, \alpha)$ belongs to the alternative parameter space $\mathcal{G}(\boldsymbol{\Sigma}; \gamma_n)$. For notational simplicity, we denote the distribution of model $(-\mathbf{v}/2, \mathbf{v}/2, \mathbf{I}, \alpha)$ by $\mathbb{P}_{\mathbf{v}}$ and the product distribution of $n$ i.i.d. samples by $\mathbb{P}_{\mathbf{v}}^n$. By the definition of the minimax risk in (2.4), we have

$$\sup_{\boldsymbol{\Sigma}} R_n^* [\mathcal{G}_0(\boldsymbol{\Sigma}), \mathcal{G}_1(\boldsymbol{\Sigma}; \gamma_n)] \geq \inf_{\phi} \left[ \mathbb{P}_{\mathbf{0}}^n(\phi = 1) + \frac{1}{|\mathcal{H}(s)|} \sum_{\mathbf{v} \in \mathcal{H}(s)} \mathbb{P}_{\mathbf{v}}^n(\phi = 0) \right].$$

We thus reduce the minimax risk to the risk of a simple-against-simple hypothesis test where the alternative hypothesis corresponds to a uniform mixture of $\{\mathbb{P}_{\mathbf{v}} : \mathbf{v} \in \mathcal{H}(s)\}$. For notational simplicity, we define $\mathbb{P}_{\mathcal{H}}^n := 1/|\mathcal{H}(s)| \cdot \sum_{\mathbf{v} \in \mathcal{H}(s)} \mathbb{P}_{\mathbf{v}}^n$. By Neyman-Pearson Lemma, we have

$$R_n^* [\mathcal{G}_0, \mathcal{G}_1(\boldsymbol{\Sigma}; \gamma_n)] \geq 1 - \mathrm{TV}(\mathbb{P}_{\mathbf{0}}^n, \mathbb{P}_{\mathcal{H}}^n).$$

Using Pinsker's inequality $\mathrm{TV}(\mathbb{P}_{\mathbf{0}}^n, \mathbb{P}_{\mathcal{H}}^n) \leq \sqrt{D_{\chi^2}(\mathbb{P}_{\mathcal{H}}^n, \mathbb{P}_{\mathbf{0}}^n)}$, for showing $R_n^*[\mathcal{G}_0(\boldsymbol{\Sigma}), \mathcal{G}_1(\boldsymbol{\Sigma}; \gamma_n)] \to 1$ as $n$ goes to infinity, it suffices to show that $D_{\chi^2}(\mathbb{P}_{\mathcal{H}}^n, \mathbb{P}_{\mathbf{0}}^n) = o(1)$. By calculation we have

$$D_{\chi^2}(\mathbb{P}_{\mathcal{H}}^n, \mathbb{P}_{\mathbf{0}}^n) = \mathbb{E}_{\mathbb{P}_{\mathbf{0}}^n} \left\{ \left[ \frac{\mathrm{d}\mathbb{P}_{\mathcal{H}}^n}{\mathrm{d}\mathbb{P}_{\mathbf{0}}^n}(Y, \boldsymbol{X}) - 1 \right]^2 \right\} = \mathbb{E}_{\mathbb{P}_{\mathbf{0}}^n} \left\{ \left[ \frac{\mathrm{d}\mathbb{P}_{\mathcal{H}}^n}{\mathrm{d}\mathbb{P}_{\mathbf{0}}^n}(Y, \boldsymbol{X}) \right]^2 \right\} - 1$$

$$= \frac{1}{|\mathcal{H}(s)|^2} \sum_{\mathbf{v}_1, \mathbf{v}_2 \in \mathcal{H}(s)} \mathbb{E}_{\mathbb{P}_{\mathbf{0}}^n} \left[ \frac{\mathrm{d}\mathbb{P}_{\mathbf{v}_1}^n \mathrm{d}\mathbb{P}_{\mathbf{v}_2}^n}{\mathrm{d}\mathbb{P}_{\mathbf{0}}^n \mathrm{d}\mathbb{P}_{\mathbf{0}}^n}(Y, \boldsymbol{X}) \right] - 1$$

$$= \frac{1}{|\mathcal{H}(s)|^2} \sum_{\mathbf{v}_1, \mathbf{v}_2 \in \mathcal{H}(s)} \left\{ \mathbb{E}_{\mathbb{P}_{\mathbf{0}}} \left[ \frac{\mathrm{d}\mathbb{P}_{\mathbf{v}_1} \mathrm{d}\mathbb{P}_{\mathbf{v}_2}}{\mathrm{d}\mathbb{P}_{\mathbf{0}} \mathrm{d}\mathbb{P}_{\mathbf{0}}}(Y, \boldsymbol{X}) \right] \right\}^n - 1. \tag{A.2}$$

We utilize the following lemma to obtain an upper bound for the last term of (A.2). See §B.1 for the proof.

**Lemma A.1.** *For any $\mathbf{v}_1, \mathbf{v}_2 \in \mathcal{H}(s)$, we have*

$$\mathbb{E}_{\mathbb{P}_{\mathbf{0}}} \left[ \frac{\mathrm{d}\mathbb{P}_{\mathbf{v}_1}}{\mathrm{d}\mathbb{P}_{\mathbf{0}}} \frac{\mathrm{d}\mathbb{P}_{\mathbf{v}_2}}{\mathrm{d}\mathbb{P}_{\mathbf{0}}}(Y, \boldsymbol{X}) \right] = \cosh\left(\langle \mathbf{v}_1, \mathbf{v}_2 \rangle / 2\right) + \alpha^2 \sinh\left(\langle \mathbf{v}_1, \mathbf{v}_2 \rangle / 2\right).$$

By Lemma A.1, we have

$$D_{\chi^2}(\mathbb{P}_{\mathcal{H}}^n, \mathbb{P}_{\mathbf{0}}^n)$$
$$= \frac{1}{|\mathcal{H}(s)|^2} \sum_{\mathbf{v}_1, \mathbf{v}_2 \in \mathcal{H}(s)} \left[ \cosh\left(1/2 \cdot \langle \mathbf{v}_1, \mathbf{v}_2 \rangle\right) + \alpha^2 \sinh\left(1/2 \cdot \langle \mathbf{v}_1, \mathbf{v}_2 \rangle\right) \right]^n - 1. \tag{A.3}$$

We define $\mathcal{C} := \{\mathcal{S} \subseteq [d] : |\mathcal{S}| = s\}$, and let $\mathbb{U}_{\mathcal{C}}$ be the uniform distribution over $\mathcal{C}$. Let $\mathcal{S}_1, \mathcal{S}_2 \sim \mathbb{U}_{\mathcal{C}}$ be two independent random sets. Then by (A.3), we have

$$D_{\chi^2}(\mathbb{P}_{\mathcal{H}}^n, \mathbb{P}_{\mathbf{0}}^n) = \mathbb{E}_{\mathcal{S}_1, \mathcal{S}_2} \left[ \cosh(\beta^2/2 \cdot |\mathcal{S}_1 \cap \mathcal{S}_2|) + \alpha^2 \sinh(\beta^2/2 \cdot |\mathcal{S}_1 \cap \mathcal{S}_2|) \right]^n - 1.$$

We use the next lemma, proved in §B.2, to bound the above right-hand side.

**Lemma A.2.** *For any $x \geq 0$ and $v \in [0, 1]$, we have*

$$\cosh(x) + v \sinh(x) \leq \exp(2vx) \vee \cosh(2x). \tag{A.4}$$

Proceeding with this result and letting random variable $Z \sim |\mathcal{S}_1 \cap \mathcal{S}_2|$, we have

$$
\begin{aligned}
D_{\chi^2}(\mathbb{P}_{\mathcal{H}}^n, \mathbb{P}_{\mathbf{0}}^n) &\leq \mathbb{E}_Z \left[ \exp(\alpha^2\beta^2 Z) \vee \cosh(\beta^2 Z) \right]^n - 1 \\
&= \mathbb{E}_Z \left[ \exp(n\alpha^2\beta^2 Z) \vee \cosh(\beta^2 Z)^n \right] - 1 \\
&= \mathbb{E}_Z \left\{ \exp(n\alpha^2\beta^2 Z) \vee \mathbb{E}_U \left[ \exp(\beta^2 ZU) \right] \right\} - 1,
\end{aligned}
\tag{A.5}
$$

where in the last step, we introduce a random variable $U$ that is the summation of $n$ independent Rademacher random variables over $\{-1, 1\}$. Then we have $\cosh(\beta^2 Z)^n = \mathbb{E}_U[\exp(\beta^2 ZU)]$. By (A.5), we have

$$
\begin{aligned}
D_{\chi^2}(\mathbb{P}_{\mathcal{H}}^n, \mathbb{P}_{\mathbf{0}}^n) &\leq \mathbb{E}_Z \mathbb{E}_U \left[ \exp(n\alpha^2\beta^2 Z) \vee \exp(\beta^2 ZU) \right] - 1 \\
&= \mathbb{E}_U \mathbb{E}_Z \left\{ \exp(n\alpha^2\beta^2) \vee \exp(\beta^2 U) \right\}^Z - 1 \\
&\leq \mathbb{E}_U \left\{ \sup_{\mathcal{S}_1 \in \mathcal{C}} \mathbb{E}_{\mathcal{S}_2} \left[ \exp(n\alpha^2\beta^2) \vee \exp(\beta^2 U) \right]^{|\mathcal{S}_1 \cap \mathcal{S}_2|} \right\} - 1.
\end{aligned}
\tag{A.6}
$$

Now we turn to bound the expectation over $\mathcal{S}_2$ in (A.6). For any fixed $\mathcal{S}_1$, we have

$$
|\mathcal{S}_1 \cap \mathcal{S}_2| = \sum_{i \in \mathcal{S}_1} V_i,
$$

where $V_i$ is binary random variable that indicates whether $i \in \mathcal{S}_2$. It is known that $V_1, \dots, V_d$ are negative associated. Hence we have

$$
\begin{aligned}
\mathbb{E}_{\mathcal{S}_2} \left[ \exp(n\alpha^2\beta^2) \vee \exp(\beta^2 U) \right]^{|\mathcal{S}_1 \cap \mathcal{S}_2|} &\leq \prod_{i \in \mathcal{S}_1} \mathbb{E}_{V_i} \left[ \exp(n\alpha^2\beta^2) \vee \exp(\beta^2 U) \right]^{V_i} \\
&= \left\{ 1 + s/d \cdot \left[ \exp(n\alpha^2\beta^2) \vee \exp(\beta^2 U) - 1 \right] \right\}^s.
\end{aligned}
\tag{A.7}
$$

Plugging (A.7) into (A.6) and expanding the polynomial term, we have

$$
\begin{aligned}
D_{\chi^2}(\mathbb{P}_{\mathcal{H}}^n, \mathbb{P}_{\mathbf{0}}^n) &\leq \sum_{k=1}^s \binom{s}{k} \cdot (s/d)^k \cdot \mathbb{E}_U \left[ \exp(n\alpha^2\beta^2) \vee \exp(\beta^2 U) - 1 \right]^k \\
&= \sum_{k=1}^s \binom{s}{k} \cdot (s/d)^k \cdot \Big( \left[ \exp(n\alpha^2\beta^2) - 1 \right]^k \cdot \mathbb{P}(U < n\alpha^2) \\
&\qquad + \mathbb{E}_U \left\{ \left[ \exp(\beta^2 U) - 1 \right]^k \mid U \geq \alpha^2 n \right\} \cdot \mathbb{P}(U \geq n\alpha^2) \Big), \\
&\leq T_1 + T_2,
\end{aligned}
$$

where $T_1$ and $T_2$ are defined as

$$
T_1 := \sum_{k=1}^s \binom{s}{k} \cdot (s/d)^k \cdot \left[ \exp(n\alpha^2\beta^2) - 1 \right]^k
$$

$$
T_2 := \sum_{k=1}^s \binom{s}{k} \cdot (s/d)^k \cdot \mathbb{E}_U \left\{ \left[ \exp(\beta^2 U) - 1 \right]^k \mid U \geq 0 \right\} \cdot \mathbb{P}(U \geq 0).
$$

It remains to bound $T_1$ and $T_2$ respectively.

**Bounding $T_1$.** Under condition $s\beta^2 = \gamma_n = o(1/\alpha^2 \cdot s \log d/n)$, we have $\beta^2 = o(1/\alpha^2 \cdot \log d/n)$. Hence, for any small constant $C > 0$, we have $\beta^2 \leq C \cdot 1/\alpha^2 \cdot \log d/n$ when $n$ is sufficiently large. Note that we assume $s = o(d^{1/2-\delta})$ for some fixed constant $\delta > 0$. Then we have

$$
\begin{aligned}
T_1 &\leq \sum_{k=1}^s \binom{s}{k} \cdot (s/d)^k \cdot \exp(\alpha^2\beta^2 nk) \leq \sum_{k=1}^s \left[ s^2 e/(kd) \right]^k \cdot \exp(\alpha^2\beta^2 nk) \\
&\leq \sum_{k=1}^s \left[ s^2 e/(kd) \right]^k \cdot \exp(Ck \log d) = \sum_{k=1}^s (s^2 e/k \cdot d^{C-1})^k \leq \sum_{k=1}^s (e/k \cdot d^{C-2\delta})^k,
\end{aligned}
$$

where the second step follows from the fact that $\binom{s}{k} \leq (es/k)^k$. Note that $C$ is chosen arbitrarily, hence we can always choose $C \leq \delta$. It implies that $e/k \cdot d^{C-2\delta} = o(1)$. We thus conclude $T_1 = o(1)$.

**Bounding $T_2$.** For term $T_2$, we observe that

$$T_2 \leq \sum_{k=1}^{s} (e/k \cdot s^2/d)^k \cdot \mathbb{E}_U \left\{ \left[ \exp(\beta^2 |U|) - 1 \right]^k \right\}$$

$$\leq \sum_{k=1}^{s} (e/k \cdot s^2/d)^k \cdot \mathbb{E}_U \left[ (\beta^2 |U|)^k + \exp(\beta^2 k |U|) \cdot \mathbb{1}(\beta^2 |U| \geq 1) \right]$$

$$\leq T_3 + T_4,$$

where $T_3$ and $T_4$ are defined as

$$T_3 := \sum_{k=1}^{s} \mathbb{E}_U (e/k \cdot s^2 \beta^2/d \cdot |U|)^k,$$

$$T_4 := \sum_{k=1}^{s} (e/k \cdot s^2/d)^k \cdot \mathbb{E}_U \left[ \exp(\beta^2 k |U|) \cdot \mathbb{1}(\beta^2 |U| \geq 1) \right].$$

Note that $U$ is summation of $n$ i.i.d. centered sub-Gaussian random variables $U_i$ each with Orlicz $\psi_2$-norm equal to one. Therefore, $U$ is also centered sub-Gaussian random variable with $||U||_{\psi_2} \leq C\sqrt{n}$ for some constant $C$. Thus it holds that

$$\mathbb{E}(|U|^k) \leq (\sqrt{k} \cdot ||U||_{\psi_2})^k \leq (C\sqrt{nk})^k.$$

Hence for term $T_3$, we have

$$T_3 \leq \sum_{k=1}^{s} \left[ Ces^2 \beta^2 \sqrt{n}/(\sqrt{k}d) \right]^k,$$

Under the condition $s\beta^2 = o(\sqrt{s \log d/n})$, we have

$$Ces^2 \beta^2 \sqrt{n}/(\sqrt{k}d) = o\left( s\sqrt{s \log d}/d \right).$$

Since $s = o(\sqrt{d})$, we have $s\sqrt{s \log d}/d = o(1)$, which implies $T_3 = o(1)$.

To obtain an upper bound for term $T_4$, we let $W = \beta^2 U$. So $W$ is centered sub-Gaussian with Orlicz norm $c\beta^2 \sqrt{n}$. Computing integral by parts, we have

$$\mathbb{E}_U \left[ \exp(\beta^2 k |U|) \cdot \mathbb{1}(\beta^2 |U| \geq 1) \right] = e^k \cdot \mathbb{P}(|W| \geq 1) + \int_{w=1}^{\infty} k e^{wk} \cdot \mathbb{P}(|W| \geq w) \mathrm{d}w. \quad \text{(A.8)}$$

Using the property of sub-Gaussianity, we have $\mathbb{P}[W \geq t] \leq C_1 \exp[-C_2 t^2/(\beta^2 \sqrt{n})^2]$ for some absolute constants $C_1, C_2 > 0$. Proceeding with (A.8) and using shorthand $\sigma = \beta^2 \sqrt{n}$, we obtain

$$\mathbb{E}_U \left[ \exp(\beta^2 k |U|) \cdot \mathbb{1}(\beta^2 |U| \geq 1) \right] \leq C_1 e^k e^{-C_2/\sigma^2} + C_1 k \int_{w=1}^{\infty} e^{wk} e^{-C_2 w^2/\sigma^2} \mathrm{d}w$$

$$= C_1 e^k e^{-C_2/\sigma^2} + C_1 k e^{k^2 \sigma^2/(4C_2)} \int_{w=1}^{\infty} e^{-\frac{C_2}{\sigma^2}(w - \frac{k\sigma^2}{2C_2})^2} \mathrm{d}w \leq C_1 e^k + C_3 k e^{k^2 \sigma^2/(4C_2)} \sigma,$$

where $C_3$ is a constant that depends on $C_1$ and $C_2$. Thus we have

$$T_4 \leq \underbrace{\sum_{k=1}^{s} C_1 \left[ s^2 e^2/(kd) \right]^k}_{T_5} + \underbrace{\sum_{k=1}^{s} C_3 \sigma k \left[ s^2 e^2/(kd) \cdot \exp(k/4 \cdot \sigma^2/C_2) \right]^k}_{T_6}. \quad \text{(A.9)}$$

Note that $s^2/d = o(1)$, we thus have $T_5 = o(1)$. Under condition $s\beta^2 = o(\sqrt{s \log d/n})$, for any small constant $C > 0$, when $n$ is large enough, we have

$$\exp(k/4 \cdot \sigma^2/C_2) \leq \exp(Ck \log d/s) \leq \exp(C \log d) \leq d^C.$$

Plugging (A.9) into $T_6$ and using $s^2 = o(d^{1-2\delta})$, we have that each term in the summation is less that

$$T_6 \leq \sum_{k=1}^{s} \sigma k \left[ e^2/(kd^{2\delta-C}) \right]^k \lesssim \sum_{k=1}^{s} k\sqrt{\log d/s} \cdot \left[ e^2/(d^{2\delta-C}) \right]^k.$$

Since the constant $C$ is chosen arbitrarily, we have $T_6 = o(1)$. Accordingly, $T_4 = o(1)$ and $T_2 = o(1)$.

Finally, combining everything together, we have $D_{\chi^2}(\mathbb{P}_{\mathcal{H}}^n, \mathbb{P}_{\mathbf{0}}^n) = o(1)$, which completes the proof.

## A.2 Proof of Theorem 3.2

We begin with some basic properties of sample sets $\{\mathbf{w}_i\}_{i=1}^n$ and $\{\mathbf{u}_i\}_{i=1}^n$. We introduce the random vector $\mathbf{W} := \mathbf{X} - \mathbf{X}'$ to capture the distribution of samples $\{\mathbf{w}_i\}_{i=1}^n$. Here $\mathbf{X}$ follows the model given in (2.1)-(2.2), and $\mathbf{X}'$ is an independent copy of $\mathbf{X}$. We note that the marginal distribution of $\mathbf{X}$ is given by $1/2 \cdot \mathcal{N}(\boldsymbol{\mu}_0, \boldsymbol{\Sigma}) + 1/2 \cdot \mathcal{N}(\boldsymbol{\mu}_1, \boldsymbol{\Sigma})$. Thus $\mathbf{W}$ follows a mixture distribution

$$\mathbf{W} \sim 1/2 \cdot \mathcal{N}(\mathbf{0}, 2\boldsymbol{\Sigma}) + 1/4 \cdot \mathcal{N}(\boldsymbol{\mu}_1 - \boldsymbol{\mu}_0, 2\boldsymbol{\Sigma}) + 1/4 \cdot \mathcal{N}(\boldsymbol{\mu}_0 - \boldsymbol{\mu}_1, 2\boldsymbol{\Sigma}). \tag{A.10}$$

Moreover, conditioning on the observed label $Y$, the distribution of $\mathbf{X}$ is given by

$$\mathbf{X}|Y = 0 \ \sim \ (1 + \alpha)/2 \cdot \mathcal{N}(\boldsymbol{\mu}_0, \boldsymbol{\Sigma}) + (1 - \alpha)/2 \cdot \mathcal{N}(\boldsymbol{\mu}_1, \boldsymbol{\Sigma}), \tag{A.11}$$

$$\mathbf{X}|Y = 1 \ \sim \ (1 + \alpha)/2 \cdot \mathcal{N}(\boldsymbol{\mu}_1, \boldsymbol{\Sigma}) + (1 - \alpha)/2 \cdot \mathcal{N}(\boldsymbol{\mu}_0, \boldsymbol{\Sigma}). \tag{A.12}$$

We introduce a random vector $\mathbf{U} := \mathbf{X}^{(1)} - \mathbf{X}^{(0)}$ that corresponds to samples $\{\mathbf{u}_i\}_{i=1}^n$. Here random vectors $\mathbf{X}^{(0)}$ and $\mathbf{X}^{(1)}$ are independent and have distributions given in (A.11), (A.12), respectively. The distribution of $\mathbf{U}$ is given by

$$\mathbf{U} \sim (1+\alpha)^2/4 \cdot \mathcal{N}(\boldsymbol{\mu}_1 - \boldsymbol{\mu}_0, 2\boldsymbol{\Sigma}) + (1-\alpha^2)/2 \cdot \mathcal{N}(\mathbf{0}, 2\boldsymbol{\Sigma}) + (1-\alpha)^2/4 \cdot \mathcal{N}(\boldsymbol{\mu}_0 - \boldsymbol{\mu}_1, 2\boldsymbol{\Sigma}). \tag{A.13}$$

Now we turn to prove Theorem 3.2. It suffices to prove this result by bounding type-I and type-II errors separately. In the end, we will show that

$$\sup_{\boldsymbol{\theta} \in \mathcal{G}_0(\boldsymbol{\Sigma})} \mathbb{P}_{\boldsymbol{\theta}}^n(\phi = 1) \leq 4d^{-1} \quad \text{and} \quad \sup_{\boldsymbol{\theta} \in \mathcal{G}_1(\boldsymbol{\Sigma}; \gamma_n)} \mathbb{P}_{\boldsymbol{\theta}}^n(\phi = 0) \leq 16d^{-1}.$$

**Type-I error.** Under the null hypothesis $\boldsymbol{\theta} \in \mathcal{G}_0(\boldsymbol{\Sigma})$, (A.10) and (A.13) reduce to

$$\mathbf{W} \sim \mathcal{N}(\mathbf{0}, 2\boldsymbol{\Sigma}), \ \ \mathbf{U} \sim \mathcal{N}(\mathbf{0}, 2\boldsymbol{\Sigma}).$$

To bound the type-I error of function $\phi_1$, we first note that

$$\frac{1}{n} \sum_{i=1}^n (\mathbf{v}^\top \boldsymbol{\Sigma}^{-1} \mathbf{w}_i)^2 = \mathbf{v}^\top \widehat{\boldsymbol{\Sigma}}_W \mathbf{v},$$

where we let $\widehat{\boldsymbol{\Sigma}}_W := 1/n \cdot \sum_{i=1}^n \boldsymbol{\Sigma}^{-1} \mathbf{w}_i \mathbf{w}_i^\top \boldsymbol{\Sigma}^{-1}$, i.e., an empirical covariance matrix of random vector $\boldsymbol{\Sigma}^{-1}\mathbf{W} \sim \mathcal{N}(\mathbf{0}, 2\boldsymbol{\Sigma}^{-1})$. For any matrix $\mathbf{A} \in \mathbb{R}^{d \times d}$ and $\mathcal{S} \subseteq [d]$, we let $[\mathbf{A}]_{\mathcal{S}} \in \mathbb{R}^{|\mathcal{S}| \times |\mathcal{S}|}$ be the submatrix of $\mathbf{A}$, which contains the entries with row and column indices in $\mathcal{S}$. By standard tail bound of Gaussian covariance estimation (see Lemma C.2), for any fixed $\mathcal{S} \in [d]$ with $|\mathcal{S}| = s$, and any $\epsilon \in (0, 1)$, when $n \geq Cs/\epsilon^2$ for some constant $C$, we have

$$\mathbb{P}_{\boldsymbol{\theta}}^n \left[ \|(\widehat{\boldsymbol{\Sigma}}_W - 2\boldsymbol{\Sigma}^{-1})_{\mathcal{S}}\|_2 \geq 2\epsilon\|(\boldsymbol{\Sigma}^{-1})_{\mathcal{S}}\|_2 \right] \leq 2e^{-n}. \tag{A.14}$$

Note that $\|(\boldsymbol{\Sigma}^{-1})_{\mathcal{S}}\|_2 \leq \|\boldsymbol{\Sigma}^{-1}\|_2$ for all $\mathcal{S} \subseteq [d]$. By taking union bound over all subsets with size $s$ in $[d]$, we have

$$\mathbb{P}_{\boldsymbol{\theta}}^n \left[ \sup_{\mathcal{S} \in [d], |\mathcal{S}| = s} \|(\widehat{\boldsymbol{\Sigma}}_W - 2\boldsymbol{\Sigma}^{-1})_{\mathcal{S}}\|_2 \geq 2\epsilon\|\boldsymbol{\Sigma}^{-1}\|_2 \right] \leq 2\binom{d}{s} e^{-n}$$

$$\overset{(a)}{\leq} 2 \exp\left[-n + s\log(ed/s)\right] \overset{(b)}{\leq} 2[s/(ed)]^s \leq 2d^{-1}.$$

Here step $(a)$ follows from the fact that $\binom{d}{s} \leq (ed/s)^s$ and step $(b)$ follows from the assumption that $n \geq 2s\log(ed/s)$. In the last step we use the fact that function $f(s) = (s/d)^s$ is monotonically decreasing for $s \in [1, d/e]$. We set $\epsilon = \sqrt{s\log(ed/s)/n}$. Under condition $n \geq 2s\log(ed/s)$, we have $\epsilon < 1$. Moreover, when $s \leq C'd$ for sufficiently small constant $C'$ that depends on $C$, we have $n \geq Cs/\epsilon^2$. Therefore, such value of $\epsilon$ leads to (A.14). Thus we conclude that

$$\mathbb{P}_{\theta}^n \left[ \frac{\mathbf{v}^\top \widehat{\boldsymbol{\Sigma}}_W \mathbf{v} - 2\mathbf{v}^\top \boldsymbol{\Sigma}^{-1}\mathbf{v}}{2\mathbf{v}^\top \boldsymbol{\Sigma}^{-1}\mathbf{v}} \geq \sqrt{\frac{s\log(ed/s)}{n}} \cdot \frac{\|\boldsymbol{\Sigma}^{-1}\|_2}{\mathbf{v}^\top \boldsymbol{\Sigma}^{-1}\mathbf{v}}, \text{for all } \mathbf{v} \in \mathcal{B}_2(s) \right] \leq 2d^{-1}$$

Note that $\|\boldsymbol{\Sigma}^{-1}\|_2/(\mathbf{v}^\top\boldsymbol{\Sigma}^{-1}\mathbf{v}) \leq \|\boldsymbol{\Sigma}^{-1}\|_2\|\boldsymbol{\Sigma}\|_2 = \kappa$. Our choice of $\tau_1$ ensures the type-I error of $\phi_1$ does not exceed $2d^{-1}$.

Now we turn to analyze the performance of $\phi_2$. Recall that $\phi_1$ simply selects the coordinate of $\bar{\mathbf{u}} := 1/n \cdot \sum_{i=1}^n \mathbf{u}_i$ that has the largest magnitude (scaled with $\text{diag}(\boldsymbol{\Sigma})^{-1/2}$) and compare it with $\tau_2$. It suffices to show all coordinates are well bounded around $0$ under null hypothesis. Denote the $j$-th coordinate of $\bar{\mathbf{u}}$ by $\bar{u}_j$. Denote the $j$-th diagonal term of $\boldsymbol{\Sigma}$ by $\sigma_j$. We have $\bar{u}_j \sim \mathcal{N}(0, 2\sigma_j/n)$. Recall that for standard normal random variable $X$, we have

$$\mathbb{P}(|X| \geq t) \leq 2\exp(-t^2/2) \text{ for any } t \geq 1. \tag{A.15}$$

Using this property and taking union bound over $j \in [d]$, we have

$$\mathbb{P}_{\boldsymbol{\theta}}^n\left(\sup_{j\in[d]} |\bar{u}_j|/\sqrt{\sigma_j} \geq 8\log d/n\right) \leq 2d \cdot \exp(-2\log d) = 2d^{-1}.$$

Accordingly, our choice of $\tau_2$ can ensure type-I error of $\phi_2$ is controlled within $2d^{-1}$.

**Type-II error.** Under the alternative hypothesis $\boldsymbol{\theta} \in \mathcal{G}_1(\boldsymbol{\Sigma};\gamma_n)$. Note that $\phi = 0$ if and only if $\phi_1 = 0$ and $\phi_2 = 0$. Thus, for any $\boldsymbol{\theta} \in \mathcal{G}_1(\boldsymbol{\Sigma};\gamma_n)$, we have

$$\mathbb{P}_{\boldsymbol{\theta}}^n(\phi = 0) = \mathbb{P}_{\boldsymbol{\theta}}^n(\phi_1 = 0 \cap \phi_2 = 0) \leq \mathbb{P}_{\boldsymbol{\theta}}^n(\phi_1 = 0) \wedge \mathbb{P}_{\boldsymbol{\theta}}^n(\phi_2 = 0). \tag{A.16}$$

We assume $\gamma_n \geq C\kappa[\sqrt{s\log d/n} \vee (1/\alpha^2 \cdot s\log d/n)]$. It suffices to bound the type-II error by considering these two cases: (i) when $\gamma_n \gtrsim \kappa\sqrt{s\log d/n}$, we show that $\mathbb{P}_{\boldsymbol{\theta}}^n(\phi_1 = 0) \leq 16d^{-1}$; (ii) when $\gamma_n \gtrsim \kappa/\alpha^2 \cdot s\log d/n$ and $16/\alpha^2 \cdot s\log d/n \leq \sqrt{s\log d/n}$, we show $\mathbb{P}_{\boldsymbol{\theta}}^n[\phi_2 = 0] \leq 7d^{-1}$.

**Case (i).** Now we consider the first case. We denote $\Delta\boldsymbol{\mu} := \boldsymbol{\mu}_1 - \boldsymbol{\mu}_0$. Let $\mathbf{v}^* := \Delta\boldsymbol{\mu}/\|\Delta\boldsymbol{\mu}\|_2$. Since $\mathbf{v}^* \in \mathcal{B}_2(s)$, we have

$$\sup_{\mathbf{v}\in\mathcal{B}_2(s)} \frac{\mathbf{v}^\top\widehat{\boldsymbol{\Sigma}}_W\mathbf{v}}{2\mathbf{v}^\top\boldsymbol{\Sigma}^{-1}\mathbf{v}} \geq \frac{\mathbf{v}^{*\top}\widehat{\boldsymbol{\Sigma}}_W\mathbf{v}^*}{2\mathbf{v}^{*\top}\boldsymbol{\Sigma}^{-1}\mathbf{v}^*}.$$

It remains to show the right hand side is larger than $1 + \tau_1$ with high probability. Note that

$$\mathbf{v}^{*\top}\widehat{\boldsymbol{\Sigma}}_W\mathbf{v}^* = \frac{1}{n}\sum_{i=1}^n (\mathbf{v}^{*\top}\boldsymbol{\Sigma}^{-1}\mathbf{w}_i)^2.$$

We define a random variable $\widetilde{W} := \mathbf{v}^{*\top}\boldsymbol{\Sigma}^{-1}\boldsymbol{W}$, whose probability distribution is given by

$$1/2 \cdot \mathcal{N}(0,\nu) + 1/4 \cdot \mathcal{N}(m,\nu) + 1/4 \cdot \mathcal{N}(-m,\nu), \tag{A.17}$$

where we define $m := \rho(\boldsymbol{\theta})/\|\Delta\boldsymbol{\mu}\|_2$ and $\nu := 2\rho(\boldsymbol{\theta})/\|\Delta\boldsymbol{\mu}\|_2^2$. Recall that $\rho(\boldsymbol{\theta}) := \Delta\boldsymbol{\mu}^\top\boldsymbol{\Sigma}^{-1}\Delta\boldsymbol{\mu}$. Let $\widetilde{w}_i := \mathbf{v}^{*\top}\boldsymbol{\Sigma}^{-1}\mathbf{w}_i$. Due to the mixture structure (A.17), we can thus cluster $\{\widetilde{w}_i\}_{i=1}^n$ into three groups $\{\widetilde{w}_i^{(k)}\}_{i=1}^{n_k}, k \in \{1,2,3\}$, based on the latent labels. The $k$-th group corresponds to the $k$-th term in (A.17). Note that $\mathbb{E}(n_1) = n/2, \mathbb{E}(n_2) = \mathbb{E}(n_3) = n/4$. Define event $\mathcal{E}_1$ as

$$\mathcal{E}_1 := \{|n_1 - n/2| \leq 1/8 \cdot n, |n_2 - n/4| \leq 1/8 \cdot n, |n_3 - n/4| \leq 1/8 \cdot n\}. \tag{A.18}$$

By Hoeffding's inequality, we have $\mathbb{P}(\mathcal{E}_1) \geq 1 - 6\exp(-n^2/32)$.

From now on, we condition on event $\mathcal{E}_1$. By the standard $\chi^2$-tail bound (Lemma C.1), for any $t \in (0,1)$ and $k \in \{1,2,3\}$, we have

$$\mathbb{P}_{\boldsymbol{\theta}}^n\left(\left|\sum_{i=1}^{n_k}(\widetilde{w}_i^{(k)} - m_k)^2 - n_k\nu\right| \geq n_k\nu t\right) \leq 2e^{-n_k t^2/8} \leq 2e^{-nt^2/64}, \tag{A.19}$$

where $m_1 = 0, m_2 = -m_3 = m$. Moreover, using tail bound of Gaussian (A.15), for $t' \geq 1/\sqrt{n_k}$ and $k = 2,3$,

$$\mathbb{P}_{\boldsymbol{\theta}}^n\left(\left|\sum_{i=1}^{n_k}\widetilde{w}_i^{(k)} - n_k m_k\right| \geq n_k\sqrt{\nu}t'\right) \leq 2e^{-n_k t'^2/2} \leq 2e^{-nt'^2/16}. \tag{A.20}$$

Excluding the small chance events in (A.19) and (A.20), we find that

$$\sum_{i=1}^{n} \widetilde{w}_i^2 = \sum_{k=1}^{3}\sum_{i=1}^{n_k}(\widetilde{w}_i^{(k)} - m_k)^2 + 2\sum_{k=2}^{3}\sum_{i=1}^{n_k} m_k \widetilde{w}_i^{(k)} - (n_2 + n_3)m^2$$

$$\geq n\nu(1 - t) + 2\sum_{k=2}^{3}\sum_{i=1}^{n_k} m_k \widetilde{w}_i^{(k)} - (n_2 + n_3)m^2$$

$$\geq n\nu(1 - t) + (n_2 + n_3)m^2 - 2(n_2 + n_3)\sqrt{\nu}t'm$$

$$\geq n\nu(1 - t) + 1/4 \cdot nm^2 - 3/2 \cdot n\sqrt{\nu}t'm,$$

where the last step follows from (A.18). Note that $2\mathbf{v}^{*\top}\boldsymbol{\Sigma}^{-1}\mathbf{v}^* = \nu$. We thus have

$$\frac{\mathbf{v}^{*\top}\widehat{\boldsymbol{\Sigma}}_W \mathbf{v}^*}{2\mathbf{v}^{*\top}\boldsymbol{\Sigma}^{-1}\mathbf{v}^*} - 1 = \frac{\sum_{i=1}^n \widetilde{w}_i^2}{2n\mathbf{v}^{*\top}\boldsymbol{\Sigma}^{-1}\mathbf{v}^*} - 1 \geq \frac{m^2}{4\nu} - t - \frac{3mt'}{2\sqrt{\nu}}$$

$$= 1/8 \cdot \rho(\boldsymbol{\theta}) - t - 3t'/4 \cdot \sqrt{2\rho(\boldsymbol{\theta})}. \tag{A.21}$$

Now we choose $t = t' = 8\sqrt{s\log(ed/s)/n}$, which is less than one under condition $n \geq 64s\log(ed/s)$. When $\rho(\boldsymbol{\theta}) \geq C\kappa\sqrt{s\log(ed/s)/n}$ for sufficiently large constant $C$, we can have $t \leq \rho(\boldsymbol{\theta})/32$ and $t' \leq \sqrt{t'} \leq \sqrt{\rho(\boldsymbol{\theta})}/48$. Accordingly, proceeding with (A.21) gives

$$1/2 \cdot \mathbf{v}^{*\top}\widehat{\boldsymbol{\Sigma}}_W \mathbf{v}^*/\mathbf{v}^{*\top}\boldsymbol{\Sigma}^{-1}\mathbf{v}^* - 1 \geq 1/16 \cdot \rho(\boldsymbol{\theta}) \geq \tau_1.$$

Plugging the value of $t, t'$ into the tail bounds in (A.19) (A.20) and using the probability of event $\mathcal{E}_1$, we have the type-II error of $\phi_1$ is most $10d^{-1} + 6e^{-n^2/32} \leq 16d^{-1}$.

**Case (ii).** Now we turn to analyze the performance of $\phi_2$. We introduce shorthands $\widetilde{\boldsymbol{\mu}} := \text{diag}(\boldsymbol{\Sigma})^{-1/2}\Delta\boldsymbol{\mu}$ and $\boldsymbol{\Lambda} := \text{diag}(\boldsymbol{\Sigma})^{1/2}$. Then it holds that

$$\rho(\boldsymbol{\theta}) = \Delta\boldsymbol{\mu}^\top\boldsymbol{\Sigma}^{-1}\Delta\boldsymbol{\mu} = \Delta\boldsymbol{\mu}^\top\boldsymbol{\Lambda}^{-1}\boldsymbol{\Lambda}\boldsymbol{\Sigma}^{-1}\boldsymbol{\Lambda}\boldsymbol{\Lambda}^{-1}\Delta\boldsymbol{\mu} \leq \|\widetilde{\boldsymbol{\mu}}\|_2^2\|\boldsymbol{\Lambda}\boldsymbol{\Sigma}^{-1}\boldsymbol{\Lambda}\|_{op}$$

$$\leq \|\widetilde{\boldsymbol{\mu}}\|_2^2\|\boldsymbol{\Lambda}\|_2^2\|\boldsymbol{\Sigma}^{-1}\|_2 \leq \kappa\|\widetilde{\boldsymbol{\mu}}\|_2^2,$$

where the last step follows from the fact that $\|\text{diag}(\boldsymbol{\Sigma})\|_2 \leq \|\boldsymbol{\Sigma}\|_2$. Suppose the $j$-th coordinate of $\widetilde{\boldsymbol{\mu}}$, denoted by $\beta$, has largest magnitude. Since $\|\widetilde{\mathbf{u}}\|_2^2 \leq s\beta^2$, we have $\beta^2 \geq \rho(\boldsymbol{\theta})/(s\kappa)$. Under condition

$$\rho(\boldsymbol{\theta}) \geq \gamma_n \geq \frac{400\kappa s\log d}{\alpha^2 n},$$

we have

$$\beta \geq 20\sqrt{\log d/(\alpha^2 n)}. \tag{A.22}$$

Let $\mathbf{v}^* = \text{sign}(\beta) \cdot \mathbf{e}_j$. We have

$$\sup_{\mathbf{v}\in\mathcal{B}_2(1)} \langle \mathbf{v}, \boldsymbol{\Lambda}^{-1}\bar{\mathbf{u}}\rangle \geq \langle \mathbf{v}^*, \boldsymbol{\Lambda}^{-1}\bar{\mathbf{u}}\rangle = \left|\frac{1}{n}\sum_{i=1}^n \widetilde{u}_{ij}\right|,$$

where we denote the $j$-th coordinate of $\boldsymbol{\Lambda}^{-1}\mathbf{u}_i$ by $\widetilde{u}_{ij}$.

Let $U_j$ be the $j$-th coordinate of $\boldsymbol{U}$. Note that $\{\widetilde{u}_{ij}\}_{i=1}^n$ are i.i.d. samples of $U_j/\sqrt{\sigma_j}$. Recall that $\sigma_j$ is the $j$-th diagonal term of $\boldsymbol{\Sigma}$. According to (A.13), $U_j/\sqrt{\sigma_j}$ has the mixture distribution

$$(1+\alpha)^2/4 \cdot \mathcal{N}(\beta, 2) + (1-\alpha^2)/2 \cdot \mathcal{N}(0, 2) + (1-\alpha)^2/4 \cdot \mathcal{N}(-\beta, 2). \tag{A.23}$$

We can cluster these samples into three groups $\{\widetilde{u}_{ij}^{(k)}\}_{i=1}^{n_k}, k \in \{1, 2, 3\}$ based on latent labels, where $k$-th group corresponds to the $k$-th term in (A.23). Using tail bound of Gaussian (A.15), we have for $t \geq 1$ and $k \in \{1, 2, 3\}$,

$$\mathbb{P}_{\boldsymbol{\theta}}^n\left(\left|\sum_{i=1}^{n_k} \widetilde{u}_{ij}^{(k)} - n_k m_k\right| \geq \sqrt{2n_k}t\right) \leq 2e^{-t^2/2},$$

where $m_1 = -m_3 = \beta, m_2 = 0$. Therefore, with probability at least $1 - 6e^{-t^2/2}$, it holds that

$$\left|\frac{1}{n}\sum_{i=1}^n \widetilde{u}_{ij} - \frac{(n_1 - n_3)\beta}{n}\right| \leq t \cdot \sum_{k=1}^{3}\sqrt{\frac{2n_k}{n^2}} \leq \frac{5t}{\sqrt{n}}. \tag{A.24}$$

It remains to bound $n_1 - n_3$. Note that $n_1 - n_3$ is a summation of $n$ i.i.d. random variables $V_i$ satisfying $\mathbb{P}(V_i = 1) = (1+\alpha)^2/4$, $\mathbb{P}(V_i = 0) = (1-\alpha^2)/2$, and $\mathbb{P}(V_i = -1) = (1-\alpha)^2/4$. Then $V_i$ has mean $\alpha$, variance $(1-\alpha^2)/2 \leq 1 - \alpha$, and $|V_i - \mathbb{E}(V_i)| \leq 2$. By Bernstein's inequality, we have that for $t' > 0$,

$$\mathbb{P}\left(|n_1 - n_3 - \alpha n| \geq t'\right) \leq \exp\left[-\frac{t'^2}{2(1-\alpha)n + 4t'/3}\right].$$

Choosing $t' = \alpha n/2$, we thus have

$$\mathbb{P}\left(|n_1 - n_3 - \alpha \cdot n| \geq \alpha n/2\right) \leq \exp\left[-\frac{\alpha^2 n}{8(1-\alpha) + 8\alpha/3}\right] \leq \exp(-\alpha^2 n/8) \leq d^{-1}, \quad \text{(A.25)}$$

where the last step follows from condition $8s \log d/(\alpha^2 n) \leq \sqrt{s \log(ed/s)/n} \leq 1$. Combining (A.24) and (A.25), we have that with high probability $1 - 6e^{-t^2/2} - d^{-1}$,

$$\left|1/n \cdot \sum_{i=1}^{n} \widetilde{u}_{ij}\right| \geq \alpha\beta/2 - 5t/\sqrt{n} \geq 10\sqrt{\log d/n} - 5t/\sqrt{n} \geq \tau_2,$$

where the second step follows from (A.22) and the last inequality holds by setting $t = \sqrt{2 \log d}$, which gives the type-II error of $\phi_2$ is at most $7d^{-1}$.

Using (A.16) and the conclusions in the above two cases, we thus show Type-II error of $\phi$ is at most $16d^{-1}$ and thus complete the proof.

## A.3   Proof of Theorem 3.3

In this section, we prove the computational lower bound. We first show that the information-theoretic lower bound in (3.4) is a lower bound of the computationally tractable minimax rate. To see this, we consider the oracle $r^*$ that returns sample average $n^{-1}\sum_{i=1}^{n} q(y_i, \mathbf{x}_i)$ for any query function $q$. As discussed in §2.2, Bernstein's inequality in (2.6) and uniform concentration of empirical process imply that $r^* \in \mathcal{R}[\xi, n, T_n, \eta(\mathcal{Q}_{\mathscr{A}})]$. In addition, every test function $\phi$ that is based on the responses of $r^*$ is also a function of $\{(y_i, \mathbf{x}_i)\}_{i=1}^{n}$. Thus combining (2.4) and (2.7), it holds that

$$\overline{R}_n^*(\mathcal{G}_0, \mathcal{G}_1; \mathscr{A}, r^*) \geq R_n^*(\mathcal{G}_0, \mathcal{G}_1).$$

Therefore, by Theorem 3.1, for any $\gamma_n$ satisfying

$$\gamma_n = o\left[\sqrt{s \log d/n} \wedge (1/\alpha^2 \cdot s \log d/n)\right],$$

we have $\lim_{n \to \infty} \overline{R}_n^*[\mathcal{G}_0, \mathcal{G}_1(\gamma_n); \mathscr{A}, r^*] = 1$. Here the equality holds because a test based on purely random guess incurs risk one.

Based on this observation, to show Theorem 3.3, it the following, we assume that

$$\gamma_n = o\left[\sqrt{s^2/n} \wedge (1/\alpha^2 \cdot s/n)\right]. \tag{A.26}$$

We show that under this assumption, there exists an oracle $r$ such that the minimax testing risk is not negligible. Similar to the derivation of the information theoretical lower bound, we also focus on the restricted testing problem defined in (A.1). Following the same notations, we denote by $\mathbb{P}_0$ the distribution of model $(\mathbf{0}, \mathbf{0}, \mathbf{I}, \alpha)$ and by $\mathbb{P}_{\mathbf{v}}$ the distribution of model $(-\mathbf{v}/2, \mathbf{v}/2, \mathbf{I}, \alpha)$ for all $\mathbf{v} \in \mathcal{H}(s) = \{\mathbf{u} \in \{0, \beta\}^d : \|\mathbf{u}\|_0 = s\}$. Here we assume that the SNR under $H_1$ satisfies $\beta^2 s = \gamma_n$. Moreover, we define $\overline{\mathbb{P}}_0$ as the distribution of the random variables returned by the statistical query model under the null hypothesis $H_0$ and define $\overline{\mathbb{P}}_{\mathbf{v}}$ correspondingly. Then the minimax testing risk $\overline{R}_n^*(\mathcal{G}_0, \mathcal{G}_1; \mathscr{A}, r)$ defined in (2.7) is lower bounded by

$$\sup_{\boldsymbol{\Sigma}} \overline{R}_n^*[\mathcal{G}_0(\boldsymbol{\Sigma}), \mathcal{G}_1(\boldsymbol{\Sigma}; \gamma_n); \mathscr{A}, r] \geq \inf_{\phi \in \mathcal{H}(\mathscr{A}, r)} \left[\overline{\mathbb{P}}_0(\phi = 1) + \frac{1}{|\mathcal{H}(s)|} \sum_{\mathbf{v} \in \mathcal{H}(s)} \overline{\mathbb{P}}_{\mathbf{v}}(\phi = 0)\right].$$

The following lemma establishes a sufficient condition that any hypothesis test under the statistical query model is asymptotically powerless. See [27] and [8] for a proof.

**Lemma A.3.** *For any algorithm $\mathscr{A} \in \mathcal{A}(T)$ and any query function $q \in \mathcal{Q}_\mathscr{A}$, we define*

$$\mathcal{C}_1(q) = \left\{\mathbf{v} \in \mathcal{H}(s) : \mathbb{E}_{\mathbb{P}_\mathbf{v}}\left[q(Y, \boldsymbol{X})\right] - \mathbb{E}_{\mathbb{P}_\mathbf{0}}\left[q(Y, \boldsymbol{X})\right] > \tau_q(\mathbb{P}_\mathbf{v})\right\},$$

$$\mathcal{C}_2(q) = \left\{\mathbf{v} \in \mathcal{H}(s) : \mathbb{E}_{\mathbb{P}_\mathbf{0}}\left[q(Y, \boldsymbol{X})\right] - \mathbb{E}_{\mathbb{P}_\mathbf{v}}\left[q(Y, \boldsymbol{X})\right] > \tau_q(\mathbb{P}_\mathbf{v})\right\}.$$

*Here $\tau_q(\mathbb{P}_\mathbf{v})$ is the tolerance parameter defined in* (2.5) *when $(Y, \boldsymbol{X}) \sim \mathbb{P}_\mathbf{v}$. Then if $T \cdot \sup_{q \in \mathcal{Q}_\mathscr{A}}\left(|\mathcal{C}_1(q)| + |\mathcal{C}_2(q)|\right)/|\mathcal{H}(s)| = o(1)$, there exists an oracle $r \in \mathcal{R}[\xi, n, T, \eta(\mathcal{Q}_\mathscr{A})]$ such that*

$$\inf_{\phi \in \mathcal{H}(\mathscr{A}, r)}\left[\overline{\mathbb{P}}_\mathbf{0}(\phi = 1) + \frac{1}{|\mathcal{H}(s)|}\sum_{\mathbf{v} \in \mathcal{H}(s)}\overline{\mathbb{P}}_\mathbf{v}(\phi = 0)\right] = 1.$$

By this lemma, we need to construct an upper bound for $\sup_{q \in \mathcal{Q}_\mathscr{A}}\left(|\mathcal{C}_1(q)| + |\mathcal{C}_2(q)|\right)$. In the sequel, we achieve this goal by studying the uniform mixture of $\{\mathbb{P}_\mathbf{v} : \mathbf{v} \in \mathcal{C}_\ell(q)\}$ for $\ell \in \{1, 2\}$. Specifically, we define

$$\mathbb{P}_{\mathcal{C}_1(q)} = \frac{1}{|\mathcal{C}_1(q)|}\sum_{\mathbf{v} \in \mathcal{C}_1(q)}\mathbb{P}_\mathbf{v} \text{ and } \mathbb{P}_{\mathcal{C}_2(q)} = \frac{1}{|\mathcal{C}_2(q)|}\sum_{\mathbf{v} \in \mathcal{C}_2(q)}\mathbb{P}_\mathbf{v}. \tag{A.27}$$

The following lemma, obtained from [8], establishes an upper bound for the $\chi^2$-divergence between $\mathbb{P}_{\mathcal{C}_\ell(q)}$ and $\mathbb{P}_\mathbf{0}$.

**Lemma A.4.** *For $\ell \in \{1, 2\}$ we define*

$$\overline{\mathcal{C}}_\ell(q, \mathbf{v}) = \operatorname*{argmax}_\mathcal{C}\left\{\frac{1}{|\mathcal{C}|}\sum_{\mathbf{v}' \in \mathcal{C} \subseteq \mathcal{H}(s)}\mathbb{E}_{\mathbb{P}_\mathbf{0}}\left[\frac{d\mathbb{P}_\mathbf{v}}{d\mathbb{P}_\mathbf{0}}\frac{d\mathbb{P}_{\mathbf{v}'}}{d\mathbb{P}_\mathbf{0}}(Y, \boldsymbol{X})\right] - 1 \,\middle|\, |\mathcal{C}| = |\mathcal{C}_\ell(q)|\right\}. \tag{A.28}$$

*Then the $\chi^2$-divergence between $\mathbb{P}_{\mathcal{C}_\ell(q)}$ and $\mathbb{P}_\mathbf{0}$ is bounded by*

$$D_{\chi^2}(\mathbb{P}_{\mathcal{C}_\ell(q)}, \mathbb{P}_\mathbf{0}) \leq \sup_{\mathbf{v} \in \mathcal{C}_\ell(q)}\frac{1}{|\mathcal{C}_\ell(q)|}\sum_{\mathbf{v}' \in \overline{\mathcal{C}}_\ell(q, \mathbf{v})}\mathbb{E}_{\mathbb{P}_\mathbf{0}}\left[\frac{d\mathbb{P}_\mathbf{v}}{d\mathbb{P}_\mathbf{0}}\frac{d\mathbb{P}_{\mathbf{v}'}}{d\mathbb{P}_\mathbf{0}}(Y, \boldsymbol{X})\right] - 1. \tag{A.29}$$

Notice that Lemma A.1 enables us to compute the right-hand side of (A.29) in closed form. For any $\alpha \in [0, 1]$, function $h_\alpha(t) = \cosh[\beta^2/2 \cdot (s - t)] + \alpha^2 \sinh[\beta^2/2 \cdot (s - t)]$ is monotone nonincreasing for $t \in \{0, \ldots, s\}$ and $f(s) = 0$. In addition, for any $\mathbf{v} \in \mathcal{H}(s)$ and any $j \in \{0, \ldots, s\}$, we define

$$\mathcal{C}_j(\mathbf{v}) = \left\{\mathbf{v}' \in \mathcal{H}(s) : |\operatorname{supp}(\mathbf{v}) \cap \operatorname{supp}(\mathbf{v}')| = s - j\right\}. \tag{A.30}$$

For $\ell \in \{1, 2\}$, any query function $q \in \mathcal{Q}_\mathscr{A}$, and any $\mathbf{v} \in \mathcal{C}_\ell(q)$, by Lemma A.1 and the definition of $\overline{\mathcal{C}}_\ell(q, \mathbf{v})$ in (A.28), there exists an integer $k_\ell(q, \mathbf{v})$ that satisfies

$$\overline{\mathcal{C}}_\ell(q, \mathbf{v}) = \mathcal{C}_0(\mathbf{v}) \cup \mathcal{C}_1(\mathbf{v}) \cup \cdots \cup \mathcal{C}_{k_\ell(q,\mathbf{v})-1}(\mathbf{v}) \cup \mathcal{C}'_\ell(q, \mathbf{v}), \tag{A.31}$$

where $\mathcal{C}'_\ell(q, \mathbf{v}) = \overline{\mathcal{C}}_\ell(q, \mathbf{v}) \setminus \bigcup_{j=0}^{k_\ell(q,\mathbf{v})-1}\mathcal{C}_j(\mathbf{v})$ has cardinality

$$|\mathcal{C}'_\ell(q, \mathbf{v})| = |\mathcal{C}_\ell(q)| - \sum_{j=0}^{k_\ell(q,\mathbf{v})-1}|\mathcal{C}_j(\mathbf{v})| < |\mathcal{C}_{k_\ell(q,\mathbf{v})}(\mathbf{v})|. \tag{A.32}$$

Thus we can sandwich the cardinality of $\overline{\mathcal{C}}_\ell(q, \mathbf{v})$ by

$$\sum_{j=0}^{k_\ell(q,\mathbf{v})}|\mathcal{C}_j(\mathbf{v})| > |\overline{\mathcal{C}}_\ell(q, \mathbf{v})| \geq \sum_{j=0}^{k_\ell(q,\mathbf{v})-1}|\mathcal{C}_j(\mathbf{v})|. \tag{A.33}$$

Combining Lemmas A.1 and A.4, we further have

$$1 + D_{\chi^2}(\mathbb{P}_{\mathcal{C}_\ell(q)}, \mathbb{P}_\mathbf{0}) \leq \frac{\sum_{i=0}^{k_\ell(q,\mathbf{v})-1}h_\alpha(j) \cdot |\mathcal{C}_j(\mathbf{v})| + h_\alpha[k_\ell(q, \mathbf{v})] \cdot |\mathcal{C}'_\ell(q, \mathbf{v})|}{\sum_{j=0}^{k_\ell(q,\mathbf{v})-1}|\mathcal{C}_j(\mathbf{v})| + |\mathcal{C}'_\ell(q, \mathbf{v})|}, \text{ for all } \mathbf{v} \in \mathcal{C}_\ell(q). \tag{A.34}$$

Moreover, by (A.34) and the monotonicity of $h_\alpha(t)$ we obtain

$$1 + D_{\chi^2}(\mathbb{P}_{\mathcal{C}_\ell(q)}, \mathbb{P}_\mathbf{0}) \leq \frac{\sum_{i=0}^{k_\ell(q,\mathbf{v})-1}h_\alpha(j) \cdot |\mathcal{C}_j(\mathbf{v})|}{\sum_{j=0}^{k_\ell(q,\mathbf{v})-1}|\mathcal{C}_j(\mathbf{v})|}. \tag{A.35}$$

By the definition of $\mathcal{C}_j(\mathbf{v})$ in (A.30), the cardinality of $\mathcal{C}_j(\mathbf{v})$ does not depend on the choice of $\mathbf{v} \in \mathcal{H}(s)$ and we have $|\mathcal{C}_j(\mathbf{v})| = \binom{s}{s-j}\binom{d-s}{j}$. Thus for any $j \in \{0, \ldots, s-1\}$ we have

$$|\mathcal{C}_{j+1}(\mathbf{v})|/|\mathcal{C}_j(\mathbf{v})| = (s-j) \cdot (d-s-j)/(j+1)^2 \geq (d-2s)/s^2. \tag{A.36}$$

Under the assumption that $s^2/d = o(1)$, the right-hand side of (A.36) is lower bounded by $\zeta = d/(2s^2)$ when $d$ and $s$ are sufficiently large. Then we have $|\mathcal{C}_j(\mathbf{v})| \leq \zeta^{j-s}|\mathcal{C}_s(\mathbf{v})|$ for $j \in \{0, \ldots, s\}$. By the definition of $k_\ell(q, \mathbf{v})$ in (A.31) and (A.32), for any $q \in \mathcal{Q}_{\mathscr{A}}$, we further obtain

$$|\mathcal{C}_\ell(q)| \leq \sum_{j=0}^{k_\ell(q,\mathbf{v})} |\mathcal{C}_j(\mathbf{v})| \leq |\mathcal{C}_s(\mathbf{v})| \sum_{j=0}^{k_\ell(q,\mathbf{v})} \zeta^{j-s}$$

$$\leq \frac{\zeta^{-[s-k_\ell(q,\mathbf{v})]}|\mathcal{H}(s)|}{1-\zeta^{-1}} \leq 2\zeta^{-[s-k_\ell(q,\mathbf{v})]}|\mathcal{H}(s)|, \tag{A.37}$$

where the last inequality follows from the fact that $\zeta^{-1} = 2s^2/d = o(1)$.

Moreover, for any two positive sequences $\{a_i\}_{i=0}^s$ and $\{b_i\}_{i=0}^s$ satisfying $a_i/a_{i-1} \geq b_i/b_{i-1} > 1$ for all $i \in [s]$, since $h_\alpha(t)$ is nonincreasing, for any $k \in [s]$, we have

$$\sum_{0 \leq i < j \leq k} (a_i b_j - a_j b_i) \cdot [h_\alpha(i) - h_\alpha(j)] \leq 0. \tag{A.38}$$

Further simplifying the terms in (A.38), we have

$$\frac{\sum_{i=0}^k [a_i h_\alpha(i)]}{\sum_{i=0}^k a_i} \leq \frac{\sum_{i=0}^k [b_i h_\alpha(i)]}{\sum_{i=0}^k b_i}. \tag{A.39}$$

In what follows, we upper bound $k_\ell(q, \mathbf{v})$ for $\ell \in \{1, 2\}$ and $\mathbf{v} \in \mathcal{C}_\ell(q)$. We employ the shorthand $k_\ell = k_\ell(q, \mathbf{v})$ to simplify the notations. Combining (A.29), (A.35), and (A.39) with $a_j = |\mathcal{C}_j(\mathbf{v})|$ and $b_j = \zeta^j$, we have

$$1 + D_{\chi^2}(\mathbb{P}_{\mathcal{C}_\ell(q)}, \mathbb{P}_0) \leq \frac{\sum_{j=0}^{k_\ell-1} \zeta^j h_\alpha(j)}{\sum_{j=0}^{k_\ell-1} \zeta^j}$$

$$= \frac{\sum_{j=0}^{k_\ell-1} \zeta^j \left\{ \cosh\left[\beta^2/2 \cdot (s-j)\right] + \alpha^2 \sinh\left[\beta^2/2 \cdot (s-j)\right] \right\}}{\sum_{j=0}^{k_\ell-1} \zeta^j}$$

$$\leq \left\{ \frac{\sum_{j=0}^{k_\ell-1} \zeta^j \cosh\left[\beta^2(s-j)\right]}{\sum_{j=0}^{k_\ell-1} \zeta^j} \right\} \bigvee \left\{ \frac{\sum_{j=0}^{k_\ell-1} \zeta^j \exp\left[\alpha^2\beta^2(s-j)\right]}{\sum_{j=0}^{k_\ell-1} \zeta^j} \right\}. \tag{A.40}$$

Here the second inequality follows from Lemma A.2. We bound the two terms in (A.40) separately. Note that for notational simplicity, we denote for any $t \in \{0, \ldots, s\}$, we define

$$f(t) = \cosh\left[\beta^2(s-t)\right], \quad g(t) = \exp\left[\alpha^2\beta^2(s-t)\right].$$

Note that both $h(t)$ and $g(t)$ are monotone non-increasing, and thus $f(t) \geq f(s) = 1$ and $g(t) \geq g(s) = 1$. Moreover, by calculation, we have $f(j-1)/f(j) \geq \cosh(\beta^2)$ for all $j \in \{1, \ldots, s\}$. Thus we have

$$f(j) \leq f(k_\ell - 1) \cdot \left[\cosh(\beta^2)\right]^{k_\ell - j - 1}, \quad \text{for all } j \in \{0, \ldots, k_\ell - 1\}.$$

Then we have

$$\frac{\sum_{j=0}^{k_\ell-1} \zeta^j f(j)}{\sum_{j=0}^{k_\ell-1} \zeta^j} \leq f(k_\ell - 1) \cdot \frac{\sum_{j=0}^{k_\ell-1} \zeta^j \left[\cosh(\beta^2)\right]^{k_\ell - j + 1}}{\sum_{j=0}^{k_\ell-1} \zeta^j}$$

$$\leq f(k_\ell - 1) \cdot \frac{\sum_{j=0}^{k_\ell-1} \left[\cosh(\beta^2)/\zeta\right]^{k_\ell - j + 1}}{\sum_{j=0}^{k_\ell-1} \zeta^{-(k_\ell - j + 1)}}$$

$$= f(k_\ell - 1) \cdot \frac{1 - \left[\cosh(\beta^2)/\zeta\right]^{k_\ell}}{1 - \zeta^{-k_\ell}} \cdot \frac{1 - \zeta^{-1}}{1 - \zeta^{-1}\cosh(\beta^2)}. \tag{A.41}$$

Since $\cosh(\beta^2) > 1$, by (A.41) we have

$$\frac{\sum_{j=0}^{k_\ell-1} \zeta^j \cosh\left[\beta^2(s-j)\right]}{\sum_{j=0}^{k_\ell-1} \zeta^j} \leq \frac{1-\zeta^{-1}}{1-\zeta^{-1}\cosh(\beta^2)} \cdot \cosh\left[\beta^2(s-k_\ell+1)\right]. \tag{A.42}$$

In addition, since $g(j-1)/g(j) = \exp(\alpha^2\beta^2)$, similar to (A.41) we have

$$\frac{\sum_{j=0}^{k_\ell-1} \zeta^j \exp\left[\alpha^2\beta^2(s-j)\right]}{\sum_{j=0}^{k_\ell-1} \zeta^j} \leq \frac{1-\zeta^{-1}}{1-\zeta^{-1}\exp(\alpha^2\beta^2)} \cdot \exp\left[\alpha^2\beta^2(s-k_\ell+1)\right] \tag{A.43}$$

Combining (A.42) and (A.43), we obtain that

$$1 + D_{\chi^2}(\mathbb{P}_{\mathcal{C}_\ell(q)}, \mathbb{P}_{\mathbf{0}})$$
$$\leq \left\{\frac{(1-\zeta^{-1})\cosh\left[\beta^2(s-k_\ell+1)\right]}{1-\zeta^{-1}\cosh(\beta^2)}\right\} \bigvee \left\{\frac{(1-\zeta^{-1})\exp\left[\alpha^2\beta^2(s-k_\ell+1)\right]}{1-\zeta^{-1}\exp(\alpha^2\beta^2)}\right\}. \tag{A.44}$$

Moreover, we use the following lemma obtained from [27] to establish a lower bound for $D_{\chi^2}(\mathbb{P}_{\mathcal{C}_\ell(q)}, \mathbb{P}_{\mathbf{0}})$.

**Lemma A.5.** *For any query function $q$ and $\ell \in \{1, 2\}$, we have*
$$D_{\chi^2}(\mathbb{P}_{\mathcal{C}_\ell(q)}, \mathbb{P}_{\mathbf{0}}) \geq \log(T/\xi)/n.$$

We denote $\sqrt{\log(T/\xi)/n}$ by $\tau$ for simplicity of notations. Combining (A.44), Lemma A.5 and inequality $\cosh(x) \leq \exp(x^2/2)$, at least one of the two inequality holds

$$(1+\tau^2) \cdot \left[1 - \zeta^{-1}\cosh(\beta^2/2)\right]/(1-\zeta^{-1}) \leq \exp\left[\beta^4/2 \cdot (s-k_\ell+1)^2\right], \tag{A.45}$$
$$(1+\tau^2) \cdot \left[1 - \zeta^{-1}\exp(\alpha^2\beta^2)\right]/(1-\zeta^{-1}) \leq \exp\left[\alpha^2\beta^2(s-k_\ell+1)\right]. \tag{A.46}$$

If (A.45) holds, taking the logarithm of the both sides, we have

$$\beta^4/2 \cdot (s-k_\ell+1)^2 \geq \log(1+\tau^2) - \log\left[\frac{1-\zeta^{-1}}{1-\zeta^{-1}\cosh(\beta^2)}\right]. \tag{A.47}$$

Whereas if (A.46) is true, it holds that

$$\alpha^2\beta^2(s-k_\ell+1) \geq \log(1+\tau^2) - \log\left[\frac{1-\zeta^{-1}}{1-\zeta^{-1}\exp(\alpha^2\beta^2)}\right]. \tag{A.48}$$

In addition, by Taylor expansion and the fact that $[\cosh(\beta^2/2) + \exp(\alpha^2\beta^2)]/\zeta = o(1)$, we have

$$\log\left[\frac{1-\zeta^{-1}}{1-\zeta^{-1}\cosh(\beta^2)}\right] = \log\left\{1 + \frac{\zeta^{-1}\left[\cosh(\beta^2)-1\right]}{1-\zeta^{-1}\cosh(\beta^2)}\right\} = O(\zeta^{-1}\beta^4), \tag{A.49}$$

$$\log\left[\frac{1-\zeta^{-1}}{1-\zeta^{-1}\exp(\alpha^2\beta^2)}\right] = \log\left\{1 + \frac{\zeta^{-1}\left[\exp(\alpha^2\beta^2)-1\right]}{1-\zeta^{-1}\exp(\alpha^2\beta^2))}\right\} = O(\zeta^{-1}\alpha^2\beta^2). \tag{A.50}$$

Since $\gamma_n = s\beta^2$, by (A.26) we have $(\alpha^2\beta^2) \vee \beta^4 = o(\log d/n)$. Hence, by (A.49) and (A.50), the second terms on the right-hand sides of (A.47) and (A.48) are asymptotically negligible compared with $\log(1+\tau^2)$. Therefore, by (A.47) and (A.48), for $\ell \in \{1, 2\}$, at least one of the following two arguments hold:

$$k_\ell(q, \mathbf{v}) \leq s + 1 - \sqrt{\log(1+\tau^2)/\beta^4}, \quad k_\ell(q, \mathbf{v}) \leq s + 1 - \log(1+\tau^2)/(2\alpha^2\beta^2).$$

Equivalently, we have

$$k_\ell(q, \mathbf{v}) \leq \left[s + 1 - \sqrt{\log(1+\tau^2)/\beta^4}\right] \vee \left[s + 1 - \log(1+\tau^2)/(2\alpha^2\beta^2)\right]. \tag{A.51}$$

Recall that $\tau = \sqrt{\log(T/\xi)/n}$ where $\xi = o(1)$. For any constant $\eta > 0$, we set $T = O(d^\eta)$. By combining Lemmas A.3 and A.4, (A.37), and (A.51), we further obtain

$$T \cdot \frac{\sup_{q\in\mathcal{Q}_{\mathscr{A}}}\left(|\mathcal{C}_1(q)| + |\mathcal{C}_2(q)|\right)}{|\mathcal{H}(s)|} \leq 4T \cdot \exp\left\{-\log\zeta \cdot \left[\sqrt{\log(1+\tau^2)/\beta^4} - 1\right]\right\} \wedge$$
$$4T \cdot \exp\left\{-\log\zeta \cdot \left[\log(1+\tau^2)/(2\alpha^2\beta^2) - 1\right]\right\}. \tag{A.52}$$

Under the assumption of the theorem, there is a sufficiently small constant $\delta > 0$ such that $s^2/d^{1-\delta} = O(1)$. Thus we have $\zeta = d/(2s^2) = \Omega(d^\delta)$. By inequality $\log(1+x) \geq x/2$, it holds that $\log(1+$

$\tau^2) \geq \tau^2/2 = \log(T/\xi)/(2n)$. Under the condition in (A.26) , we have

$$\frac{\log(T/\xi)}{2n\beta^4} \bigvee \frac{\log(T/\xi)}{4n\alpha^2\beta^2} > \frac{\log(1/\xi)}{2n\beta^4} \bigvee \frac{\log(1/\xi)}{4\alpha^2\beta^2} \to \infty. \tag{A.53}$$

Hence if $n$ is sufficiently large, the left-hand side in (A.53) is greater than an absolute constant $C$ satisfiying $\delta(C-1) > \eta$. Then by (A.52) we have

$$T \cdot \frac{\sup_{q \in \mathcal{Q}_{\mathscr{A}}} (|\mathcal{C}_1(q)| + |\mathcal{C}_2(q)|)}{|\mathcal{H}(s)|} = O[4d^\eta \zeta^{-(C-1)}] = O[4d^\eta d^{-\delta(C-1)}] = o(1). \tag{A.54}$$

Combining (A.54) and Lemma A.3, we conclude that $\overline{R}_n^*(\mathcal{G}_0, \mathcal{G}_1; \mathscr{A}, r) \to 1$ if (A.26) holds. This concludes the proof of Theorem 3.3.

## A.4 Proof of Theorem 3.4

To ease notation, we denote the joint distribution of $(Y, \boldsymbol{X})$ by $\mathbb{P}_{\boldsymbol{\theta}}$ where the model parameter is given by $\boldsymbol{\theta} = (\boldsymbol{\mu}_0, \boldsymbol{\mu}_1, \boldsymbol{\Sigma}, \alpha)$. In addition, we let $\Delta\boldsymbol{\mu} = \boldsymbol{\mu}_1 - \boldsymbol{\mu}_0$. Thus $\Delta\boldsymbol{\mu} = \mathbf{0}$ for all $\boldsymbol{\theta} \in \mathcal{G}_0(\boldsymbol{\Sigma})$ and $\Delta\boldsymbol{\mu} \in \mathcal{B}(s)$ for all $\boldsymbol{\theta} \in \mathcal{G}_1(\boldsymbol{\Sigma}; \gamma_n)$. In what follows, we bound the type-I and type-II errors of $\overline{\phi}$ respectively.

**Type-I error.** For any $\boldsymbol{\theta} \in \mathcal{G}_0(\boldsymbol{\Sigma})$, by the definition of $\overline{\phi}$, the type-I error is bounded by

$$\overline{\mathbb{P}}_{\boldsymbol{\theta}}(\overline{\phi} = 1) \leq \overline{\mathbb{P}}_{\boldsymbol{\theta}}(\overline{\phi}_1 = 1) + \overline{\mathbb{P}}_{\boldsymbol{\theta}}(\overline{\phi}_2 = 1).$$

For test function $\overline{\phi}_1$, since marginally, $\boldsymbol{X} \sim 1/2 \cdot \mathcal{N}(\boldsymbol{\mu}_0, \boldsymbol{\Sigma}) + 1/2 \cdot \mathcal{N}(\boldsymbol{\mu}_1, \boldsymbol{\Sigma})$, for any $\boldsymbol{\theta} \in \mathcal{G}_0(\boldsymbol{\Sigma}) \cup \mathcal{G}_1(\boldsymbol{\Sigma}; \gamma_n)$, for any $j \in [d]$, we have

$$\mathbb{E}_{\mathbb{P}_{\boldsymbol{\theta}}}(X_j^2/\sigma_j - 1) - \left[\mathbb{E}_{\mathbb{P}_{\boldsymbol{\theta}}}(X_j/\sqrt{\sigma_j})\right]^2$$
$$= 1/4 \cdot (\mu_{0,j} - \mu_{1,j})^2/\sigma_j = 1/4 \cdot (\Delta\boldsymbol{\mu})_j^2/\sigma_j, \tag{A.55}$$

Here $\mu_{0,j}$ and $\mu_{1,j}$ denote the $j$-th entries of $\boldsymbol{\mu}_0$ and $\boldsymbol{\mu}_1$, and $(\Delta\boldsymbol{\mu})_j$ is the $j$-th entry of $\Delta\boldsymbol{\mu}$. In addition, by the definition of $q_j$ in (3.13) we have

$$\left|\left[\mathbb{E}_{\mathbb{P}_{\boldsymbol{\theta}}} q_j(Y, \boldsymbol{X})\right]^2 - \left[\mathbb{E}_{\mathbb{P}_{\boldsymbol{\theta}}}(X_j/\sqrt{\sigma_j})\right]^2\right|$$

$$\leq 2\left|\mathbb{E}_{\mathbb{P}_{\boldsymbol{\theta}}}(X_j/\sqrt{\sigma_j})\right| \cdot \left|\mathbb{E}_{\mathbb{P}_{\boldsymbol{\theta}}}(X_j/\sqrt{\sigma_j}) - \mathbb{E}_{\mathbb{P}_{\boldsymbol{\theta}}} q_j(Y, \boldsymbol{X})\right| + \left|\mathbb{E}_{\mathbb{P}_{\boldsymbol{\theta}}}(X_j/\sqrt{\sigma_j}) - \mathbb{E}_{\mathbb{P}_{\boldsymbol{\theta}}} q_j(Y, \boldsymbol{X})\right|^2.$$

Since $X_j/\sqrt{\sigma_j} - q_j(Y, \boldsymbol{X}) = X_j/\sqrt{\sigma_j} \cdot \mathbb{1}\{|X_j/\sqrt{\sigma_j}| > R \cdot \sqrt{\log d}\}$, by Cauchy-Schwarz inequality we have

$$\left|\mathbb{E}_{\mathbb{P}_{\boldsymbol{\theta}}}(X_j/\sqrt{\sigma_j}) - \mathbb{E}_{\mathbb{P}_{\boldsymbol{\theta}}} q_j(Y, \boldsymbol{X})\right|^2 \leq \mathbb{E}_{\mathbb{P}_{\boldsymbol{\theta}}}(X_j^2/\sigma_j) \cdot \mathbb{P}_{\boldsymbol{\theta}}\left(|X_j/\sqrt{\sigma_j}| > R \cdot \sqrt{\log d}\right). \tag{A.56}$$

Since $\|\boldsymbol{\mu}_0\|_\infty \vee \|\boldsymbol{\mu}_1\|_\infty \leq C_0$ and $\{X_j/\sqrt{\sigma_j}\}_{i=1}^d$ are sub-Gaussian random variables, for any $t > 0$, there exists a constant $C_1$ such that

$$\mathbb{P}_{\boldsymbol{\theta}}\left(|X_j/\sqrt{\sigma_j}| > t\right) \leq 2\exp(-C_1 t^2). \tag{A.57}$$

Thus setting $t = R \cdot \sqrt{\log d}$ for some sufficiently large $R$, by (A.56) and (A.57) we obtain

$$\left|\mathbb{E}_{\mathbb{P}_{\boldsymbol{\theta}}}(X_j/\sqrt{\sigma_j}) - \mathbb{E}_{\mathbb{P}_{\boldsymbol{\theta}}} q_j(Y, \boldsymbol{X})\right| \leq C_2 d^{-1}$$

for some constant $C_2$. Thus we have

$$\left|\left[\mathbb{E}_{\mathbb{P}_{\boldsymbol{\theta}}} q_j(Y, \boldsymbol{X})\right]^2 - \left[\mathbb{E}_{\mathbb{P}_{\boldsymbol{\theta}}}(X_j/\sqrt{\sigma_j})\right]^2\right| \leq 2C_0 \cdot C_2 d^{-1} + C_2^2 d^{-2} \leq 1/16 \cdot (\Delta\boldsymbol{\mu})_j^2/\sigma_j. \tag{A.58}$$

In addition, since $X_j^2/\sigma_j - 1 - \widetilde{q}_j(Y, \boldsymbol{X}) = (X_j^2/\sigma_j - 1) \cdot \mathbb{1}\{|X_j/\sqrt{\sigma_j}| > R \cdot \sqrt{\log d}\}$, for $\widetilde{q}_j$ defined in (3.14), we similarly we obtain

$$\left|\mathbb{E}_{\mathbb{P}_{\boldsymbol{\theta}}} \widetilde{q}_j(Y, \boldsymbol{X}) - \mathbb{E}_{\mathbb{P}_{\boldsymbol{\theta}}}(X_j^2/\sigma_j - 1)\right| \leq 1/16 \cdot (\Delta\boldsymbol{\mu})_j^2/\sigma_j. \tag{A.59}$$

Combining (A.58) and (A.59) we have

$$\mathbb{E}_{\mathbb{P}_{\boldsymbol{\theta}}} \widetilde{q}_j(Y, \boldsymbol{X}) - [\mathbb{E}_{\mathbb{P}_{\boldsymbol{\theta}}} q_j(Y, \boldsymbol{X})]^2 \geq 1/8 \cdot (\Delta\boldsymbol{\mu})_j^2/\sigma_j \ \text{ for all } \ j \in [d].$$

Taking supremum over $j \in [d]$, we have

$$\sup_{j \in [d]} \left\{\mathbb{E}_{\mathbb{P}_{\boldsymbol{\theta}}} \widetilde{q}_j(Y, \boldsymbol{X}) - [\mathbb{E}_{\mathbb{P}_{\boldsymbol{\theta}}} q_j(Y, \boldsymbol{X})]^2\right\} \geq 1/8 \cdot \sup_{j \in [d]} \left[(\Delta\boldsymbol{\mu})_j^2/\sigma_j\right]. \tag{A.60}$$

Note that the test function $\overline{\phi}$ involves $4d$ queries functions. Thus, for any $\boldsymbol{\theta} \in \mathcal{G}_0(\boldsymbol{\Sigma}) \cup \mathcal{G}_1(\boldsymbol{\Sigma}; \gamma_n)$, under $\mathbb{P}_{\boldsymbol{\theta}}$ the tolerance parameters for $q_j$ and $\widetilde{q}_j$ are given by

$$\tau_{q_j} \leq R\sqrt{\log d} \cdot \sqrt{[\log(4d/\xi)]/n}, \ \ \tau_{\widetilde{q}_j} \leq R^2 \log d \cdot \sqrt{[\log(4d/\xi)]/n}, \ \text{for all } j \in [d]. \quad \text{(A.61)}$$

Under the assumption that

$$\sup_{j \in [d]} (\Delta\boldsymbol{\mu})_j^2/\sigma_j = \Omega\left[\log^2 d \cdot \log(d/\xi)/(\alpha^2 n) \wedge \log d \cdot \sqrt{\log(d/\xi)/n}\right],$$

we have

$$\tau_{q_j} \vee \tau_{\widetilde{q}_j} \leq R^2 \log d \cdot \sqrt{[\log(4d/\xi)]/n}$$

$$\leq (1/C) \cdot \left\{\sup_{j \in [d]} [(\Delta\boldsymbol{\mu})_j^2/\sigma_j] \vee \alpha \cdot \sup_{j \in [d]} |(\Delta\boldsymbol{\mu})_j/\sqrt{\sigma_j}|\right\}, \quad \text{(A.62)}$$

where the absolute constant $C$ is the same as in (3.16). Note that we denote $R^2 \log d \cdot \sqrt{\log(4d/\xi)/n}$ by $\overline{\tau}_1$. Hence by (A.62), for any $\boldsymbol{\theta} \in \mathcal{G}_0(\boldsymbol{\Sigma})$, the type-I error of $\overline{\phi}_1$ is bounded by

$$\mathbb{\overline{P}}_{\boldsymbol{\theta}}\left[\sup_{j \in [d]} (Z_{\widetilde{q}_j} - Z_{q_j}^2) \geq C\overline{\tau}_1\right]$$

$$= \mathbb{\overline{P}}_{\boldsymbol{\theta}}\left(\bigcup_{j \in [d]} \left\{(Z_{\widetilde{q}_j} - Z_{q_j}^2) - \{\mathbb{E}_{\mathbb{P}_{\boldsymbol{\theta}}}\widetilde{q}_j(Y, \boldsymbol{X}) - [\mathbb{E}_{\mathbb{P}_{\boldsymbol{\theta}}} q_j(Y, \boldsymbol{X})]^2\} \geq C\overline{\tau}_1\right\}\right)$$

$$\leq \mathbb{\overline{P}}_{\boldsymbol{\theta}}\left(\bigcup_{j \in [d]} \left\{Z_{\widetilde{q}_j} - \mathbb{E}_{\mathbb{P}_{\boldsymbol{\theta}}}\widetilde{q}_j(Y, \boldsymbol{X}) \geq \overline{\tau}_1\right\}\right) + \mathbb{\overline{P}}_{\boldsymbol{\theta}}\left(\bigcup_{j \in [d]} \left\{Z_{q_j}^2 - [\mathbb{E}_{\mathbb{P}_{\boldsymbol{\theta}}} q_j(Y, \boldsymbol{X})]^2 \geq (C-1)\overline{\tau}_1\right\}\right).$$

For the first term, we have

$$\mathbb{\overline{P}}_{\boldsymbol{\theta}}\left(\bigcup_{j \in [d]} \left\{Z_{\widetilde{q}_j} - \mathbb{E}_{\mathbb{P}_{\boldsymbol{\theta}}}\widetilde{q}_j(Y, \boldsymbol{X}) \geq \overline{\tau}_1\right\}\right)$$

$$\leq \mathbb{\overline{P}}_{\boldsymbol{\theta}}\left(\bigcup_{j \in [d]} \left\{\left|Z_{\widetilde{q}_j} - \mathbb{E}_{\mathbb{P}_{\boldsymbol{\theta}}}\widetilde{q}_j(Y, \boldsymbol{X})\right| \geq \tau_{\widetilde{q}_j}\right\}\right) \leq \xi. \quad \text{(A.63)}$$

Note that under the null hypothesis $\boldsymbol{\theta} \in \mathcal{G}_0(\boldsymbol{\Sigma})$, we have $\mathbb{E}_{\mathbb{P}_{\boldsymbol{\theta}}} q_j(Y, \boldsymbol{X}) = \mu_{0,j}/\sqrt{\sigma_j}$. Under the assumption that $\|\boldsymbol{\mu}_0\|_\infty \vee \|\boldsymbol{\mu}_1\|_\infty \leq C_0$, when $n$ is sufficiently large such that

$$\overline{\tau}_1 \leq 3(C-1)^{-1} C_0/\sqrt{\sigma_j},$$

by $Z_{q_j}^2 - [\mathbb{E}_{\mathbb{P}_{\boldsymbol{\theta}}} q_j(Y, \boldsymbol{X})]^2 \geq (C-1)\overline{\tau}_1$ we have

$$|Z_{q_j} - \mathbb{E}_{\mathbb{P}_{\boldsymbol{\theta}}} q_j(Y, \boldsymbol{X})| \geq (C-1)\overline{\tau}_1 \cdot \sqrt{\sigma_j}/(3C_0). \quad \text{(A.64)}$$

Thus we can set absolute constant $C$ sufficiently large such that $|Z_{q_j} - \mathbb{E}_{\mathbb{P}_{\boldsymbol{\theta}}} q_j(Y, \boldsymbol{X})| \geq \overline{\tau}_1$. Thus by (A.64) we have

$$\mathbb{\overline{P}}_{\boldsymbol{\theta}}\left(\bigcup_{j \in [d]} \left\{Z_{q_j}^2 - [\mathbb{E}_{\mathbb{P}_{\boldsymbol{\theta}}} q_j(Y, \boldsymbol{X})]^2 \geq (C-1)\overline{\tau}_1\right\}\right)$$

$$\leq \mathbb{\overline{P}}_{\boldsymbol{\theta}}\left(\bigcup_{j \in [d]} \left\{\left|Z_{q_j} - \mathbb{E}_{\mathbb{P}_{\boldsymbol{\theta}}} q_j(Y, \boldsymbol{X})\right| \geq \tau_{q_j}\right\}\right) \leq \xi. \quad \text{(A.65)}$$

Combining (A.63) and (A.65), we can bound the type-I error of $\overline{\phi}_1$ by $2\xi$. For the type-I error of $\overline{\phi}_2$, we define $\boldsymbol{Z} = (2Y - 1) \cdot \boldsymbol{X}$. Under the data-generating model defined in (2.1) and (2.2), the distribution of $\boldsymbol{Z}$ is given by

$$\boldsymbol{Z} \sim \frac{1+\alpha}{4}\mathcal{N}(-\boldsymbol{\mu}_0, \boldsymbol{\Sigma}) + \frac{1+\alpha}{4}\mathcal{N}(\boldsymbol{\mu}_1, \boldsymbol{\Sigma}) + \frac{1-\alpha}{4}\mathcal{N}(\boldsymbol{\mu}_0, \boldsymbol{\Sigma}) + \frac{1-\alpha}{4}\mathcal{N}(-\boldsymbol{\mu}_1, \boldsymbol{\Sigma}).$$

Then by definition, for all $\boldsymbol{\theta} \in \mathcal{G}_0(\boldsymbol{\Sigma})$, we have

$$\mathbb{E}_{\mathbb{P}_{\boldsymbol{\theta}}}[\mathbf{v}^\top \text{diag}(\boldsymbol{\Sigma})^{-1/2} \boldsymbol{Z}] = 0, \ \text{for all } \mathbf{v} \in \mathcal{B}_2(1). \quad \text{(A.66)}$$

In addition, for any $\boldsymbol{\theta} \in \mathcal{G}_1(\boldsymbol{\Sigma}; \gamma_n)$, by the distribution of $\boldsymbol{Z}$, for all $\mathbf{v} \in \mathcal{B}_2(1)$, we have

$$\mathbb{E}_{\mathbb{P}_{\boldsymbol{\theta}}}[\mathbf{v}^\top \text{diag}(\boldsymbol{\Sigma})^{-1/2} \boldsymbol{Z}] = \alpha/2 \cdot \mathbf{v}^\top \text{diag}(\boldsymbol{\Sigma})^{-1/2} \Delta\boldsymbol{\mu}. \quad \text{(A.67)}$$

Moreover, by definition we have

$$\mathbf{v}^\top \text{diag}(\boldsymbol{\Sigma})^{-1/2} \boldsymbol{Z} - \overline{q}_{\mathbf{v}}(Y, \boldsymbol{X}) = \mathbf{v}^\top \text{diag}(\boldsymbol{\Sigma})^{-1/2} \boldsymbol{Z} \cdot \mathbb{1}\{|\mathbf{v}^\top \text{diag}(\boldsymbol{\Sigma})^{-1/2} \boldsymbol{Z}| \leq R\sqrt{\log d}\}.$$

By setting the constant $R$ sufficiently large, for any for any $\boldsymbol{\theta} \in \mathcal{G}_0(\boldsymbol{\Sigma}) \cup \mathcal{G}_1(\boldsymbol{\Sigma}; \gamma_n)$, we have

$$\left|\mathbb{E}_{\mathbb{P}_{\boldsymbol{\theta}}}\overline{q}_{\mathbf{v}}(Y, \boldsymbol{X}) - \mathbb{E}_{\mathbb{P}_{\boldsymbol{\theta}}}(\mathbf{v}^\top \text{diag}(\boldsymbol{\Sigma})^{-1/2} \boldsymbol{Z})\right| \leq \alpha/4 \cdot |(\Delta\boldsymbol{\mu})_j/\sqrt{\sigma_j}|.$$

Combining (A.66) and (A.67) we obtain that

$$\mathbb{E}_{\mathbb{P}_{\boldsymbol{\theta}}}\overline{q}_{\mathbf{v}}(Y,\boldsymbol{X}) \leq \alpha/4 \cdot |(\Delta\boldsymbol{\mu})_j/\sqrt{\sigma_j}| \ \text{ for all } \ \boldsymbol{\theta} \in \mathcal{G}_0(\boldsymbol{\Sigma});$$
$$\mathbb{E}_{\mathbb{P}_{\boldsymbol{\theta}}}\overline{q}_{\mathbf{v}}(Y,\boldsymbol{X}) \geq \alpha/4 \cdot |(\Delta\boldsymbol{\mu})_j/\sqrt{\sigma_j}| \ \text{ for all } \ \boldsymbol{\theta} \in \mathcal{G}_1(\boldsymbol{\Sigma},\gamma_n).$$

Thus, taking the supremum over $\mathcal{B}_2(1)$ yields

$$\sup_{\mathbf{v}\in\mathcal{B}_2(1)} \mathbb{E}_{\mathbb{P}_{\boldsymbol{\theta}}}\overline{q}_{\mathbf{v}}(Y,\boldsymbol{X}) \geq \alpha/4 \cdot \sup_{j\in[d]} |(\Delta\boldsymbol{\mu})_j/\sqrt{\sigma_j}| \ \text{ for all } \ \boldsymbol{\theta} \in \mathcal{G}_1(\boldsymbol{\Sigma},\gamma_n). \tag{A.68}$$

In addition, since we have $4d$ queries, by Definition 2.2, the tolerance parameters for $\overline{q}_{\mathbf{v}}$'s are bouded by

$$\tau_{\overline{q}_{\mathbf{v}}} \leq R\sqrt{\log d} \cdot \sqrt{[\log(4d/\xi)]/n}, \ \text{ for all } \mathbf{v} \in \mathcal{B}_2(1).$$

Note that we denote $\overline{\tau}_2 = R\sqrt{\log d} \cdot \sqrt{[\log(4d/\xi)]/n}$. Similar to (A.62), we have

$$\tau_{\overline{q}_{\mathbf{v}}} \leq \overline{\tau}_1 \leq (1/C) \cdot \left\{ \sup_{j\in[d]}[(\Delta\boldsymbol{\mu})_j^2/\sigma_j] \vee \sup_{j\in[d]} \alpha|(\Delta\boldsymbol{\mu})_j/\sqrt{\sigma_j}| \right\}. \tag{A.69}$$

Hence by (A.69), for any $\boldsymbol{\theta} \in \mathcal{G}_0(\boldsymbol{\Sigma})$, the type-I error of $\overline{\phi}_2$ is bounded by

$$\overline{\mathbb{P}}_{\boldsymbol{\theta}}\left( \sup_{\mathbf{v}\in\mathcal{B}_2(1)} Z_{\overline{q}_{\mathbf{v}}} \geq 2\overline{\tau}_1 \right) = \overline{\mathbb{P}}_{\boldsymbol{\theta}}\left( \bigcup_{\mathbf{v}\in\mathcal{B}_2(1)} \{ Z_{\overline{q}_{\mathbf{v}}} - \mathbb{E}_{\mathbb{P}_{\boldsymbol{\theta}}}[\overline{q}_{\mathbf{v}}(Y,\boldsymbol{X})] > \overline{\tau}_1 \} \right)$$

$$\leq \overline{\mathbb{P}}_{\boldsymbol{\theta}}\left( \bigcup_{\mathbf{v}\in\mathcal{B}_2(1)} \{ |Z_{\overline{q}_{\mathbf{v}}} - \mathbb{E}_{\mathbb{P}_{\boldsymbol{\theta}}}[\overline{q}_{\mathbf{v}}(Y,\boldsymbol{X})]| \geq \tau_{\overline{q}_{\mathbf{v}}} \} \right) \leq \xi. \tag{A.70}$$

Combining (A.63), (A.65), and (A.70), we have

$$\overline{\mathbb{P}}_{\boldsymbol{\theta}}(\overline{\phi} = 1) \leq 3\xi, \ \text{ for all } \ \boldsymbol{\theta} \in \mathcal{G}_0(\boldsymbol{\Sigma}).$$

**Type-II error.** Now we consider $\boldsymbol{\theta} \in \mathcal{G}_1(\boldsymbol{\Sigma};\gamma_n)$. Note that $\overline{\phi} = 0$ if $\overline{\phi}_1 = 0$ and $\overline{\phi}_2 = 0$. Thus, for any $\boldsymbol{\theta} \in \mathcal{G}_1(\boldsymbol{\Sigma};\gamma_n)$, we have

$$\overline{\mathbb{P}}_{\boldsymbol{\theta}}(\overline{\phi} = 0) = \overline{\mathbb{P}}_{\boldsymbol{\theta}}(\overline{\phi}_1 = 0 \cap \overline{\phi}_2 = 0) \leq \overline{\mathbb{P}}_{\boldsymbol{\theta}}(\overline{\phi}_1 = 0) \wedge \overline{\mathbb{P}}_{\boldsymbol{\theta}}(\overline{\phi}_2 = 0).$$

Recall that we denote $\Delta\boldsymbol{\mu} = \boldsymbol{\mu}_1 - \boldsymbol{\mu}_0$. Similar to the proof of Theorem 3.2, we consider two cases of the condition

$$\sup_{j\in[d]}(\Delta\boldsymbol{\mu})_j^2/\sigma_j = \Omega\left[ \log(d/\xi)/(\alpha^2 \cdot n) \wedge \sqrt{\log(d/\xi)/n} \right].$$

**Case (i).** We show that the type-II error of $\overline{\phi}_1$ is negligible under the assumption that

$$\sup_{j\in[d]}(\Delta\boldsymbol{\mu})_j^2/\sigma_j = \Omega\left[ \sqrt{\log(d/\xi)/n} \right].$$

Let $j^* = \mathrm{argmax}_{j\in[d]}(\Delta\boldsymbol{\mu})_j^2/\sigma_j$. Then by (A.62), when we have

$$1 + C\overline{\tau} \leq (\Delta\boldsymbol{\mu})_{j^*}^2/\sigma_{j^*} + 1 - C\overline{\tau} = \mathbb{E}_{\mathbb{P}_{\boldsymbol{\theta}}}\widetilde{q}_{j^*}(Y,\boldsymbol{X}) - [\mathbb{E}_{\mathbb{P}_{\boldsymbol{\theta}}}q_{j^*}(Y,\boldsymbol{X})]^2 - C\overline{\tau}. \tag{A.71}$$

Thus combining (A.58), (A.59), and (A.71), we have

$$\overline{\mathbb{P}}_{\boldsymbol{\theta}}\left[ \sup_{j\in[d]}(Z_{q_j} - Z_{\widetilde{q}_j}^2) < C\overline{\tau}_1 \right]$$

$$\leq \overline{\mathbb{P}}_{\boldsymbol{\theta}}\left\{ Z_{\widetilde{q}_j^*} - Z_{q_j}^2 < \mathbb{E}_{\mathbb{P}_{\boldsymbol{\theta}}}\widetilde{q}_{j^*}(Y,\boldsymbol{X}) - [\mathbb{E}_{\mathbb{P}_{\boldsymbol{\theta}}}q_{j^*}(Y,\boldsymbol{X})]^2 - C\overline{\tau}_1 \right\}$$

$$\leq \overline{\mathbb{P}}_{\boldsymbol{\theta}}\left[ \mathbb{E}_{\mathbb{P}_{\boldsymbol{\theta}}}\widetilde{q}_{j^*}(Y,\boldsymbol{X}) - Z_{\widetilde{q}_{j^*}} > \overline{\tau}_1 \right] + \overline{\mathbb{P}}_{\boldsymbol{\theta}}\left\{ [\mathbb{E}_{\mathbb{P}_{\boldsymbol{\theta}}}q_{j^*}(Y,\boldsymbol{X})]^2 - Z_{q_j^*}^2 > (C-1)\overline{\tau}_1 \right\}. \tag{A.72}$$

Moreover, by (A.62) the first term on the right-hand side of (A.72) can be further bounded by

$$\overline{\mathbb{P}}_{\boldsymbol{\theta}}\left[ \mathbb{E}_{\mathbb{P}_{\boldsymbol{\theta}}}\widetilde{q}_{j^*}(Y,\boldsymbol{X}) - Z_{\widetilde{q}_{j^*}} > \overline{\tau}_1 \right]$$

$$\leq \overline{\mathbb{P}}_{\boldsymbol{\theta}}\left\{ \mathbb{E}_{\mathbb{P}_{\boldsymbol{\theta}}}\widetilde{q}_{j^*}(Y,\boldsymbol{X}) - Z_{\widetilde{q}_{j^*}} \geq \tau_{\widetilde{q}_{j^*}} \right\}$$

$$\leq \overline{\mathbb{P}}_{\boldsymbol{\theta}}\left( \bigcup_{j\in[d]} \{ |Z_{\widetilde{q}_j} - \mathbb{E}_{\mathbb{P}_{\boldsymbol{\theta}}}\widetilde{q}_j(Y,\boldsymbol{X})]| \geq \tau_{\widetilde{q}_j} \} \right) \leq \xi. \tag{A.73}$$

Similarly, for the second term on the right-hand side of (A.72), by (A.62) and (A.64)we have

$$\overline{\mathbb{P}}_{\boldsymbol{\theta}}\left\{[\mathbb{E}_{\mathbb{P}_{\boldsymbol{\theta}}} q_{j^*}(Y, \boldsymbol{X})]^2 - Z_{q_j^*}^2 > (C-1)\overline{\tau}_1\right\}$$

$$\leq \overline{\mathbb{P}}_{\boldsymbol{\theta}}\left(\bigcup_{j\in[d]}\left\{|Z_{q_j} - \mathbb{E}_{\mathbb{P}_{\boldsymbol{\theta}}}\widetilde{q}_j(Y, \boldsymbol{X})]| \geq \tau_{q_j}\right\}\right) \leq \xi. \tag{A.74}$$

Therefore, combining (A.73) and (A.74), we conclude that the type-II error of $\overline{\phi}_2$ is no more than $2\xi$.

**Case (ii).** Now we assume study the type-II error of $\overline{\phi}_2$ under the assumption that

$$\sup_{j\in[d]}(\Delta\boldsymbol{\mu})_j^2/\sigma_j = \Omega\left[\log(d/\xi)/(\alpha^2 \cdot n)\right].$$

Let $j^* = \operatorname{argmax}_{j\in[d]}(\Delta\boldsymbol{\mu})_j^2/\sigma_j$ and $\mathbf{v}^* = \operatorname{argmax}_{\mathbf{v}\in\mathcal{B}_2(1)}\mathbb{E}_{\mathbb{P}_{\boldsymbol{\theta}}}\overline{q}_{\mathbf{v}}(Y, \boldsymbol{X})$. Then by (A.62) and (A.68), when $C > 4$ we have

$$2\overline{\tau}_2 \leq \alpha/2 \cdot \sup_{j\in[d]}|(\Delta\boldsymbol{\mu})_j/\sqrt{\sigma_j}| - 2\overline{\tau}_2 = \mathbb{E}_{\mathbb{P}_{\boldsymbol{\theta}}}\overline{q}_{\mathbf{v}^*}(Y, \boldsymbol{X}) - 2\overline{\tau}_2. \tag{A.75}$$

Then by (A.69) and (A.75) the type-II error of $\overline{\phi}_2$ is bounded by

$$\overline{\mathbb{P}}_{\boldsymbol{\theta}}\left(\sup_{\mathbf{v}\in\mathcal{B}_2(1)} Z_{\overline{q}_{\mathbf{v}}} < 2\overline{\tau}_2\right) \leq \overline{\mathbb{P}}_{\boldsymbol{\theta}}\left[\sup_{\mathbf{v}\in\mathcal{B}_2(1)} Z_{\overline{q}_{\mathbf{v}}} < \mathbb{E}_{\mathbb{P}_{\boldsymbol{\theta}}}\overline{q}_{\mathbf{v}^*}(Y, \boldsymbol{X}) - 2\overline{\tau}_2\right]$$

$$\leq \overline{\mathbb{P}}_{\boldsymbol{\theta}}\left[Z_{\overline{q}_{\mathbf{v}^*}} < \mathbb{E}_{\mathbb{P}_{\boldsymbol{\theta}}}\overline{q}_{\mathbf{v}^*}(Y, \boldsymbol{X}) - 2\overline{\tau}_2\right]$$

$$\leq \overline{\mathbb{P}}_{\boldsymbol{\theta}}\left(\bigcup_{\mathbf{v}\in\mathcal{B}_2(1)}\left\{|Z_{\overline{q}_{\mathbf{v}}} - \mathbb{E}_{\mathbb{P}_{\boldsymbol{\theta}}}[\overline{q}_{\mathbf{v}}(Y, \boldsymbol{X})]| \geq \tau_{\overline{q}_{\mathbf{v}}}\right\}\right) \leq \xi. \tag{A.76}$$

Thus by (A.76), the type-II error of $\overline{\phi}_2$ is no more than $\xi$. Then together with Case (i), we have $\overline{\mathbb{P}}_{\boldsymbol{\theta}}(\overline{\phi}) \leq 2\xi$ for all $\boldsymbol{\theta} \in \mathcal{G}_1(\boldsymbol{\Sigma}; \gamma_n)$. Therefore the total risk of $\overline{\phi}$ is bounded by

$$\overline{R}_n(\overline{\phi}) = \sup_{\boldsymbol{\theta}\in\mathcal{G}_0(\boldsymbol{\Sigma})}\overline{\mathbb{P}}_{\boldsymbol{\theta}}(\overline{\phi} = 1) + \sup_{\boldsymbol{\theta}\in\mathcal{G}_1(\boldsymbol{\Sigma};\gamma_n)}\overline{\mathbb{P}}_{\boldsymbol{\theta}}(\overline{\phi} = 0) \leq 5\xi.$$

# B Proofs for Technical Lemmas

In this section, we prove the technical lemmas which appear in the proofs of the main results.

## B.1 Proof of Lemma A.1

Under $\mathbb{P}_0$, $\boldsymbol{X}$ and $Y$ are independent with $\boldsymbol{X} \sim \mathcal{N}(\mathbf{0}, \mathbf{I})$ and $Y$ is uniform over $\{0, 1\}$. We denote by $f(\mathbf{x}; \boldsymbol{\mu})$ the density of $\mathcal{N}(\boldsymbol{\mu}, \mathbf{I})$ and by $p_0(y, \mathbf{x})$ the density of $\mathbb{P}_0$. Then for any $y \in \{0, 1\}$ and $\mathbf{x} \in \mathbb{R}^d$, we have $p_0(y, \mathbf{x}) = 1/2 \cdot f(\mathbf{x}; \mathbf{0})$. In addition, for any $\mathbf{v} \in \mathcal{H}(s)$, we denote the density of $\mathbb{P}_{\mathbf{v}}$ by $p_{\mathbf{v}}(y, \mathbf{x})$. By the definition of the statistical model, we have

$$p_{\mathbf{v}}(1, \mathbf{x}) = (1+\alpha)/4 \cdot f(\mathbf{x}; \mathbf{v}/2) + (1-\alpha)/4 \cdot f(\mathbf{x}; -\mathbf{v}/2),$$

$$p_{\mathbf{v}}(0, \mathbf{x}) = (1-\alpha)/4 \cdot f(\mathbf{x}; \mathbf{v}/2) + (1+\alpha)/4 \cdot f(\mathbf{x}; -\mathbf{v}/2).$$

Thus for any $y \in \{0, 1\}$ and $\mathbf{x} \in \mathbb{R}^d$, we have

$$\frac{d\mathbb{P}_{\mathbf{v}}}{d\mathbb{P}_0}(y, \mathbf{x}) = \frac{1}{2}\cdot\left[\frac{f(\mathbf{x}; \mathbf{v}/2)}{f(\mathbf{x}; \mathbf{0})} + \frac{f(\mathbf{x}; -\mathbf{v}/2)}{f(\mathbf{x}; \mathbf{0})}\right] + \frac{\alpha(2y-1)}{2}\cdot\left[\frac{f(\mathbf{x}; \mathbf{v}/2)}{f(\mathbf{x}; \mathbf{0})} - \frac{f(\mathbf{x}; -\mathbf{v}/2)}{f(\mathbf{x}; \mathbf{0})}\right]. \tag{B.1}$$

Note that by definition, for any $\boldsymbol{\mu} \in \mathbb{R}^d$, we have

$$g(\mathbf{x}; \boldsymbol{\mu}) := f(\mathbf{x}; \boldsymbol{\mu})/f(\mathbf{x}; \mathbf{0}) = \exp(\boldsymbol{\mu}^\top\mathbf{x} - 1/2 \cdot \|\boldsymbol{\mu}\|_2^2).$$

Thus (B.1) is reduced to

$$\frac{d\mathbb{P}_{\mathbf{v}}}{d\mathbb{P}_0}(y, \mathbf{x}) = [g(\mathbf{x}, \mathbf{v}/2) + g(\mathbf{x}; -\mathbf{v}/2)]/2 + \alpha(2y-1)\cdot[g(\mathbf{x}, \mathbf{v}/2) - g(\mathbf{x}; -\mathbf{v}/2)]/2. \tag{B.2}$$

For any $\mathbf{v}_1, \mathbf{v}_2 \in \mathcal{H}(s)$, by (B.2) we have

$$\mathbb{E}_{\mathbb{P}_0}\left[\frac{d\mathbb{P}_{\mathbf{v}_1}}{d\mathbb{P}_0}\frac{d\mathbb{P}_{\mathbf{v}_2}}{d\mathbb{P}_0}(Y, \boldsymbol{X})\right]$$

$$= \mathbb{E}_{\mathbb{P}_0}\{[g(\boldsymbol{X}, \mathbf{v}_1/2) + g(\boldsymbol{X}; -\mathbf{v}_1/2)]\cdot[g(\boldsymbol{X}, \mathbf{v}_2/2) + g(\boldsymbol{X}; -\mathbf{v}_2/2)]/4\}$$

$$+ \alpha^2\cdot\mathbb{E}_{\mathbb{P}_0}\{[g(\boldsymbol{X}, \mathbf{v}_1/2) - g(\boldsymbol{X}; -\mathbf{v}_1/2)]\cdot[g(\boldsymbol{X}, \mathbf{v}_2/2) - g(\boldsymbol{X}; -\mathbf{v}_2/2)]/4\}, \tag{B.3}$$

where we use the independence of $Y$ and $\boldsymbol{X}$ under $\mathbb{P}_0$. In what follows, we calculate the two terms on the right-hand side of (B.3), respectively. Let $\eta_1$ and $\eta_2$ be two independent Rademacher random variables over $\{-1, 1\}$. Then for $\ell \in \{1, 2\}$, we have

$$[g(\boldsymbol{X}, \mathbf{v}_\ell/2) + g(\boldsymbol{X}; -\mathbf{v}_\ell/2)]/2 = \mathbb{E}_{\eta_\ell}\left[g(\boldsymbol{X}, \eta_\ell \mathbf{v}_\ell/2)\right], \tag{B.4}$$

$$[g(\boldsymbol{X}, \mathbf{v}_\ell/2) - g(\boldsymbol{X}; -\mathbf{v}_\ell/2)]/2 = \mathbb{E}_{\eta_\ell}\left[\eta_\ell \cdot g(\boldsymbol{X}, \eta_\ell \mathbf{v}_\ell/2)\right]. \tag{B.5}$$

Then by (B.4) and (B.5) we have

$$\begin{aligned}
&\mathbb{E}_{\mathbb{P}_0}\left\{[g(\boldsymbol{X}, \mathbf{v}_1/2) + g(\boldsymbol{X}; -\mathbf{v}_1/2)] \cdot [g(\boldsymbol{X}, \mathbf{v}_2/2) + g(\boldsymbol{X}; -\mathbf{v}_2/2)]/4\right\} \\
&\quad = \mathbb{E}_{\mathbb{P}_0}\mathbb{E}_{\eta_1, \eta_2}\left[g(\boldsymbol{X}; \eta_1 \mathbf{v}_1/2) \cdot g(\boldsymbol{X}; \eta_2 \mathbf{v}_2/2)\right] \\
&\quad = \mathbb{E}_{\eta_1, \eta_2}\mathbb{E}_{\mathbb{P}_0}\exp\left[\boldsymbol{X}^\top(\eta_1\mathbf{v}_1 + \eta_2\mathbf{v}_2)/2 - 1/8 \cdot (\|\mathbf{v}_1\|_2^2 + \|\mathbf{v}_2\|_2^2)\right]. \tag{B.6}
\end{aligned}$$

Using the moment-generating function of $\boldsymbol{X}$, by (B.6) we have

$$\begin{aligned}
&\mathbb{E}_{\mathbb{P}_0}\left\{[g(\boldsymbol{X}, \mathbf{v}_1/2) + g(\boldsymbol{X}; -\mathbf{v}_1/2)] \cdot [g(\boldsymbol{X}, \mathbf{v}_2/2) + g(\boldsymbol{X}; -\mathbf{v}_2/2)]/4\right\} \\
&\quad = \mathbb{E}_{\eta_1, \eta_2}\left[\exp(1/2 \cdot \eta_1\eta_2 \cdot \mathbf{v}_1^\top \mathbf{v}_2) = \cosh(1/2 \cdot \langle \mathbf{v}_1, \mathbf{v}_2 \rangle)\right].
\end{aligned}$$

Similarly, for (B.5) we have

$$\begin{aligned}
&\mathbb{E}_{\mathbb{P}_0}\left\{[g(\boldsymbol{X}, \mathbf{v}_1/2) - g(\boldsymbol{X}; -\mathbf{v}_1/2)] \cdot [g(\boldsymbol{X}, \mathbf{v}_2/2) - g(\boldsymbol{X}; -\mathbf{v}_2/2)]/4\right\} \\
&\quad = \mathbb{E}_{\mathbb{P}_0}\mathbb{E}_{\eta_1, \eta_2}\left[\eta_1\eta_2 \cdot g(\boldsymbol{X}; \eta_1 \mathbf{v}_1/2) \cdot g(\boldsymbol{X}; \eta_2 \mathbf{v}_2/2)\right] \\
&\quad = \mathbb{E}_{\eta_1, \eta_2}\left[\eta_1\eta_2 \cdot \exp(1/2 \cdot \eta_1\eta_2 \cdot \mathbf{v}_1^\top \mathbf{v}_2)\right] = \sinh(1/2 \cdot \langle \mathbf{v}_1, \mathbf{v}_2 \rangle).
\end{aligned}$$

Thus we conclude the proof of Lemma A.1.

### B.2 Proof of Lemma A.2

It is straightforward to verify (A.4) holds when $x = 0$. We focus on region $x > 0$. It is then sufficient to prove the result for these two cases below.

**Case 1:** We consider the case $v \le 1/(2x) \cdot \log[\cosh(2x)]$. Then we need to prove

$$\cosh(x) + v\sinh(x) \le \cosh(2x). \tag{B.7}$$

Using the bound of $v$, it remains to show the function

$$f(x) = 1/(2x) \cdot \log[\cosh(2x)] \cdot \sinh(x) + \cosh(x) - \cosh(2x) \le 0.$$

holds for all $x > 0$. It's easy to verify $f(x)$ is monotonically decreasing over $(0, \infty]$ and $\lim_{x \to 0} f(x) = 0$. We thus finish proving (B.7).

**Case 2:** We consider the case $v \ge 1/(2x) \cdot \log[\cosh(2x)]$. We would like to show

$$\cosh(x) + v\sinh(x) \le \exp(2vx). \tag{B.8}$$

Let us define $g(v) := \exp(2vx) - \cosh(x) - v\sinh(x)$. We have that for any $x \ge 0$,

$$g'(v) = 2x\exp(2vx) - \sinh(x) \ge 2x\cosh(2x) - \sinh(x) \ge 0.$$

Hence, $g(v)$ is a monotonically increasing function. We thus have

$$\begin{aligned}
g(v) &\ge g\left\{1/(2x) \cdot \log[\cosh(2x)]\right\} \\
&= \cosh(2x) - \cosh(x) - 1/(2x) \cdot \log[\cosh(2x)] \cdot \sinh(x) = -f(x) \ge 0.
\end{aligned}$$

We thus finish proving (B.8).

## C Supporting Lemmas

In this section we list the supporting lemmas that establish two concentration inequalities for Gaussian random variables.

**Lemma C.1** ($\chi^2$-tail bound, [16]). *Let $X_1, \ldots, X_n$ be $n$ i.i.d. standard normal random variables. For all $t \in (0, 1)$,*

$$\mathbb{P}\left(\left|\frac{1}{n}\sum_{i=1}^{n} X_i^2 - 1\right| \ge t\right) \le 2\exp(-nt^2/8).$$

**Lemma C.2** (Gaussian covariance estimation, [25]). *Suppose $\{\boldsymbol{X}_i\}_{i=1}^{n}$ are $n$ i.i.d. Gaussian random vectors in $\mathbb{R}^d$ and $\boldsymbol{X}_1 \sim \mathcal{N}(\boldsymbol{0}, \boldsymbol{\Sigma})$. For every $\epsilon \in (0, 1)$, and $t \ge 1$, if $n \ge C(t/\epsilon)^2 d$ for some*

*constant $C$, then with probability at least $1 - 2e^{-t^2 n}$,*

$$\|\widehat{\boldsymbol{\Sigma}} - \boldsymbol{\Sigma}\|_2 \leq \epsilon \|\boldsymbol{\Sigma}\|_2,$$

*where $\widehat{\boldsymbol{\Sigma}} := 1/n \cdot \sum_{i=1}^{n} \boldsymbol{X}_i \boldsymbol{X}_i^{\top}$.*

[Supplementary Material 2]

# A Proofs of the Main Results

## A.1 Proof of Theorem 3.1

In this section, we prove the information-theoretic lower bound. In specific, we focus on the restricted testing problem

$$H_0 : \boldsymbol{\theta} = (\mathbf{0}, \mathbf{0}, \mathbf{I}, \alpha) \text{ versus. } H_1 : \boldsymbol{\theta} = (-\mathbf{v}/2, \mathbf{v}/2, \mathbf{I}, \alpha), \tag{A.1}$$

where

$$\mathbf{v} \in \mathcal{H}(s) := \{\mathbf{u} \in \{0, \beta\}^d \colon \|\mathbf{u}\|_0 = s\}.$$

Here we set $s\beta^2 = \gamma_n$ to ensure that $(-\mathbf{v}/2, \mathbf{v}/2, \mathbf{I}, \alpha)$ belongs to the alternative parameter space $\mathcal{G}(\boldsymbol{\Sigma}; \gamma_n)$. For notational simplicity, we denote the distribution of model $(-\mathbf{v}/2, \mathbf{v}/2, \mathbf{I}, \alpha)$ by $\mathbb{P}_{\mathbf{v}}$ and the product distribution of $n$ i.i.d. samples by $\mathbb{P}_{\mathbf{v}}^n$. By the definition of the minimax risk in (2.4), we have

$$\sup_{\boldsymbol{\Sigma}} R_n^* [\mathcal{G}_0(\boldsymbol{\Sigma}), \mathcal{G}_1(\boldsymbol{\Sigma}; \gamma_n)] \geq \inf_{\phi} \left[ \mathbb{P}_{\mathbf{0}}^n(\phi = 1) + \frac{1}{|\mathcal{H}(s)|} \sum_{\mathbf{v} \in \mathcal{H}(s)} \mathbb{P}_{\mathbf{v}}^n(\phi = 0) \right].$$

We thus reduce the minimax risk to the risk of a simple-against-simple hypothesis test where the alternative hypothesis corresponds to a uniform mixture of $\{\mathbb{P}_{\mathbf{v}} : \mathbf{v} \in \mathcal{H}(s)\}$. For notational simplicity, we define $\mathbb{P}_{\mathcal{H}}^n := 1/|\mathcal{H}(s)| \cdot \sum_{\mathbf{v} \in \mathcal{H}(s)} \mathbb{P}_{\mathbf{v}}^n$. By Neyman-Pearson Lemma, we have

$$R_n^* [\mathcal{G}_0, \mathcal{G}_1(\boldsymbol{\Sigma}; \gamma_n)] \geq 1 - \mathrm{TV}(\mathbb{P}_{\mathbf{0}}^n, \mathbb{P}_{\mathcal{H}}^n).$$

Using Pinsker's inequality $\mathrm{TV}(\mathbb{P}_{\mathbf{0}}^n, \mathbb{P}_{\mathcal{H}}^n) \leq \sqrt{D_{\chi^2}(\mathbb{P}_{\mathcal{H}}^n, \mathbb{P}_{\mathbf{0}}^n)}$, for showing $R_n^*[\mathcal{G}_0(\boldsymbol{\Sigma}), \mathcal{G}_1(\boldsymbol{\Sigma}; \gamma_n)] \to 1$ as $n$ goes to infinity, it suffices to show that $D_{\chi^2}(\mathbb{P}_{\mathcal{H}}^n, \mathbb{P}_{\mathbf{0}}^n) = o(1)$. By calculation we have

$$D_{\chi^2}(\mathbb{P}_{\mathcal{H}}^n, \mathbb{P}_{\mathbf{0}}^n) = \mathbb{E}_{\mathbb{P}_{\mathbf{0}}^n} \left\{ \left[ \frac{\mathrm{d}\mathbb{P}_{\mathcal{H}}^n}{\mathrm{d}\mathbb{P}_{\mathbf{0}}^n}(Y, \boldsymbol{X}) - 1 \right]^2 \right\} = \mathbb{E}_{\mathbb{P}_{\mathbf{0}}^n} \left\{ \left[ \frac{\mathrm{d}\mathbb{P}_{\mathcal{H}}^n}{\mathrm{d}\mathbb{P}_{\mathbf{0}}^n}(Y, \boldsymbol{X}) \right]^2 \right\} - 1$$

$$= \frac{1}{|\mathcal{H}(s)|^2} \sum_{\mathbf{v}_1, \mathbf{v}_2 \in \mathcal{H}(s)} \mathbb{E}_{\mathbb{P}_{\mathbf{0}}^n} \left[ \frac{\mathrm{d}\mathbb{P}_{\mathbf{v}_1}^n \mathrm{d}\mathbb{P}_{\mathbf{v}_2}^n}{\mathrm{d}\mathbb{P}_{\mathbf{0}}^n \mathrm{d}\mathbb{P}_{\mathbf{0}}^n}(Y, \boldsymbol{X}) \right] - 1$$

$$= \frac{1}{|\mathcal{H}(s)|^2} \sum_{\mathbf{v}_1, \mathbf{v}_2 \in \mathcal{H}(s)} \left\{ \mathbb{E}_{\mathbb{P}_{\mathbf{0}}} \left[ \frac{\mathrm{d}\mathbb{P}_{\mathbf{v}_1} \mathrm{d}\mathbb{P}_{\mathbf{v}_2}}{\mathrm{d}\mathbb{P}_{\mathbf{0}} \mathrm{d}\mathbb{P}_{\mathbf{0}}}(Y, \boldsymbol{X}) \right] \right\}^n - 1. \tag{A.2}$$

We utilize the following lemma to obtain an upper bound for the last term of (A.2). See §B.1 for the proof.

**Lemma A.1.** *For any $\mathbf{v}_1, \mathbf{v}_2 \in \mathcal{H}(s)$, we have*

$$\mathbb{E}_{\mathbb{P}_{\mathbf{0}}} \left[ \frac{\mathrm{d}\mathbb{P}_{\mathbf{v}_1}}{\mathrm{d}\mathbb{P}_{\mathbf{0}}} \frac{\mathrm{d}\mathbb{P}_{\mathbf{v}_2}}{\mathrm{d}\mathbb{P}_{\mathbf{0}}}(Y, \boldsymbol{X}) \right] = \cosh\left(\langle \mathbf{v}_1, \mathbf{v}_2 \rangle / 2\right) + \alpha^2 \sinh\left(\langle \mathbf{v}_1, \mathbf{v}_2 \rangle / 2\right).$$

By Lemma A.1, we have

$$D_{\chi^2}(\mathbb{P}_{\mathcal{H}}^n, \mathbb{P}_{\mathbf{0}}^n)$$
$$= \frac{1}{|\mathcal{H}(s)|^2} \sum_{\mathbf{v}_1, \mathbf{v}_2 \in \mathcal{H}(s)} \left[ \cosh\left(1/2 \cdot \langle \mathbf{v}_1, \mathbf{v}_2 \rangle\right) + \alpha^2 \sinh\left(1/2 \cdot \langle \mathbf{v}_1, \mathbf{v}_2 \rangle\right) \right]^n - 1. \tag{A.3}$$

We define $\mathcal{C} := \{\mathcal{S} \subseteq [d] : |\mathcal{S}| = s\}$, and let $\mathbb{U}_{\mathcal{C}}$ be the uniform distribution over $\mathcal{C}$. Let $\mathcal{S}_1, \mathcal{S}_2 \sim \mathbb{U}_{\mathcal{C}}$ be two independent random sets. Then by (A.3), we have

$$D_{\chi^2}(\mathbb{P}_{\mathcal{H}}^n, \mathbb{P}_{\mathbf{0}}^n) = \mathbb{E}_{\mathcal{S}_1, \mathcal{S}_2} \left[ \cosh(\beta^2/2 \cdot |\mathcal{S}_1 \cap \mathcal{S}_2|) + \alpha^2 \sinh(\beta^2/2 \cdot |\mathcal{S}_1 \cap \mathcal{S}_2|) \right]^n - 1.$$

We use the next lemma, proved in §B.2, to bound the above right-hand side.

**Lemma A.2.** *For any $x \geq 0$ and $v \in [0, 1]$, we have*

$$\cosh(x) + v \sinh(x) \leq \exp(2vx) \vee \cosh(2x). \tag{A.4}$$

Proceeding with this result and letting random variable $Z \sim |\mathcal{S}_1 \cap \mathcal{S}_2|$, we have

$$
\begin{aligned}
D_{\chi^2}(\mathbb{P}^n_{\mathcal{H}}, \mathbb{P}^n_{\mathbf{0}}) &\leq \mathbb{E}_Z \left[ \exp(\alpha^2 \beta^2 Z) \vee \cosh(\beta^2 Z) \right]^n - 1 \\
&= \mathbb{E}_Z \left[ \exp(n\alpha^2 \beta^2 Z) \vee \cosh(\beta^2 Z)^n \right] - 1 \\
&= \mathbb{E}_Z \left\{ \exp(n\alpha^2 \beta^2 Z) \vee \mathbb{E}_U \left[ \exp(\beta^2 ZU) \right] \right\} - 1, \quad\quad\quad\quad (A.5)
\end{aligned}
$$

where in the last step, we introduce a random variable $U$ that is the summation of $n$ independent Rademacher random variables over $\{-1, 1\}$. Then we have $\cosh(\beta^2 Z)^n = \mathbb{E}_U[\exp(\beta^2 ZU)]$. By (A.5), we have

$$
\begin{aligned}
D_{\chi^2}(\mathbb{P}^n_{\mathcal{H}}, \mathbb{P}^n_{\mathbf{0}}) &\leq \mathbb{E}_Z \mathbb{E}_U \left[ \exp(n\alpha^2 \beta^2 Z) \vee \exp(\beta^2 ZU) \right] - 1 \\
&= \mathbb{E}_U \mathbb{E}_Z \left\{ \exp(n\alpha^2 \beta^2) \vee \exp(\beta^2 U) \right\}^Z - 1 \\
&\leq \mathbb{E}_U \left\{ \sup_{\mathcal{S}_1 \in \mathcal{C}} \mathbb{E}_{\mathcal{S}_2} \left[ \exp(n\alpha^2 \beta^2) \vee \exp(\beta^2 U) \right]^{|\mathcal{S}_1 \cap \mathcal{S}_2|} \right\} - 1. \quad (A.6)
\end{aligned}
$$

Now we turn to bound the expectation over $\mathcal{S}_2$ in (A.6). For any fixed $\mathcal{S}_1$, we have

$$
|\mathcal{S}_1 \cap \mathcal{S}_2| = \sum_{i \in \mathcal{S}_1} V_i,
$$

where $V_i$ is binary random variable that indicates whether $i \in \mathcal{S}_2$. It is known that $V_1, \ldots, V_d$ are negative associated. Hence we have

$$
\begin{aligned}
\mathbb{E}_{\mathcal{S}_2} \left[ \exp(n\alpha^2 \beta^2) \vee \exp(\beta^2 U) \right]^{|\mathcal{S}_1 \cap \mathcal{S}_2|} &\leq \prod_{i \in \mathcal{S}_1} \mathbb{E}_{V_i} \left[ \exp(n\alpha^2 \beta^2) \vee \exp(\beta^2 U) \right]^{V_i} \\
&= \left\{ 1 + s/d \cdot \left[ \exp(n\alpha^2 \beta^2) \vee \exp(\beta^2 U) - 1 \right] \right\}^s. \quad (A.7)
\end{aligned}
$$

Plugging (A.7) into (A.6) and expanding the polynomial term, we have

$$
\begin{aligned}
D_{\chi^2}(\mathbb{P}^n_{\mathcal{H}}, \mathbb{P}^n_{\mathbf{0}}) &\leq \sum_{k=1}^s \binom{s}{k} \cdot (s/d)^k \cdot \mathbb{E}_U \left[ \exp(n\alpha^2 \beta^2) \vee \exp(\beta^2 U) - 1 \right]^k \\
&= \sum_{k=1}^s \binom{s}{k} \cdot (s/d)^k \cdot \left( \left[ \exp(n\alpha^2 \beta^2) - 1 \right]^k \cdot \mathbb{P}(U < n\alpha^2) \right. \\
&\quad\quad + \left. \mathbb{E}_U \left\{ \left[ \exp(\beta^2 U) - 1 \right]^k \mid U \geq \alpha^2 n \right\} \cdot \mathbb{P}(U \geq n\alpha^2) \right), \\
&\leq T_1 + T_2,
\end{aligned}
$$

where $T_1$ and $T_2$ are defined as

$$
T_1 := \sum_{k=1}^s \binom{s}{k} \cdot (s/d)^k \cdot \left[ \exp(n\alpha^2 \beta^2) - 1 \right]^k
$$

$$
T_2 := \sum_{k=1}^s \binom{s}{k} \cdot (s/d)^k \cdot \mathbb{E}_U \left\{ \left[ \exp(\beta^2 U) - 1 \right]^k \mid U \geq 0 \right\} \cdot \mathbb{P}(U \geq 0).
$$

It remains to bound $T_1$ and $T_2$ respectively.

**Bounding $T_1$.** Under condition $s\beta^2 = \gamma_n = o(1/\alpha^2 \cdot s \log d/n)$, we have $\beta^2 = o(1/\alpha^2 \cdot \log d/n)$. Hence, for any small constant $C > 0$, we have $\beta^2 \leq C \cdot 1/\alpha^2 \cdot \log d/n$ when $n$ is sufficiently large. Note that we assume $s = o(d^{1/2-\delta})$ for some fixed constant $\delta > 0$. Then we have

$$
\begin{aligned}
T_1 &\leq \sum_{k=1}^s \binom{s}{k} \cdot (s/d)^k \cdot \exp(\alpha^2 \beta^2 nk) \leq \sum_{k=1}^s \left[ s^2 e/(kd) \right]^k \cdot \exp(\alpha^2 \beta^2 nk) \\
&\leq \sum_{k=1}^s \left[ s^2 e/(kd) \right]^k \cdot \exp(Ck \log d) = \sum_{k=1}^s (s^2 e/k \cdot d^{C-1})^k \leq \sum_{k=1}^s (e/k \cdot d^{C-2\delta})^k,
\end{aligned}
$$

where the second step follows from the fact that $\binom{s}{k} \leq (es/k)^k$. Note that $C$ is chosen arbitrarily, hence we can always choose $C \leq \delta$. It implies that $e/k \cdot d^{C-2\delta} = o(1)$. We thus conclude $T_1 = o(1)$.

**Bounding $T_2$.** For term $T_2$, we observe that

$$T_2 \leq \sum_{k=1}^{s} (e/k \cdot s^2/d)^k \cdot \mathbb{E}_U \left\{ \left[ \exp(\beta^2 |U|) - 1 \right]^k \right\}$$

$$\leq \sum_{k=1}^{s} (e/k \cdot s^2/d)^k \cdot \mathbb{E}_U \left[ (\beta^2 |U|)^k + \exp(\beta^2 k |U|) \cdot \mathbb{1}(\beta^2 |U| \geq 1) \right]$$

$$\leq T_3 + T_4,$$

where $T_3$ and $T_4$ are defined as

$$T_3 := \sum_{k=1}^{s} \mathbb{E}_U (e/k \cdot s^2 \beta^2 / d \cdot |U|)^k,$$

$$T_4 := \sum_{k=1}^{s} (e/k \cdot s^2/d)^k \cdot \mathbb{E}_U \left[ \exp(\beta^2 k |U|) \cdot \mathbb{1}(\beta^2 |U| \geq 1) \right].$$

Note that $U$ is summation of $n$ i.i.d. centered sub-Gaussian random variables $U_i$ each with Orlicz $\psi_2$-norm equal to one. Therefore, $U$ is also centered sub-Gaussian random variable with $||U||_{\psi_2} \leq C\sqrt{n}$ for some constant $C$. Thus it holds that

$$\mathbb{E}(|U|^k) \leq (\sqrt{k} \cdot ||U||_{\psi_2})^k \leq (C\sqrt{nk})^k.$$

Hence for term $T_3$, we have

$$T_3 \leq \sum_{k=1}^{s} \left[ Ces^2 \beta^2 \sqrt{n}/(\sqrt{k}d) \right]^k,$$

Under the condition $s\beta^2 = o(\sqrt{s \log d/n})$, we have

$$Ces^2 \beta^2 \sqrt{n}/(\sqrt{k}d) = o\left( s\sqrt{s \log d}/d \right).$$

Since $s = o(\sqrt{d})$, we have $s\sqrt{s \log d}/d = o(1)$, which implies $T_3 = o(1)$.

To obtain an upper bound for term $T_4$, we let $W = \beta^2 U$. So $W$ is centered sub-Gaussian with Orlicz norm $c\beta^2 \sqrt{n}$. Computing integral by parts, we have

$$\mathbb{E}_U \left[ \exp(\beta^2 k |U|) \cdot \mathbb{1}(\beta^2 |U| \geq 1) \right] = e^k \cdot \mathbb{P}(|W| \geq 1) + \int_{w=1}^{\infty} k e^{wk} \cdot \mathbb{P}(|W| \geq w) \mathrm{d}w. \quad \text{(A.8)}$$

Using the property of sub-Gaussianity, we have $\mathbb{P}[W \geq t] \leq C_1 \exp[-C_2 t^2/(\beta^2 \sqrt{n})^2]$ for some absolute constants $C_1, C_2 > 0$. Proceeding with (A.8) and using shorthand $\sigma = \beta^2 \sqrt{n}$, we obtain

$$\mathbb{E}_U \left[ \exp(\beta^2 k |U|) \cdot \mathbb{1}(\beta^2 |U| \geq 1) \right] \leq C_1 e^k e^{-C_2/\sigma^2} + C_1 k \int_{w=1}^{\infty} e^{wk} e^{-C_2 w^2/\sigma^2} \mathrm{d}w$$

$$= C_1 e^k e^{-C_2/\sigma^2} + C_1 k e^{k^2 \sigma^2/(4C_2)} \int_{w=1}^{\infty} e^{-\frac{C_2}{\sigma^2}(w - \frac{k\sigma^2}{2C_2})^2} \mathrm{d}w \leq C_1 e^k + C_3 k e^{k^2 \sigma^2/(4C_2)} \sigma,$$

where $C_3$ is a constant that depends on $C_1$ and $C_2$. Thus we have

$$T_4 \leq \underbrace{\sum_{k=1}^{s} C_1 \left[ s^2 e^2/(kd) \right]^k}_{T_5} + \underbrace{\sum_{k=1}^{s} C_3 \sigma k \left[ s^2 e^2/(kd) \cdot \exp(k/4 \cdot \sigma^2/C_2) \right]^k}_{T_6}. \quad \text{(A.9)}$$

Note that $s^2/d = o(1)$, we thus have $T_5 = o(1)$. Under condition $s\beta^2 = o(\sqrt{s \log d/n})$, for any small constant $C > 0$, when $n$ is large enough, we have

$$\exp(k/4 \cdot \sigma^2/C_2) \leq \exp(Ck \log d/s) \leq \exp(C \log d) \leq d^C.$$

Plugging (A.9) into $T_6$ and using $s^2 = o(d^{1-2\delta})$, we have that each term in the summation is less that

$$T_6 \leq \sum_{k=1}^{s} \sigma k \left[ e^2/(kd^{2\delta-C}) \right]^k \lesssim \sum_{k=1}^{s} k\sqrt{\log d/s} \cdot \left[ e^2/(d^{2\delta-C}) \right]^k.$$

Since the constant $C$ is chosen arbitrarily, we have $T_6 = o(1)$. Accordingly, $T_4 = o(1)$ and $T_2 = o(1)$.

Finally, combining everything together, we have $D_{\chi^2}(\mathbb{P}^n_{\mathcal{H}}, \mathbb{P}^n_{\mathbf{0}}) = o(1)$, which completes the proof.

## A.2   Proof of Theorem 3.2

We begin with some basic properties of sample sets $\{\mathbf{w}_i\}_{i=1}^n$ and $\{\mathbf{u}_i\}_{i=1}^n$. We introduce the random vector $\mathbf{W} := \mathbf{X} - \mathbf{X}'$ to capture the distribution of samples $\{\mathbf{w}_i\}_{i=1}^n$. Here $\mathbf{X}$ follows the model given in (2.1)-(2.2), and $\mathbf{X}'$ is an independent copy of $\mathbf{X}$. We note that the marginal distribution of $\mathbf{X}$ is given by $1/2 \cdot \mathcal{N}(\boldsymbol{\mu}_0, \boldsymbol{\Sigma}) + 1/2 \cdot \mathcal{N}(\boldsymbol{\mu}_1, \boldsymbol{\Sigma})$. Thus $\mathbf{W}$ follows a mixture distribution

$$\mathbf{W} \sim 1/2 \cdot \mathcal{N}(\mathbf{0}, 2\boldsymbol{\Sigma}) + 1/4 \cdot \mathcal{N}(\boldsymbol{\mu}_1 - \boldsymbol{\mu}_0, 2\boldsymbol{\Sigma}) + 1/4 \cdot \mathcal{N}(\boldsymbol{\mu}_0 - \boldsymbol{\mu}_1, 2\boldsymbol{\Sigma}). \tag{A.10}$$

Moreover, conditioning on the observed label $Y$, the distribution of $\mathbf{X}$ is given by

$$\mathbf{X}|Y = 0 \ \sim \ (1+\alpha)/2 \cdot \mathcal{N}(\boldsymbol{\mu}_0, \boldsymbol{\Sigma}) + (1-\alpha)/2 \cdot \mathcal{N}(\boldsymbol{\mu}_1, \boldsymbol{\Sigma}), \tag{A.11}$$

$$\mathbf{X}|Y = 1 \ \sim \ (1+\alpha)/2 \cdot \mathcal{N}(\boldsymbol{\mu}_1, \boldsymbol{\Sigma}) + (1-\alpha)/2 \cdot \mathcal{N}(\boldsymbol{\mu}_0, \boldsymbol{\Sigma}). \tag{A.12}$$

We introduce a random vector $\mathbf{U} := \mathbf{X}^{(1)} - \mathbf{X}^{(0)}$ that corresponds to samples $\{\mathbf{u}_i\}_{i=1}^n$. Here random vectors $\mathbf{X}^{(0)}$ and $\mathbf{X}^{(1)}$ are independent and have distributions given in (A.11), (A.12), respectively. The distribution of $\mathbf{U}$ is given by

$$\mathbf{U} \sim (1+\alpha)^2/4 \cdot \mathcal{N}(\boldsymbol{\mu}_1 - \boldsymbol{\mu}_0, 2\boldsymbol{\Sigma}) + (1-\alpha^2)/2 \cdot \mathcal{N}(\mathbf{0}, 2\boldsymbol{\Sigma}) + (1-\alpha)^2/4 \cdot \mathcal{N}(\boldsymbol{\mu}_0 - \boldsymbol{\mu}_1, 2\boldsymbol{\Sigma}). \tag{A.13}$$

Now we turn to prove Theorem 3.2. It suffices to prove this result by bounding type-I and type-II errors separately. In the end, we will show that

$$\sup_{\boldsymbol{\theta} \in \mathcal{G}_0(\boldsymbol{\Sigma})} \mathbb{P}^n_{\boldsymbol{\theta}}(\phi = 1) \leq 4d^{-1} \ \text{ and } \ \sup_{\boldsymbol{\theta} \in \mathcal{G}_1(\boldsymbol{\Sigma}; \gamma_n)} \mathbb{P}^n_{\boldsymbol{\theta}}(\phi = 0) \leq 16d^{-1}.$$

**Type-I error.**   Under the null hypothesis $\boldsymbol{\theta} \in \mathcal{G}_0(\boldsymbol{\Sigma})$, (A.10) and (A.13) reduce to

$$\mathbf{W} \sim \mathcal{N}(\mathbf{0}, 2\boldsymbol{\Sigma}), \ \ \mathbf{U} \sim \mathcal{N}(\mathbf{0}, 2\boldsymbol{\Sigma}).$$

To bound the type-I error of function $\phi_1$, we first note that

$$\frac{1}{n} \sum_{i=1}^n (\mathbf{v}^\top \boldsymbol{\Sigma}^{-1} \mathbf{w}_i)^2 = \mathbf{v}^\top \widehat{\boldsymbol{\Sigma}}_W \mathbf{v},$$

where we let $\widehat{\boldsymbol{\Sigma}}_W := 1/n \cdot \sum_{i=1}^n \boldsymbol{\Sigma}^{-1} \mathbf{w}_i \mathbf{w}_i^\top \boldsymbol{\Sigma}^{-1}$, i.e., an empirical covariance matrix of random vector $\boldsymbol{\Sigma}^{-1} \mathbf{W} \sim \mathcal{N}(\mathbf{0}, 2\boldsymbol{\Sigma}^{-1})$. For any matrix $\mathbf{A} \in \mathbb{R}^{d \times d}$ and $\mathcal{S} \subseteq [d]$, we let $[\mathbf{A}]_{\mathcal{S}} \in \mathbb{R}^{|\mathcal{S}| \times |\mathcal{S}|}$ be the submatrix of $\mathbf{A}$, which contains the entries with row and column indices in $\mathcal{S}$. By standard tail bound of Gaussian covariance estimation (see Lemma C.2), for any fixed $\mathcal{S} \in [d]$ with $|\mathcal{S}| = s$, and any $\epsilon \in (0, 1)$, when $n \geq Cs/\epsilon^2$ for some constant $C$, we have

$$\mathbb{P}^n_{\boldsymbol{\theta}} \left[ \|(\widehat{\boldsymbol{\Sigma}}_W - 2\boldsymbol{\Sigma}^{-1})_{\mathcal{S}}\|_2 \geq 2\epsilon \|(\boldsymbol{\Sigma}^{-1})_{\mathcal{S}}\|_2 \right] \leq 2e^{-n}. \tag{A.14}$$

Note that $\|(\boldsymbol{\Sigma}^{-1})_{\mathcal{S}}\|_2 \leq \|\boldsymbol{\Sigma}^{-1}\|_2$ for all $\mathcal{S} \subseteq [d]$. By taking union bound over all subsets with size $s$ in $[d]$, we have

$$\mathbb{P}^n_{\boldsymbol{\theta}} \left[ \sup_{\mathcal{S} \in [d], |\mathcal{S}|=s} \|(\widehat{\boldsymbol{\Sigma}}_W - 2\boldsymbol{\Sigma}^{-1})_{\mathcal{S}}\|_2 \geq 2\epsilon \|\boldsymbol{\Sigma}^{-1}\|_2 \right] \leq 2\binom{d}{s} e^{-n}$$

$$\overset{(a)}{\leq} 2 \exp\left[-n + s \log(ed/s)\right] \overset{(b)}{\leq} 2[s/(ed)]^s \leq 2d^{-1}.$$

Here step $(a)$ follows from the fact that $\binom{d}{s} \leq (ed/s)^s$ and step $(b)$ follows from the assumption that $n \geq 2s \log(ed/s)$. In the last step we use the fact that function $f(s) = (s/d)^s$ is monotonically decreasing for $s \in [1, d/e]$. We set $\epsilon = \sqrt{s \log(ed/s)/n}$. Under condition $n \geq 2s \log(ed/s)$, we have $\epsilon < 1$. Moreover, when $s \leq C'd$ for sufficiently small constant $C'$ that depends on $C$, we have $n \geq Cs/\epsilon^2$. Therefore, such value of $\epsilon$ leads to (A.14). Thus we conclude that

$$\mathbb{P}^n_{\theta} \left[ \frac{\mathbf{v}^\top \widehat{\boldsymbol{\Sigma}}_W \mathbf{v} - 2\mathbf{v}^\top \boldsymbol{\Sigma}^{-1} \mathbf{v}}{2\mathbf{v}^\top \boldsymbol{\Sigma}^{-1} \mathbf{v}} \geq \sqrt{\frac{s \log(ed/s)}{n}} \cdot \frac{\|\boldsymbol{\Sigma}^{-1}\|_2}{\mathbf{v}^\top \boldsymbol{\Sigma}^{-1} \mathbf{v}}, \text{for all } \mathbf{v} \in \mathcal{B}_2(s) \right] \leq 2d^{-1}$$

Note that $\|\mathbf{\Sigma}^{-1}\|_2/(\mathbf{v}^\top \mathbf{\Sigma}^{-1}\mathbf{v}) \le \|\mathbf{\Sigma}^{-1}\|_2 \|\mathbf{\Sigma}\|_2 = \kappa$. Our choice of $\tau_1$ ensures the type-I error of $\phi_1$ does not exceed $2d^{-1}$.

Now we turn to analyze the performance of $\phi_2$. Recall that $\phi_1$ simply selects the coordinate of $\bar{\mathbf{u}} := 1/n \cdot \sum_{i=1}^n \mathbf{u}_i$ that has the largest magnitude (scaled with $\mathrm{diag}(\mathbf{\Sigma})^{-1/2}$) and compare it with $\tau_2$. It suffices to show all coordinates are well bounded around $0$ under null hypothesis. Denote the $j$-th coordinate of $\bar{\mathbf{u}}$ by $\bar{u}_j$. Denote the $j$-th diagonal term of $\mathbf{\Sigma}$ by $\sigma_j$. We have $\bar{u}_j \sim \mathcal{N}(0, 2\sigma_j/n)$. Recall that for standard normal random variable $X$, we have

$$\mathbb{P}(|X| \ge t) \le 2\exp(-t^2/2) \ \text{ for any } \ t \ge 1. \tag{A.15}$$

Using this property and taking union bound over $j \in [d]$, we have

$$\mathbb{P}_{\boldsymbol{\theta}}^n \left( \sup_{j \in [d]} |\bar{u}_j|/\sqrt{\sigma_j} \ge 8\log d/n \right) \le 2d \cdot \exp(-2\log d) = 2d^{-1}.$$

Accordingly, our choice of $\tau_2$ can ensure type-I error of $\phi_2$ is controlled within $2d^{-1}$.

**Type-II error.** Under the alternative hypothesis $\boldsymbol{\theta} \in \mathcal{G}_1(\mathbf{\Sigma}; \gamma_n)$. Note that $\phi = 0$ if and only if $\phi_1 = 0$ and $\phi_2 = 0$. Thus, for any $\boldsymbol{\theta} \in \mathcal{G}_1(\mathbf{\Sigma}; \gamma_n)$, we have

$$\mathbb{P}_{\boldsymbol{\theta}}^n(\phi = 0) = \mathbb{P}_{\boldsymbol{\theta}}^n(\phi_1 = 0 \cap \phi_2 = 0) \le \mathbb{P}_{\boldsymbol{\theta}}^n(\phi_1 = 0) \wedge \mathbb{P}_{\boldsymbol{\theta}}^n(\phi_2 = 0). \tag{A.16}$$

We assume $\gamma_n \ge C\kappa[\sqrt{s\log d/n} \vee (1/\alpha^2 \cdot s\log d/n)]$. It suffices to bound the type-II error by considering these two cases: (i) when $\gamma_n \gtrsim \kappa\sqrt{s\log d/n}$, we show that $\mathbb{P}_{\boldsymbol{\theta}}^n(\phi_1 = 0) \le 16d^{-1}$; (ii) when $\gamma_n \gtrsim \kappa/\alpha^2 \cdot s\log d/n$ and $16/\alpha^2 \cdot s\log d/n \le \sqrt{s\log d/n}$, we show $\mathbb{P}_{\boldsymbol{\theta}}^n[\phi_2 = 0] \le 7d^{-1}$.

**Case (i).** Now we consider the first case. We denote $\Delta\boldsymbol{\mu} := \boldsymbol{\mu}_1 - \boldsymbol{\mu}_0$. Let $\mathbf{v}^* := \Delta\boldsymbol{\mu}/\|\Delta\boldsymbol{\mu}\|_2$. Since $\mathbf{v}^* \in \mathcal{B}_2(s)$, we have

$$\sup_{\mathbf{v} \in \mathcal{B}_2(s)} \frac{\mathbf{v}^\top \widehat{\mathbf{\Sigma}}_W \mathbf{v}}{2\mathbf{v}^\top \mathbf{\Sigma}^{-1}\mathbf{v}} \ge \frac{\mathbf{v}^{*\top}\widehat{\mathbf{\Sigma}}_W \mathbf{v}^*}{2\mathbf{v}^{*\top}\mathbf{\Sigma}^{-1}\mathbf{v}^*}.$$

It remains to show the right hand side is larger than $1 + \tau_1$ with high probability. Note that

$$\mathbf{v}^{*\top}\widehat{\mathbf{\Sigma}}_W \mathbf{v}^* = \frac{1}{n}\sum_{i=1}^n (\mathbf{v}^{*\top}\mathbf{\Sigma}^{-1}\mathbf{w}_i)^2.$$

We define a random variable $\widetilde{W} := \mathbf{v}^{*\top}\mathbf{\Sigma}^{-1}\boldsymbol{W}$, whose probability distribution is given by

$$1/2 \cdot \mathcal{N}(0, \nu) + 1/4 \cdot \mathcal{N}(m, \nu) + 1/4 \cdot \mathcal{N}(-m, \nu), \tag{A.17}$$

where we define $m := \rho(\boldsymbol{\theta})/\|\Delta\boldsymbol{\mu}\|_2$ and $\nu := 2\rho(\boldsymbol{\theta})/\|\Delta\boldsymbol{\mu}\|_2^2$. Recall that $\rho(\boldsymbol{\theta}) := \Delta\boldsymbol{\mu}^\top \mathbf{\Sigma}^{-1}\Delta\boldsymbol{\mu}$. Let $\widetilde{w}_i := \mathbf{v}^{*\top}\mathbf{\Sigma}^{-1}\mathbf{w}_i$. Due to the mixture structure (A.17), we can thus cluster $\{\widetilde{w}_i\}_{i=1}^n$ into three groups $\{\widetilde{w}_i^{(k)}\}_{i=1}^{n_k}, k \in \{1,2,3\}$, based on the latent labels. The $k$-th group corresponds to the $k$-th term in (A.17). Note that $\mathbb{E}(n_1) = n/2, \mathbb{E}(n_2) = \mathbb{E}(n_3) = n/4$. Define event $\mathcal{E}_1$ as

$$\mathcal{E}_1 := \{|n_1 - n/2| \le 1/8 \cdot n, \ |n_2 - n/4| \le 1/8 \cdot n, \ |n_3 - n/4| \le 1/8 \cdot n\}. \tag{A.18}$$

By Hoeffding's inequality, we have $\mathbb{P}(\mathcal{E}_1) \ge 1 - 6\exp(-n^2/32)$.

From now on, we condition on event $\mathcal{E}_1$. By the standard $\chi^2$-tail bound (Lemma C.1), for any $t \in (0,1)$ and $k \in \{1,2,3\}$, we have

$$\mathbb{P}_{\boldsymbol{\theta}}^n \left( \left| \sum_{i=1}^{n_k} (\widetilde{w}_i^{(k)} - m_k)^2 - n_k\nu \right| \ge n_k\nu t \right) \le 2e^{-n_k t^2/8} \le 2e^{-nt^2/64}, \tag{A.19}$$

where $m_1 = 0, m_2 = -m_3 = m$. Moreover, using tail bound of Gaussian (A.15), for $t' \ge 1/\sqrt{n_k}$ and $k = 2, 3$,

$$\mathbb{P}_{\boldsymbol{\theta}}^n \left( \left| \sum_{i=1}^{n_k} \widetilde{w}_i^{(k)} - n_k m_k \right| \ge n_k\sqrt{\nu}t' \right) \le 2e^{-n_k t'^2/2} \le 2e^{-nt'^2/16}. \tag{A.20}$$

Excluding the small chance events in (A.19) and (A.20), we find that

$$\sum_{i=1}^{n} \widetilde{w}_i^2 = \sum_{k=1}^{3} \sum_{i=1}^{n_k} (\widetilde{w}_i^{(k)} - m_k)^2 + 2 \sum_{k=2}^{3} \sum_{i=1}^{n_k} m_k \widetilde{w}_i^{(k)} - (n_2 + n_3)m^2$$

$$\geq n\nu(1-t) + 2 \sum_{k=2}^{3} \sum_{i=1}^{n_k} m_k \widetilde{w}_i^{(k)} - (n_2 + n_3)m^2$$

$$\geq n\nu(1-t) + (n_2 + n_3)m^2 - 2(n_2 + n_3)\sqrt{\nu}t'm$$

$$\geq n\nu(1-t) + 1/4 \cdot nm^2 - 3/2 \cdot n\sqrt{\nu}t'm,$$

where the last step follows from (A.18). Note that $2\mathbf{v}^{*\top}\boldsymbol{\Sigma}^{-1}\mathbf{v}^* = \nu$. We thus have

$$\frac{\mathbf{v}^{*\top}\widehat{\boldsymbol{\Sigma}}_W\mathbf{v}^*}{2\mathbf{v}^{*\top}\boldsymbol{\Sigma}^{-1}\mathbf{v}^*} - 1 = \frac{\sum_{i=1}^{n}\widetilde{w}_i^2}{2n\mathbf{v}^{*\top}\boldsymbol{\Sigma}^{-1}\mathbf{v}^*} - 1 \geq \frac{m^2}{4\nu} - t - \frac{3mt'}{2\sqrt{\nu}}$$

$$= 1/8 \cdot \rho(\boldsymbol{\theta}) - t - 3t'/4 \cdot \sqrt{2\rho(\boldsymbol{\theta})}. \tag{A.21}$$

Now we choose $t = t' = 8\sqrt{s\log(ed/s)/n}$, which is less than one under condition $n \geq 64s\log(ed/s)$. When $\rho(\boldsymbol{\theta}) \geq C\kappa\sqrt{s\log(ed/s)/n}$ for sufficiently large constant $C$, we can have $t \leq \rho(\boldsymbol{\theta})/32$ and $t' \leq \sqrt{t'} \leq \sqrt{\rho(\boldsymbol{\theta})}/48$. Accordingly, proceeding with (A.21) gives

$$1/2 \cdot \mathbf{v}^{*\top}\widehat{\boldsymbol{\Sigma}}_W\mathbf{v}^*/\mathbf{v}^{*\top}\boldsymbol{\Sigma}^{-1}\mathbf{v}^* - 1 \geq 1/16 \cdot \rho(\boldsymbol{\theta}) \geq \tau_1.$$

Plugging the value of $t, t'$ into the tail bounds in (A.19) (A.20) and using the probability of event $\mathcal{E}_1$, we have the type-II error of $\phi_1$ is most $10d^{-1} + 6e^{-n^2/32} \leq 16d^{-1}$.

**Case (ii).** Now we turn to analyze the performance of $\phi_2$. We introduce shorthands $\widetilde{\boldsymbol{\mu}} := \mathrm{diag}(\boldsymbol{\Sigma})^{-1/2}\Delta\boldsymbol{\mu}$ and $\boldsymbol{\Lambda} := \mathrm{diag}(\boldsymbol{\Sigma})^{1/2}$. Then it holds that

$$\rho(\boldsymbol{\theta}) = \Delta\boldsymbol{\mu}^{\top}\boldsymbol{\Sigma}^{-1}\Delta\boldsymbol{\mu} = \Delta\boldsymbol{\mu}^{\top}\boldsymbol{\Lambda}^{-1}\boldsymbol{\Lambda}\boldsymbol{\Sigma}^{-1}\boldsymbol{\Lambda}\boldsymbol{\Lambda}^{-1}\Delta\boldsymbol{\mu} \leq \|\widetilde{\boldsymbol{\mu}}\|_2^2 \|\boldsymbol{\Lambda}\boldsymbol{\Sigma}^{-1}\boldsymbol{\Lambda}\|_{op}$$

$$\leq \|\widetilde{\boldsymbol{\mu}}\|_2^2 \|\boldsymbol{\Lambda}\|_2^2 \|\boldsymbol{\Sigma}^{-1}\|_2 \leq \kappa\|\widetilde{\boldsymbol{\mu}}\|_2^2,$$

where the last step follows from the fact that $\|\mathrm{diag}(\boldsymbol{\Sigma})\|_2 \leq \|\boldsymbol{\Sigma}\|_2$. Suppose the $j$-th coordinate of $\widetilde{\boldsymbol{\mu}}$, denoted by $\beta$, has largest magnitude. Since $\|\widetilde{\mathbf{u}}\|_2^2 \leq s\beta^2$, we have $\beta^2 \geq \rho(\boldsymbol{\theta})/(s\kappa)$. Under condition

$$\rho(\boldsymbol{\theta}) \geq \gamma_n \geq \frac{400\kappa s\log d}{\alpha^2 n},$$

we have

$$\beta \geq 20\sqrt{\log d/(\alpha^2 n)}. \tag{A.22}$$

Let $\mathbf{v}^* = \mathrm{sign}(\beta) \cdot \mathbf{e}_j$. We have

$$\sup_{\mathbf{v} \in \mathcal{B}_2(1)} \langle \mathbf{v}, \boldsymbol{\Lambda}^{-1}\bar{\mathbf{u}} \rangle \geq \langle \mathbf{v}^*, \boldsymbol{\Lambda}^{-1}\bar{\mathbf{u}} \rangle = \left| \frac{1}{n}\sum_{i=1}^{n} \widetilde{u}_{ij} \right|,$$

where we denote the $j$-th coordinate of $\boldsymbol{\Lambda}^{-1}\mathbf{u}_i$ by $\widetilde{u}_{ij}$.

Let $U_j$ be the $j$-th coordinate of $\boldsymbol{U}$. Note that $\{\widetilde{u}_{ij}\}_{i=1}^{n}$ are i.i.d. samples of $U_j/\sqrt{\sigma_j}$. Recall that $\sigma_j$ is the $j$-th diagonal term of $\boldsymbol{\Sigma}$. According to (A.13), $U_j/\sqrt{\sigma_j}$ has the mixture distribution

$$(1+\alpha)^2/4 \cdot \mathcal{N}(\beta, 2) + (1-\alpha^2)/2 \cdot \mathcal{N}(0, 2) + (1-\alpha)^2/4 \cdot \mathcal{N}(-\beta, 2). \tag{A.23}$$

We can cluster these samples into three groups $\{\widetilde{u}_{ij}^{(k)}\}_{i=1}^{n_k}, k \in \{1, 2, 3\}$ based on latent labels, where $k$-th group corresponds to the $k$-th term in (A.23). Using tail bound of Gaussian (A.15), we have for $t \geq 1$ and $k \in \{1, 2, 3\}$,

$$\mathbb{P}_{\boldsymbol{\theta}}^n\left( \left| \sum_{i=1}^{n_k} \widetilde{u}_{ij}^{(k)} - n_k m_k \right| \geq \sqrt{2n_k}t \right) \leq 2e^{-t^2/2},$$

where $m_1 = -m_3 = \beta, m_2 = 0$. Therefore, with probability at least $1 - 6e^{-t^2/2}$, it holds that

$$\left| \frac{1}{n}\sum_{i=1}^{n} \widetilde{u}_{ij} - \frac{(n_1 - n_3)\beta}{n} \right| \leq t \cdot \sum_{k=1}^{3} \sqrt{\frac{2n_k}{n^2}} \leq \frac{5t}{\sqrt{n}}. \tag{A.24}$$

It remains to bound $n_1 - n_3$. Note that $n_1 - n_3$ is a summation of $n$ i.i.d. random variables $V_i$ satisfying $\mathbb{P}(V_i = 1) = (1+\alpha)^2/4$, $\mathbb{P}(V_i = 0) = (1-\alpha^2)/2$, and $\mathbb{P}(V_i = -1) = (1-\alpha)^2/4$. Then $V_i$ has mean $\alpha$, variance $(1 - \alpha^2)/2 \leq 1 - \alpha$, and $|V_i - \mathbb{E}(V_i)| \leq 2$. By Bernstein's inequality, we have that for $t' > 0$,

$$\mathbb{P}\left(|n_1 - n_3 - \alpha n| \geq t'\right) \leq \exp\left[-\frac{t'^2}{2(1-\alpha)n + 4t'/3}\right].$$

Choosing $t' = \alpha n/2$, we thus have

$$\mathbb{P}\left(|n_1 - n_3 - \alpha \cdot n| \geq \alpha n/2\right) \leq \exp\left[-\frac{\alpha^2 n}{8(1-\alpha) + 8\alpha/3}\right] \leq \exp(-\alpha^2 n/8) \leq d^{-1}, \quad \text{(A.25)}$$

where the last step follows from condition $8s \log d/(\alpha^2 n) \leq \sqrt{s \log(ed/s)/n} \leq 1$. Combining (A.24) and (A.25), we have that with high probability $1 - 6e^{-t^2/2} - d^{-1}$,

$$\left|1/n \cdot \sum_{i=1}^{n} \widetilde{u}_{ij}\right| \geq \alpha\beta/2 - 5t/\sqrt{n} \geq 10\sqrt{\log d/n} - 5t/\sqrt{n} \geq \tau_2,$$

where the second step follows from (A.22) and the last inequality holds by setting $t = \sqrt{2\log d}$, which gives the type-II error of $\phi_2$ is at most $7d^{-1}$.

Using (A.16) and the conclusions in the above two cases, we thus show Type-II error of $\phi$ is at most $16d^{-1}$ and thus complete the proof.

### A.3 Proof of Theorem 3.3

In this section, we prove the computational lower bound. We first show that the information-theoretic lower bound in (3.4) is a lower bound of the computationally tractable minimax rate. To see this, we consider the oracle $r^*$ that returns sample average $n^{-1}\sum_{i=1}^{n} q(y_i, \mathbf{x}_i)$ for any query function $q$. As discussed in §2.2, Bernstein's inequality in (2.6) and uniform concentration of empirical process imply that $r^* \in \mathcal{R}[\xi, n, T_n, \eta(\mathcal{Q}_{\mathscr{A}})]$. In addition, every test function $\phi$ that is based on the responses of $r^*$ is also a function of $\{(y_i, \mathbf{x}_i)\}_{i=1}^{n}$. Thus combining (2.4) and (2.7), it holds that

$$\overline{R}_n^*(\mathcal{G}_0, \mathcal{G}_1; \mathscr{A}, r^*) \geq R_n^*(\mathcal{G}_0, \mathcal{G}_1).$$

Therefore, by Theorem 3.1, for any $\gamma_n$ satisfying

$$\gamma_n = o\left[\sqrt{s \log d/n} \wedge (1/\alpha^2 \cdot s \log d/n)\right],$$

we have $\lim_{n \to \infty} \overline{R}_n^*[\mathcal{G}_0, \mathcal{G}_1(\gamma_n); \mathscr{A}, r^*] = 1$. Here the equality holds because a test based on purely random guess incurs risk one.

Based on this observation, to show Theorem 3.3, it the following, we assume that

$$\gamma_n = o\left[\sqrt{s^2/n} \wedge (1/\alpha^2 \cdot s/n)\right]. \tag{A.26}$$

We show that under this assumption, there exists an oracle $r$ such that the minimax testing risk is not negligible. Similar to the derivation of the information theoretical lower bound, we also focus on the restricted testing problem defined in (A.1). Following the same notations, we denote by $\mathbb{P}$ the distribution of model $(\mathbf{0}, \mathbf{0}, \mathbf{I}, \alpha)$ and by $\mathbb{P}$ the distribution of model $(-\mathbf{v}/2, \mathbf{v}/2, \mathbf{I}, \alpha)$ for all $\mathbf{v} \in \mathcal{H}(s) = \{\mathbf{u} \in \{0, \beta\}^d : \|\mathbf{u}\|_0 = s\}$. Here we assume that the SNR under $H_1$ satisfies $\beta^2 s = \gamma_n$. Moreover, we define $\overline{\mathbb{P}}$ as the distribution of the random variables returned by the statistical query model under the null hypothesis $H_0$ and define $\overline{\mathbb{P}}$ correspondingly. Then the minimax testing risk $\overline{R}_n^*(\mathcal{G}_0, \mathcal{G}_1; \mathscr{A}, r)$ defined in (2.7) is lower bounded by

$$\sup_{\boldsymbol{\Sigma}} \overline{R}_n^*[\mathcal{G}_0(\boldsymbol{\Sigma}), \mathcal{G}_1(\boldsymbol{\Sigma}; \gamma_n); \mathscr{A}, r] \geq \inf_{\phi \in \mathcal{H}(\mathscr{A}, r)} \left[\overline{\mathbb{P}}\,(\phi = 1) + \frac{1}{|\mathcal{H}(s)|} \sum_{\in \mathcal{H}(s)} \overline{\mathbb{P}}\,(\phi = 0)\right].$$

The following lemma establishes a sufficient condition that any hypothesis test under the statistical query model is asymptotically powerless. See [27] and [8] for a proof.

**Lemma A.3.** *For any algorithm* $\mathscr{A} \in \mathcal{A}(T)$ *and any query function* $q \in \mathcal{Q}_{\mathscr{A}}$, *we define*

$$\mathcal{C}_1(q) = \left\{ \mathbf{v} \in \mathcal{H}(s) : \mathbb{E}_{\mathbb{P}_{\mathbf{v}}}[q(Y, \boldsymbol{X})] - \mathbb{E}_{\mathbb{P}}[q(Y, \boldsymbol{X})] > \tau_q(\mathbb{P}) \right\},$$

$$\mathcal{C}_2(q) = \left\{ \mathbf{v} \in \mathcal{H}(s) : \mathbb{E}_{\mathbb{P}}[q(Y, \boldsymbol{X})] - \mathbb{E}_{\mathbb{P}_{\mathbf{v}}}[q(Y, \boldsymbol{X})] > \tau_q(\mathbb{P}) \right\}.$$

*Here* $\tau_q(\mathbb{P})$ *is the tolerance parameter defined in* (2.5) *when* $(Y, \boldsymbol{X}) \sim \mathbb{P}$. *Then if* $T \cdot \sup_{q \in \mathcal{Q}_{\mathscr{A}}} (|\mathcal{C}_1(q)| + |\mathcal{C}_2(q)|) / |\mathcal{H}(s)| = o(1)$, *there exists an oracle* $r \in \mathcal{R}[\xi, n, T, \eta(\mathcal{Q}_{\mathscr{A}})]$ *such that*

$$\inf_{\phi \in \mathcal{H}(\mathscr{A}, r)} \left[ \overline{\mathbb{P}}_{\mathbf{0}}(\phi = 1) + \frac{1}{|\mathcal{H}(s)|} \sum_{\mathbf{v} \in \mathcal{H}(s)} \overline{\mathbb{P}}_{\mathbf{v}}(\phi = 0) \right] = 1.$$

By this lemma, we need to construct an upper bound for $\sup_{q \in \mathcal{Q}_{\mathscr{A}}} (|\mathcal{C}_1(q)| + |\mathcal{C}_2(q)|)$. In the sequel, we achieve this goal by studying the uniform mixture of $\{\mathbb{P}_{\mathbf{v}} : \mathbf{v} \in \mathcal{C}_\ell(q)\}$ for $\ell \in \{1, 2\}$. Specifically, we define

$$\mathbb{P}_{\mathcal{C}_1(q)} = \frac{1}{|\mathcal{C}_1(q)|} \sum_{\mathbf{v} \in \mathcal{C}_1(q)} \mathbb{P}_{\mathbf{v}} \quad \text{and} \quad \mathbb{P}_{\mathcal{C}_2(q)} = \frac{1}{|\mathcal{C}_2(q)|} \sum_{\mathbf{v} \in \mathcal{C}_2(q)} \mathbb{P}_{\mathbf{v}}. \tag{A.27}$$

The following lemma, obtained from [8], establishes an upper bound for the $\chi^2$-divergence between $\mathbb{P}_{\mathcal{C}_\ell(q)}$ and $\mathbb{P}$.

**Lemma A.4.** *For* $\ell \in \{1, 2\}$ *we define*

$$\overline{\mathcal{C}}_\ell(q, \mathbf{v}) = \operatorname*{argmax}_{\mathcal{C}} \left\{ \frac{1}{|\mathcal{C}|} \sum_{\mathbf{v}' \in \mathcal{C} \subseteq \mathcal{H}(s)} \mathbb{E}_{\mathbb{P}_{\mathbf{0}}} \left[ \frac{d\mathbb{P}_{\mathbf{v}}}{d\mathbb{P}} \frac{d\mathbb{P}_{\mathbf{v}'}}{d\mathbb{P}}(Y, \boldsymbol{X}) \right] - 1 \,\middle|\, |\mathcal{C}| = |\mathcal{C}_\ell(q)| \right\}. \tag{A.28}$$

*Then the* $\chi^2$*-divergence between* $\mathbb{P}_{\mathcal{C}_\ell(q)}$ *and* $\mathbb{P}$ *is bounded by*

$$D_{\chi^2}(\mathbb{P}_{\mathcal{C}_\ell(q)}, \mathbb{P}) \leq \sup_{\mathbf{v} \in \mathcal{C}_\ell(q)} \frac{1}{|\mathcal{C}_\ell(q)|} \sum_{\mathbf{v}' \in \overline{\mathcal{C}}_\ell(q, \mathbf{v})} \mathbb{E}_{\mathbb{P}_{\mathbf{0}}} \left[ \frac{d\mathbb{P}_{\mathbf{v}}}{d\mathbb{P}} \frac{d\mathbb{P}_{\mathbf{v}'}}{d\mathbb{P}}(Y, \boldsymbol{X}) \right] - 1. \tag{A.29}$$

Notice that Lemma A.1 enables us to compute the right-hand side of (A.29) in closed form. For any $\alpha \in [0, 1]$, function $h_\alpha(t) = \cosh[\beta^2/2 \cdot (s - t)] + \alpha^2 \sinh[\beta^2/2 \cdot (s - t)]$ is monotone nonincreasing for $t \in \{0, \ldots, s\}$ and $f(s) = 0$. In addition, for any $\mathbf{v} \in \mathcal{H}(s)$ and any $j \in \{0, \ldots, s\}$, we define

$$\mathcal{C}_j(\mathbf{v}) = \left\{ \mathbf{v}' \in \mathcal{H}(s) : |\operatorname{supp}(\mathbf{v}) \cap \operatorname{supp}(\mathbf{v}')| = s - j \right\}. \tag{A.30}$$

For $\ell \in \{1, 2\}$, any query function $q \in \mathcal{Q}_{\mathscr{A}}$, and any $\mathbf{v} \in \mathcal{C}_\ell(q)$, by Lemma A.1 and the definition of $\overline{\mathcal{C}}_\ell(q, \mathbf{v})$ in (A.28), there exists an integer $k_\ell(q, \mathbf{v})$ that satisfies

$$\overline{\mathcal{C}}_\ell(q, \mathbf{v}) = \mathcal{C}_0(\mathbf{v}) \cup \mathcal{C}_1(\mathbf{v}) \cup \cdots \cup \mathcal{C}_{k_\ell(q, \mathbf{v})-1}(\mathbf{v}) \cup \mathcal{C}'_\ell(q, \mathbf{v}), \tag{A.31}$$

where $\mathcal{C}'_\ell(q, \mathbf{v}) = \overline{\mathcal{C}}_\ell(q, \mathbf{v}) \setminus \bigcup_{j=0}^{k_\ell(q, \mathbf{v})-1} \mathcal{C}_j(\mathbf{v})$ has cardinality

$$|\mathcal{C}'_\ell(q, \mathbf{v})| = |\mathcal{C}_\ell(q)| - \sum_{j=0}^{k_\ell(q, \mathbf{v})-1} |\mathcal{C}_j(\mathbf{v})| < |\mathcal{C}_{k_\ell(q, \mathbf{v})}(\mathbf{v})|. \tag{A.32}$$

Thus we can sandwich the cardinality of $\overline{\mathcal{C}}_\ell(q, \mathbf{v})$ by

$$\sum_{j=0}^{k_\ell(q, \mathbf{v})} |\mathcal{C}_j(\mathbf{v})| > |\overline{\mathcal{C}}_\ell(q, \mathbf{v})| \geq \sum_{j=0}^{k_\ell(q, \mathbf{v})-1} |\mathcal{C}_j(\mathbf{v})|. \tag{A.33}$$

Combining Lemmas A.1 and A.4, we further have

$$1 + D_{\chi^2}(\mathbb{P}_{\mathcal{C}_\ell(q)}, \mathbb{P}) \leq \frac{\sum_{i=0}^{k_\ell(q, \mathbf{v})-1} h_\alpha(j) \cdot |\mathcal{C}_j(\mathbf{v})| + h_\alpha[k_\ell(q, \mathbf{v})] \cdot |\mathcal{C}'_\ell(q, \mathbf{v})|}{\sum_{j=0}^{k_\ell(q, \mathbf{v})-1} |\mathcal{C}_j(\mathbf{v})| + |\mathcal{C}'_\ell(q, \mathbf{v})|}, \quad \text{for all } \mathbf{v} \in \mathcal{C}_\ell(q). \tag{A.34}$$

Moreover, by (A.34) and the monotonicity of $h_\alpha(t)$ we obtain

$$1 + D_{\chi^2}(\mathbb{P}_{\mathcal{C}_\ell(q)}, \mathbb{P}) \leq \frac{\sum_{i=0}^{k_\ell(q, \mathbf{v})-1} h_\alpha(j) \cdot |\mathcal{C}_j(\mathbf{v})|}{\sum_{j=0}^{k_\ell(q, \mathbf{v})-1} |\mathcal{C}_j(\mathbf{v})|}. \tag{A.35}$$

By the definition of $\mathcal{C}_j(\mathbf{v})$ in (A.30), the cardinality of $\mathcal{C}_j(\mathbf{v})$ does not depend on the choice of $\mathbf{v} \in \mathcal{H}(s)$ and we have $|\mathcal{C}_j(\mathbf{v})| = \binom{s}{s-j}\binom{d-s}{j}$. Thus for any $j \in \{0,\ldots,s-1\}$ we have

$$|\mathcal{C}_{j+1}(\mathbf{v})|/|\mathcal{C}_j(\mathbf{v})| = (s-j)\cdot(d-s-j)/(j+1)^2 \geq (d-2s)/s^2. \tag{A.36}$$

Under the assumption that $s^2/d = o(1)$, the right-hand side of (A.36) is lower bounded by $\zeta = d/(2s^2)$ when $d$ and $s$ are sufficiently large. Then we have $|\mathcal{C}_j(\mathbf{v})| \leq \zeta^{j-s}|\mathcal{C}_s(\mathbf{v})|$ for $j \in \{0,\ldots,s\}$. By the definition of $k_\ell(q,\mathbf{v})$ in (A.31) and (A.32), for any $q \in \mathcal{Q}_{\mathscr{A}}$, we further obtain

$$
\begin{aligned}
|\mathcal{C}_\ell(q)| &\leq \sum_{j=0}^{k_\ell(q,\ )} |\mathcal{C}_j(\mathbf{v})| \leq |\mathcal{C}_s(\mathbf{v})| \sum_{j=0}^{k_\ell(q,\ )} \zeta^{j-s} \\
&\leq \frac{\zeta^{-[s-k_\ell(q,\ )]}|\mathcal{H}(s)|}{1-\zeta^{-1}} \leq 2\zeta^{-[s-k_\ell(q,\ )]}|\mathcal{H}(s)|,
\end{aligned}
\tag{A.37}
$$

where the last inequality follows from the fact that $\zeta^{-1} = 2s^2/d = o(1)$.

Moreover, for any two positive sequences $\{a_i\}_{i=0}^s$ and $\{b_i\}_{i=0}^s$ satisfying $a_i/a_{i-1} \geq b_i/b_{i-1} > 1$ for all $i \in [s]$, since $h_\alpha(t)$ is nonincreasing, for any $k \in [s]$, we have

$$\sum_{0 \leq i < j \leq k} (a_i b_j - a_j b_i) \cdot [h_\alpha(i) - h_\alpha(j)] \leq 0. \tag{A.38}$$

Further simplifying the terms in (A.38), we have

$$\frac{\sum_{i=0}^k [a_i h_\alpha(i)]}{\sum_{i=0}^k a_i} \leq \frac{\sum_{i=0}^k [b_i h_\alpha(i)]}{\sum_{i=0}^k b_i}. \tag{A.39}$$

In what follows, we upper bound $k_\ell(q,\mathbf{v})$ for $\ell \in \{1,2\}$ and $\mathbf{v} \in \mathcal{C}_\ell(q)$. We employ the shorthand $k_\ell = k_\ell(q,\mathbf{v})$ to simplify the notations. Combining (A.29), (A.35), and (A.39) with $a_j = |\mathcal{C}_j(\mathbf{v})|$ and $b_j = \zeta^j$, we have

$$
\begin{aligned}
1 + D_{\chi^2}(\mathbb{P}_{\mathcal{C}_\ell(q)}, \mathbb{P}\ ) &\leq \frac{\sum_{j=0}^{k_\ell-1} \zeta^j h_\alpha(j)}{\sum_{j=0}^{k_\ell-1} \zeta^j} \\
&= \frac{\sum_{j=0}^{k_\ell-1} \zeta^j \left\{ \cosh\left[\beta^2/2 \cdot (s-j)\right] + \alpha^2 \sinh\left[\beta^2/2 \cdot (s-j)\right] \right\}}{\sum_{j=0}^{k_\ell-1} \zeta^j} \\
&\leq \left\{ \frac{\sum_{j=0}^{k_\ell-1} \zeta^j \cosh\left[\beta^2(s-j)\right]}{\sum_{j=0}^{k_\ell-1} \zeta^j} \right\} \bigvee \left\{ \frac{\sum_{j=0}^{k_\ell-1} \zeta^j \exp\left[\alpha^2\beta^2(s-j)\right]}{\sum_{j=0}^{k_\ell-1} \zeta^j} \right\}.
\end{aligned}
\tag{A.40}
$$

Here the second inequality follows from Lemma A.2. We bound the two terms in (A.40) separately. Note that for notational simplicity, we denote for any $t \in \{0,\ldots,s\}$, we define

$$f(t) = \cosh\left[\beta^2(s-t)\right], \quad g(t) = \exp\left[\alpha^2\beta^2(s-t)\right].$$

Note that both $h(t)$ and $g(t)$ are monotone non-increasing, and thus $f(t) \geq f(s) = 1$ and $g(t) \geq g(s) = 1$. Moreover, by calculation, we have $f(j-1)/f(j) \geq \cosh(\beta^2)$ for all $j \in \{1,\ldots,s\}$. Thus we have

$$f(j) \leq f(k_\ell - 1) \cdot \left[\cosh(\beta^2)\right]^{k_\ell-j-1}, \quad \text{for all } j \in \{0,\ldots,k_\ell-1\}.$$

Then we have

$$
\begin{aligned}
\frac{\sum_{j=0}^{k_\ell-1} \zeta^j f(j)}{\sum_{j=0}^{k_\ell-1} \zeta^j} &\leq f(k_\ell - 1) \cdot \frac{\sum_{j=0}^{k_\ell-1} \zeta^j \left[\cosh(\beta^2)\right]^{k_\ell-j+1}}{\sum_{j=0}^{k_\ell-1} \zeta^j} \\
&\leq f(k_\ell - 1) \cdot \frac{\sum_{j=0}^{k_\ell-1} \left[\cosh(\beta^2)/\zeta\right]^{k_\ell-j+1}}{\sum_{j=0}^{k_\ell-1} \zeta^{-(k_\ell-j+1)}} \\
&= f(k_\ell - 1) \cdot \frac{1 - \left[\cosh(\beta^2)/\zeta\right]^{k_\ell}}{1 - \zeta^{-k_\ell}} \cdot \frac{1 - \zeta^{-1}}{1 - \zeta^{-1}\cosh(\beta^2)}.
\end{aligned}
\tag{A.41}
$$

Since $\cosh(\beta^2) > 1$, by (A.41) we have

$$\frac{\sum_{j=0}^{k_\ell - 1} \zeta^j \cosh\left[\beta^2(s-j)\right]}{\sum_{j=0}^{k_\ell - 1} \zeta^j} \leq \frac{1 - \zeta^{-1}}{1 - \zeta^{-1}\cosh(\beta^2)} \cdot \cosh\left[\beta^2(s - k_\ell + 1)\right]. \tag{A.42}$$

In addition, since $g(j-1)/g(j) = \exp(\alpha^2\beta^2)$, similar to (A.41) we have

$$\frac{\sum_{j=0}^{k_\ell - 1} \zeta^j \exp\left[\alpha^2\beta^2(s-j)\right]}{\sum_{j=0}^{k_\ell - 1} \zeta^j} \leq \frac{1 - \zeta^{-1}}{1 - \zeta^{-1}\exp(\alpha^2\beta^2)} \cdot \exp\left[\alpha^2\beta^2(s - k_\ell + 1)\right] \tag{A.43}$$

Combining (A.42) and (A.43), we obtain that

$$1 + D_{\chi^2}(\mathbb{P}_{\mathcal{C}_\ell(q)}, \mathbb{P}\;)$$
$$\leq \left\{\frac{(1 - \zeta^{-1})\cosh\left[\beta^2(s - k_\ell + 1)\right]}{1 - \zeta^{-1}\cosh(\beta^2)}\right\} \bigvee \left\{\frac{(1 - \zeta^{-1})\exp\left[\alpha^2\beta^2(s - k_\ell + 1)\right]}{1 - \zeta^{-1}\exp(\alpha^2\beta^2)}\right\}. \tag{A.44}$$

Moreover, we use the following lemma obtained from [27] to establish a lower bound for $D_{\chi^2}(\mathbb{P}_{\mathcal{C}_\ell(q)}, \mathbb{P}\;)$.

**Lemma A.5.** *For any query function $q$ and $\ell \in \{1, 2\}$, we have*
$$D_{\chi^2}(\mathbb{P}_{\mathcal{C}_\ell(q)}, \mathbb{P}\;) \geq \log(T/\xi)/n.$$

We denote $\sqrt{\log(T/\xi)/n}$ by $\tau$ for simplicity of notations. Combining (A.44), Lemma A.5 and inequality $\cosh(x) \leq \exp(x^2/2)$, at least one of the two inequality holds

$$(1 + \tau^2) \cdot \left[1 - \zeta^{-1}\cosh(\beta^2/2)\right]/(1 - \zeta^{-1}) \leq \exp\left[\beta^4/2 \cdot (s - k_\ell + 1)^2\right], \tag{A.45}$$
$$(1 + \tau^2) \cdot \left[1 - \zeta^{-1}\exp(\alpha^2\beta^2)\right]/(1 - \zeta^{-1}) \leq \exp\left[\alpha^2\beta^2(s - k_\ell + 1)\right]. \tag{A.46}$$

If (A.45) holds, taking the logarithm of the both sides, we have

$$\beta^4/2 \cdot (s - k_\ell + 1)^2 \geq \log(1 + \tau^2) - \log\left[\frac{1 - \zeta^{-1}}{1 - \zeta^{-1}\cosh(\beta^2)}\right]. \tag{A.47}$$

Whereas if (A.46) is true, it holds that

$$\alpha^2\beta^2(s - k_\ell + 1) \geq \log(1 + \tau^2) - \log\left[\frac{1 - \zeta^{-1}}{1 - \zeta^{-1}\exp(\alpha^2\beta^2)}\right]. \tag{A.48}$$

In addition, by Taylor expansion and the fact that $[\cosh(\beta^2/2) + \exp(\alpha^2\beta^2)]/\zeta = o(1)$, we have

$$\log\left[\frac{1 - \zeta^{-1}}{1 - \zeta^{-1}\cosh(\beta^2)}\right] = \log\left\{1 + \frac{\zeta^{-1}\left[\cosh(\beta^2) - 1\right]}{1 - \zeta^{-1}\cosh(\beta^2)}\right\} = O(\zeta^{-1}\beta^4), \tag{A.49}$$

$$\log\left[\frac{1 - \zeta^{-1}}{1 - \zeta^{-1}\exp(\alpha^2\beta^2)}\right] = \log\left\{1 + \frac{\zeta^{-1}\left[\exp(\alpha^2\beta^2) - 1\right]}{1 - \zeta^{-1}\exp(\alpha^2\beta^2))}\right\} = O(\zeta^{-1}\alpha^2\beta^2). \tag{A.50}$$

Since $\gamma_n = s\beta^2$, by (A.26) we have $(\alpha^2\beta^2) \vee \beta^4 = o(\log d/n)$. Hence, by (A.49) and (A.50), the second terms on the right-hand sides of (A.47) and (A.48) are asymptotically negligible compared with $\log(1 + \tau^2)$. Therefore, by (A.47) and (A.48), for $\ell \in \{1, 2\}$, at least one of the following two arguments hold:

$$k_\ell(q, \mathbf{v}) \leq s + 1 - \sqrt{\log(1 + \tau^2)/\beta^4}, \quad k_\ell(q, \mathbf{v}) \leq s + 1 - \log(1 + \tau^2)/(2\alpha^2\beta^2).$$

Equivalently, we have

$$k_\ell(q, \mathbf{v}) \leq \left[s + 1 - \sqrt{\log(1 + \tau^2)/\beta^4}\right] \vee \left[s + 1 - \log(1 + \tau^2)/(2\alpha^2\beta^2)\right]. \tag{A.51}$$

Recall that $\tau = \sqrt{\log(T/\xi)/n}$ where $\xi = o(1)$. For any constant $\eta > 0$, we set $T = O(d^\eta)$. By combining Lemmas A.3 and A.4, (A.37), and (A.51), we further obtain

$$T \cdot \frac{\sup_{q \in \mathcal{Q}_{\mathscr{A}}} \left(|\mathcal{C}_1(q)| + |\mathcal{C}_2(q)|\right)}{|\mathcal{H}(s)|} \leq 4T \cdot \exp\left\{-\log\zeta \cdot \left[\sqrt{\log(1 + \tau^2)/\beta^4} - 1\right]\right\} \wedge$$
$$4T \cdot \exp\left\{-\log\zeta \cdot \left[\log(1 + \tau^2)/(2\alpha^2\beta^2) - 1\right]\right\}. \tag{A.52}$$

Under the assumption of the theorem, there is a sufficiently small constant $\delta > 0$ such that $s^2/d^{1-\delta} = O(1)$. Thus we have $\zeta = d/(2s^2) = \Omega(d^\delta)$. By inequality $\log(1 + x) \geq x/2$, it holds that $\log(1 +$

$\tau^2) \geq \tau^2/2 = \log(T/\xi)/(2n)$. Under the condition in (A.26) , we have

$$\frac{\log(T/\xi)}{2n\beta^4} \bigvee \frac{\log(T/\xi)}{4n\alpha^2\beta^2} > \frac{\log(1/\xi)}{2n\beta^4} \bigvee \frac{\log(1/\xi)}{4\alpha^2\beta^2} \to \infty. \tag{A.53}$$

Hence if $n$ is sufficiently large, the left-hand side in (A.53) is greater than an absolute constant $C$ satisfiying $\delta(C-1) > \eta$. Then by (A.52) we have

$$T \cdot \frac{\sup_{q \in \mathcal{Q}_{\mathscr{A}}}(|\mathcal{C}_1(q)| + |\mathcal{C}_2(q)|)}{|\mathcal{H}(s)|} = O[4d^\eta \zeta^{-(C-1)}] = O[4d^\eta d^{-\delta(C-1)}] = o(1). \tag{A.54}$$

Combining (A.54) and Lemma A.3, we conclude that $\overline{R}_n^*(\mathcal{G}_0, \mathcal{G}_1; \mathscr{A}, r) \to 1$ if (A.26) holds. This concludes the proof of Theorem 3.3.

## A.4   Proof of Theorem 3.4

To ease notation, we denote the joint distribution of $(Y, \boldsymbol{X})$ by $\mathbb{P}_{\boldsymbol{\theta}}$ where the model parameter is given by $\boldsymbol{\theta} = (\boldsymbol{\mu}_0, \boldsymbol{\mu}_1, \boldsymbol{\Sigma}, \alpha)$. In addition, we let $\Delta\boldsymbol{\mu} = \boldsymbol{\mu}_1 - \boldsymbol{\mu}_0$. Thus $\Delta\boldsymbol{\mu} = \mathbf{0}$ for all $\boldsymbol{\theta} \in \mathcal{G}_0(\boldsymbol{\Sigma})$ and $\Delta\boldsymbol{\mu} \in \mathcal{B}(s)$ for all $\boldsymbol{\theta} \in \mathcal{G}_1(\boldsymbol{\Sigma}; \gamma_n)$. In what follows, we bound the type-I and type-II errors of $\overline{\phi}$ respectively.

**Type-I error.**   For any $\boldsymbol{\theta} \in \mathcal{G}_0(\boldsymbol{\Sigma})$, by the definition of $\overline{\phi}$, the type-I error is bounded by

$$\overline{\mathbb{P}}_{\boldsymbol{\theta}}(\overline{\phi} = 1) \leq \overline{\mathbb{P}}_{\boldsymbol{\theta}}(\overline{\phi}_1 = 1) + \overline{\mathbb{P}}_{\boldsymbol{\theta}}(\overline{\phi}_2 = 1).$$

For test function $\overline{\phi}_1$, since marginally, $\boldsymbol{X} \sim 1/2 \cdot \mathcal{N}(\boldsymbol{\mu}_0, \boldsymbol{\Sigma}) + 1/2 \cdot \mathcal{N}(\boldsymbol{\mu}_1, \boldsymbol{\Sigma})$, for any $\boldsymbol{\theta} \in \mathcal{G}_0(\boldsymbol{\Sigma}) \cup \mathcal{G}_1(\boldsymbol{\Sigma}; \gamma_n)$, for any $j \in [d]$, we have

$$\mathbb{E}_{\mathbb{P}_{\boldsymbol{\theta}}}(X_j^2/\sigma_j - 1) - \left[\mathbb{E}_{\mathbb{P}_{\boldsymbol{\theta}}}(X_j/\sqrt{\sigma_j})\right]^2$$
$$= 1/4 \cdot (\mu_{0,j} - \mu_{1,j})^2/\sigma_j = 1/4 \cdot (\Delta\boldsymbol{\mu})_j^2/\sigma_j, \tag{A.55}$$

Here $\mu_{0,j}$ and $\mu_{1,j}$ denote the $j$-th entries of $\boldsymbol{\mu}_0$ and $\boldsymbol{\mu}_1$, and $(\Delta\boldsymbol{\mu})_j$ is the $j$-th entry of $\Delta\boldsymbol{\mu}$. In addition, by the definition of $q_j$ in (3.13) we have

$$\left|\left[\mathbb{E}_{\mathbb{P}_{\boldsymbol{\theta}}}q_j(Y, \boldsymbol{X})\right]^2 - \left[\mathbb{E}_{\mathbb{P}_{\boldsymbol{\theta}}}(X_j/\sqrt{\sigma_j})\right]^2\right|$$

$$\leq 2\left|\mathbb{E}_{\mathbb{P}_{\boldsymbol{\theta}}}(X_j/\sqrt{\sigma_j})\right| \cdot \left|\mathbb{E}_{\mathbb{P}_{\boldsymbol{\theta}}}(X_j/\sqrt{\sigma_j}) - \mathbb{E}_{\mathbb{P}_{\boldsymbol{\theta}}}q_j(Y, \boldsymbol{X})\right| + \left|\mathbb{E}_{\mathbb{P}_{\boldsymbol{\theta}}}(X_j/\sqrt{\sigma_j}) - \mathbb{E}_{\mathbb{P}_{\boldsymbol{\theta}}}q_j(Y, \boldsymbol{X})\right|^2.$$

Since $X_j/\sqrt{\sigma_j} - q_j(Y, \boldsymbol{X}) = X_j/\sqrt{\sigma_j} \cdot \mathbb{1}\{|X_j/\sqrt{\sigma_j}| > R \cdot \sqrt{\log d}\}$, by Cauchy-Schwarz inequality we have

$$\left|\mathbb{E}_{\mathbb{P}_{\boldsymbol{\theta}}}(X_j/\sqrt{\sigma_j}) - \mathbb{E}_{\mathbb{P}_{\boldsymbol{\theta}}}q_j(Y, \boldsymbol{X})\right|^2 \leq \mathbb{E}_{\mathbb{P}_{\boldsymbol{\theta}}}(X_j^2/\sigma_j) \cdot \mathbb{P}_{\boldsymbol{\theta}}(|X_j/\sqrt{\sigma_j}| > R \cdot \sqrt{\log d}). \tag{A.56}$$

Since $\|\boldsymbol{\mu}_0\|_\infty \vee \|\boldsymbol{\mu}_1\|_\infty \leq C_0$ and $\{X_j/\sqrt{\sigma_j}\}_{i=1}^d$ are sub-Gaussian random variables, for any $t > 0$, there exists a constant $C_1$ such that

$$\mathbb{P}_{\boldsymbol{\theta}}(|X_j/\sqrt{\sigma_j}| > t) \leq 2\exp(-C_1 t^2). \tag{A.57}$$

Thus setting $t = R \cdot \sqrt{\log d}$ for some sufficiently large $R$, by (A.56) and (A.57) we obtain

$$\left|\mathbb{E}_{\mathbb{P}_{\boldsymbol{\theta}}}(X_j/\sqrt{\sigma_j}) - \mathbb{E}_{\mathbb{P}_{\boldsymbol{\theta}}}q_j(Y, \boldsymbol{X})\right| \leq C_2 d^{-1}$$

for some constant $C_2$. Thus we have

$$\left|\left[\mathbb{E}_{\mathbb{P}_{\boldsymbol{\theta}}}q_j(Y, \boldsymbol{X})\right]^2 - \left[\mathbb{E}_{\mathbb{P}_{\boldsymbol{\theta}}}(X_j/\sqrt{\sigma_j})\right]^2\right| \leq 2C_0 \cdot C_2 d^{-1} + C_2^2 d^{-2} \leq 1/16 \cdot (\Delta\boldsymbol{\mu})_j^2/\sigma_j. \tag{A.58}$$

In addition, since $X_j^2/\sigma_j - 1 - \widetilde{q}_j(Y, \boldsymbol{X}) = (X_j^2/\sigma_j - 1) \cdot \mathbb{1}\{|X_j/\sqrt{\sigma_j}| > R \cdot \sqrt{\log d}\}$, for $\widetilde{q}_j$ defined in (3.14), we similarly we obtain

$$\left|\mathbb{E}_{\mathbb{P}_{\boldsymbol{\theta}}}\widetilde{q}_j(Y, \boldsymbol{X}) - \mathbb{E}_{\mathbb{P}_{\boldsymbol{\theta}}}(X_j^2/\sigma_j - 1)\right| \leq 1/16 \cdot (\Delta\boldsymbol{\mu})_j^2/\sigma_j. \tag{A.59}$$

Combining (A.58) and (A.59) we have

$$\mathbb{E}_{\mathbb{P}_{\boldsymbol{\theta}}}\widetilde{q}_j(Y, \boldsymbol{X}) - [\mathbb{E}_{\mathbb{P}_{\boldsymbol{\theta}}}q_j(Y, \boldsymbol{X})]^2 \geq 1/8 \cdot (\Delta\boldsymbol{\mu})_j^2/\sigma_j \ \text{ for all } \ j \in [d].$$

Taking supremum over $j \in [d]$, we have

$$\sup_{j \in [d]}\left\{\mathbb{E}_{\mathbb{P}_{\boldsymbol{\theta}}}\widetilde{q}_j(Y, \boldsymbol{X}) - [\mathbb{E}_{\mathbb{P}_{\boldsymbol{\theta}}}q_j(Y, \boldsymbol{X})]^2\right\} \geq 1/8 \cdot \sup_{j \in [d]}\left[(\Delta\boldsymbol{\mu})_j^2/\sigma_j\right]. \tag{A.60}$$

Note that the test function $\overline{\phi}$ involves $4d$ queries functions. Thus, for any $\boldsymbol{\theta} \in \mathcal{G}_0(\boldsymbol{\Sigma}) \cup \mathcal{G}_1(\boldsymbol{\Sigma}; \gamma_n)$, under $\mathbb{P}_{\boldsymbol{\theta}}$ the tolerance parameters for $q_j$ and $\widetilde{q}_j$ are given by

$$\tau_{q_j} \leq R\sqrt{\log d} \cdot \sqrt{[\log(4d/\xi)]/n}, \ \ \tau_{\widetilde{q}_j} \leq R^2 \log d \cdot \sqrt{[\log(4d/\xi)]/n}, \text{ for all } j \in [d]. \quad (\text{A}.61)$$

Under the assumption that

$$\sup_{j \in [d]} (\Delta \boldsymbol{\mu})_j^2 / \sigma_j = \Omega \left[ \log^2 d \cdot \log(d/\xi)/(\alpha^2 n) \wedge \log d \cdot \sqrt{\log(d/\xi)/n} \right],$$

we have

$$\tau_{q_j} \vee \tau_{\widetilde{q}_j} \leq R^2 \log d \cdot \sqrt{[\log(4d/\xi)]/n}$$

$$\leq (1/C) \cdot \left\{ \sup_{j \in [d]} [(\Delta \boldsymbol{\mu})_j^2 / \sigma_j] \vee \alpha \cdot \sup_{j \in [d]} |(\Delta \boldsymbol{\mu})_j / \sqrt{\sigma_j}| \right\}, \quad (\text{A}.62)$$

where the absolute constant $C$ is the same as in (3.16). Note that we denote $R^2 \log d \cdot \sqrt{\log(4d/\xi)/n}$ by $\overline{\tau}_1$. Hence by (A.62), for any $\boldsymbol{\theta} \in \mathcal{G}_0(\boldsymbol{\Sigma})$, the type-I error of $\overline{\phi}_1$ is bounded by

$$\mathbb{P}_{\boldsymbol{\theta}} \left[ \sup_{j \in [d]} (Z_{\widetilde{q}_j} - Z_{q_j}^2) \geq C\overline{\tau}_1 \right]$$

$$= \mathbb{P}_{\boldsymbol{\theta}} \left( \bigcup_{j \in [d]} \left\{ (Z_{\widetilde{q}_j} - Z_{q_j}^2) - \{ \mathbb{E}_{\mathbb{P}_{\boldsymbol{\theta}}} \widetilde{q}_j(Y, \boldsymbol{X}) - [\mathbb{E}_{\mathbb{P}_{\boldsymbol{\theta}}} q_j(Y, \boldsymbol{X})]^2 \} \geq C\overline{\tau}_1 \right\} \right)$$

$$\leq \mathbb{P}_{\boldsymbol{\theta}} \left( \bigcup_{j \in [d]} \left\{ Z_{\widetilde{q}_j} - \mathbb{E}_{\boldsymbol{\theta}} \widetilde{q}_j(Y, \boldsymbol{X}) \geq \overline{\tau}_1 \right\} \right) + \mathbb{P}_{\boldsymbol{\theta}} \left( \bigcup_{j \in [d]} \left\{ Z_{q_j}^2 - [\mathbb{E}_{\mathbb{P}_{\boldsymbol{\theta}}} q_j(Y, \boldsymbol{X})]^2 \geq (C-1)\overline{\tau}_1 \right\} \right).$$

For the first term, we have

$$\mathbb{P}_{\boldsymbol{\theta}} \left( \bigcup_{j \in [d]} \left\{ Z_{\widetilde{q}_j} - \mathbb{E}_{\boldsymbol{\theta}} \widetilde{q}_j(Y, \boldsymbol{X}) \geq \overline{\tau}_1 \right\} \right)$$

$$\leq \mathbb{P}_{\boldsymbol{\theta}} \left( \bigcup_{j \in [d]} \left\{ |Z_{\widetilde{q}_j} - \mathbb{E}_{\boldsymbol{\theta}} \widetilde{q}_j(Y, \boldsymbol{X})| \geq \tau_{\widetilde{q}_j} \right\} \right) \leq \xi. \quad (\text{A}.63)$$

Note that under the null hypothesis $\boldsymbol{\theta} \in \mathcal{G}_0(\boldsymbol{\Sigma})$, we have $\mathbb{E}_{\mathbb{P}_{\boldsymbol{\theta}}} q_j(Y, \boldsymbol{X}) = \mu_{0,j}/\sqrt{\sigma_j}$. Under the assumption that $\|\boldsymbol{\mu}_0\|_\infty \vee \|\boldsymbol{\mu}_1\|_\infty \leq C_0$, when $n$ is sufficiently large such that

$$\overline{\tau}_1 \leq 3(C-1)^{-1} C_0 / \sqrt{\sigma_j},$$

by $Z_{q_j}^2 - [\mathbb{E}_{\mathbb{P}_{\boldsymbol{\theta}}} q_j(Y, \boldsymbol{X})]^2 \geq (C-1)\overline{\tau}_1$ we have

$$|Z_{q_j} - \mathbb{E}_{\mathbb{P}_{\boldsymbol{\theta}}} q_j(Y, \boldsymbol{X})| \geq (C-1)\overline{\tau}_1 \cdot \sqrt{\sigma_j}/(3C_0). \quad (\text{A}.64)$$

Thus we can set absolute constant $C$ sufficiently large such that $|Z_{q_j} - \mathbb{E}_{\mathbb{P}_{\boldsymbol{\theta}}} q_j(Y, \boldsymbol{X})| \geq \overline{\tau}_1$. Thus by (A.64) we have

$$\mathbb{P}_{\boldsymbol{\theta}} \left( \bigcup_{j \in [d]} \left\{ Z_{q_j}^2 - [\mathbb{E}_{\mathbb{P}_{\boldsymbol{\theta}}} q_j(Y, \boldsymbol{X})]^2 \geq (C-1)\overline{\tau}_1 \right\} \right)$$

$$\leq \mathbb{P}_{\boldsymbol{\theta}} \left( \bigcup_{j \in [d]} \left\{ |Z_{q_j} - \mathbb{E}_{\mathbb{P}_{\boldsymbol{\theta}}} q_j(Y, \boldsymbol{X})| \geq \tau_{q_j} \right\} \right) \leq \xi. \quad (\text{A}.65)$$

Combining (A.63) and (A.65), we can bound the type-I error of $\overline{\phi}_1$ by $2\xi$. For the type-I error of $\overline{\phi}_2$, we define $\boldsymbol{Z} = (2Y - 1) \cdot \boldsymbol{X}$. Under the data-generating model defined in (2.1) and (2.2), the distribution of $\boldsymbol{Z}$ is given by

$$\boldsymbol{Z} \sim \frac{1+\alpha}{4} \mathcal{N}(-\boldsymbol{\mu}_0, \boldsymbol{\Sigma}) + \frac{1+\alpha}{4} \mathcal{N}(\boldsymbol{\mu}_1, \boldsymbol{\Sigma}) + \frac{1-\alpha}{4} \mathcal{N}(\boldsymbol{\mu}_0, \boldsymbol{\Sigma}) + \frac{1-\alpha}{4} \mathcal{N}(-\boldsymbol{\mu}_1, \boldsymbol{\Sigma}).$$

Then by definition, for all $\boldsymbol{\theta} \in \mathcal{G}_0(\boldsymbol{\Sigma})$, we have

$$\mathbb{E}_{\mathbb{P}_{\boldsymbol{\theta}}} [\mathbf{v}^\top \text{diag}(\boldsymbol{\Sigma})^{-1/2} \boldsymbol{Z}] = 0, \text{ for all } \mathbf{v} \in \mathcal{B}_2(1). \quad (\text{A}.66)$$

In addition, for any $\boldsymbol{\theta} \in \mathcal{G}_1(\boldsymbol{\Sigma}; \gamma_n)$, by the distribution of $\boldsymbol{Z}$, for all $\mathbf{v} \in \mathcal{B}_2(1)$, we have

$$\mathbb{E}_{\mathbb{P}_{\boldsymbol{\theta}}} [\mathbf{v}^\top \text{diag}(\boldsymbol{\Sigma})^{-1/2} \boldsymbol{Z}] = \alpha/2 \cdot \mathbf{v}^\top \text{diag}(\boldsymbol{\Sigma})^{-1/2} \Delta \boldsymbol{\mu}. \quad (\text{A}.67)$$

Moreover, by definition we have

$$\mathbf{v}^\top \text{diag}(\boldsymbol{\Sigma})^{-1/2} \boldsymbol{Z} - \overline{q} \ (Y, \boldsymbol{X}) = \mathbf{v}^\top \text{diag}(\boldsymbol{\Sigma})^{-1/2} \boldsymbol{Z} \cdot \mathbb{1}\{ |\mathbf{v}^\top \text{diag}(\boldsymbol{\Sigma})^{-1/2} \boldsymbol{Z}| \leq R\sqrt{\log d} \}.$$

By setting the constant $R$ sufficiently large, for any for any $\boldsymbol{\theta} \in \mathcal{G}_0(\boldsymbol{\Sigma}) \cup \mathcal{G}_1(\boldsymbol{\Sigma}; \gamma_n)$, we have

$$\left| \mathbb{E}_{\mathbb{P}_{\boldsymbol{\theta}}} \overline{q} \ (Y, \boldsymbol{X}) - \mathbb{E}_{\mathbb{P}_{\boldsymbol{\theta}}} (\mathbf{v}^\top \text{diag}(\boldsymbol{\Sigma})^{-1/2} \boldsymbol{Z}) \right| \leq \alpha/4 \cdot |(\Delta \boldsymbol{\mu})_j / \sqrt{\sigma_j}|.$$

Combining (A.66) and (A.67) we obtain that

$$\mathbb{E}_{\mathbb{P}_{\boldsymbol{\theta}}}\overline{q}\ (Y,\boldsymbol{X}) \leq \alpha/4 \cdot |(\Delta\boldsymbol{\mu})_j/\sqrt{\sigma_j}| \ \text{ for all } \ \boldsymbol{\theta} \in \mathcal{G}_0(\boldsymbol{\Sigma});$$
$$\mathbb{E}_{\mathbb{P}_{\boldsymbol{\theta}}}\overline{q}\ (Y,\boldsymbol{X}) \geq \alpha/4 \cdot |(\Delta\boldsymbol{\mu})_j/\sqrt{\sigma_j}| \ \text{ for all } \ \boldsymbol{\theta} \in \mathcal{G}_1(\boldsymbol{\Sigma},\gamma_n).$$

Thus, taking the supremum over $\mathcal{B}_2(1)$ yields

$$\sup_{\in \mathcal{B}_2(1)} \mathbb{E}_{\mathbb{P}_{\boldsymbol{\theta}}}\overline{q}\ (Y,\boldsymbol{X}) \geq \alpha/4 \cdot \sup_{j \in [d]} |(\Delta\boldsymbol{\mu})_j/\sqrt{\sigma_j}| \ \text{ for all } \ \boldsymbol{\theta} \in \mathcal{G}_1(\boldsymbol{\Sigma},\gamma_n). \tag{A.68}$$

In addition, since we have $4d$ queries, by Definition 2.2, the tolerance parameters for $\overline{q}$ 's are bouded by

$$\tau_{\overline{q}} \ \leq R\sqrt{\log d} \cdot \sqrt{[\log(4d/\xi)]/n}, \ \text{for all } \mathbf{v} \in \mathcal{B}_2(1).$$

Note that we denote $\overline{\tau}_2 = R\sqrt{\log d} \cdot \sqrt{[\log(4d/\xi)]/n}$. Similar to (A.62), we have

$$\tau_{\overline{q}} \ \leq \overline{\tau}_1 \leq (1/C) \cdot \left\{ \sup_{j \in [d]}[(\Delta\boldsymbol{\mu})_j^2/\sigma_j] \vee \sup_{j \in [d]} \alpha|(\Delta\boldsymbol{\mu})_j/\sqrt{\sigma_j}| \right\}. \tag{A.69}$$

Hence by (A.69), for any $\boldsymbol{\theta} \in \mathcal{G}_0(\boldsymbol{\Sigma})$, the type-I error of $\overline{\phi}_2$ is bounded by

$$\overline{\mathbb{P}}_{\boldsymbol{\theta}} \left( \sup_{\in \mathcal{B}_2(1)} Z_{\overline{q}} \ \geq 2\overline{\tau}_1 \right) = \overline{\mathbb{P}}_{\boldsymbol{\theta}} \left( \bigcup\nolimits_{\in \mathcal{B}_2(1)} \left\{ Z_{\overline{q}} \ - \mathbb{E}_{\ \boldsymbol{\theta}}[\overline{q}\ (Y,\boldsymbol{X})] > \overline{\tau}_1 \right\} \right)$$
$$\leq \overline{\mathbb{P}}_{\boldsymbol{\theta}} \left( \bigcup\nolimits_{\in \mathcal{B}_2(1)} \left\{ |Z_{\overline{q}} \ - \mathbb{E}_{\ \boldsymbol{\theta}}[\overline{q}\ (Y,\boldsymbol{X})]| \geq \tau_{\overline{q}} \ \right\} \right) \leq \xi. \tag{A.70}$$

Combining (A.63), (A.65), and (A.70), we have

$$\overline{\mathbb{P}}_{\boldsymbol{\theta}}(\overline{\phi} = 1) \leq 3\xi, \ \text{ for all } \boldsymbol{\theta} \in \mathcal{G}_0(\boldsymbol{\Sigma}).$$

**Type-II error.** Now we consider $\boldsymbol{\theta} \in \mathcal{G}_1(\boldsymbol{\Sigma}; \gamma_n)$. Note that $\overline{\phi} = 0$ if $\overline{\phi}_1 = 0$ and $\overline{\phi}_2 = 0$. Thus, for any $\boldsymbol{\theta} \in \mathcal{G}_1(\boldsymbol{\Sigma}; \gamma_n)$, we have

$$\overline{\mathbb{P}}_{\boldsymbol{\theta}}(\overline{\phi} = 0) = \overline{\mathbb{P}}_{\boldsymbol{\theta}}(\overline{\phi}_1 = 0 \cap \overline{\phi}_2 = 0) \leq \overline{\mathbb{P}}_{\boldsymbol{\theta}}(\overline{\phi}_1 = 0) \wedge \overline{\mathbb{P}}_{\boldsymbol{\theta}}(\overline{\phi}_2 = 0).$$

Recall that we denote $\Delta\boldsymbol{\mu} = \boldsymbol{\mu}_1 - \boldsymbol{\mu}_0$. Similar to the proof of Theorem 3.2, we consider two cases of the condition

$$\sup_{j \in [d]} (\Delta\boldsymbol{\mu})_j^2/\sigma_j = \Omega \left[ \log(d/\xi)/(\alpha^2 \cdot n) \wedge \sqrt{\log(d/\xi)/n} \right].$$

**Case (i).** We show that the type-II error of $\overline{\phi}_1$ is negligible under the assumption that

$$\sup_{j \in [d]} (\Delta\boldsymbol{\mu})_j^2/\sigma_j = \Omega \left[ \sqrt{\log(d/\xi)/n} \right].$$

Let $j^* = \mathrm{argmax}_{j \in [d]}(\Delta\boldsymbol{\mu})_j^2/\sigma_j$. Then by (A.62), when we have

$$1 + C\overline{\tau} \leq (\Delta\boldsymbol{\mu})_{j^*}^2/\sigma_{j^*} + 1 - C\overline{\tau} = \mathbb{E}_{\ \boldsymbol{\theta}}\widetilde{q}_{j^*}(Y,\boldsymbol{X}) - [\mathbb{E}_{\ \boldsymbol{\theta}}q_{j^*}(Y,\boldsymbol{X})]^2 - C\overline{\tau}. \tag{A.71}$$

Thus combining (A.58), (A.59), and (A.71), we have

$$\overline{\mathbb{P}}_{\boldsymbol{\theta}} \left[ \sup_{j \in [d]} (Z_{q_j} - Z_{\widetilde{q}_j}^2) < C\overline{\tau}_1 \right]$$
$$\leq \overline{\mathbb{P}}_{\boldsymbol{\theta}} \left\{ Z_{\widetilde{q}_j^*} - Z_{q_j}^2 < \mathbb{E}_{\mathbb{P}_{\boldsymbol{\theta}}}\widetilde{q}_{j^*}(Y,\boldsymbol{X}) - [\mathbb{E}_{\mathbb{P}_{\boldsymbol{\theta}}}q_{j^*}(Y,\boldsymbol{X})]^2 - C\overline{\tau}_1 \right\}$$
$$\leq \overline{\mathbb{P}}_{\boldsymbol{\theta}} \left[ \mathbb{E}_{\mathbb{P}_{\boldsymbol{\theta}}}\widetilde{q}_{j^*}(Y,\boldsymbol{X}) - Z_{\widetilde{q}_{j^*}} > \overline{\tau}_1 \right] + \overline{\mathbb{P}}_{\boldsymbol{\theta}} \left\{ [\mathbb{E}_{\mathbb{P}_{\boldsymbol{\theta}}}q_{j^*}(Y,\boldsymbol{X})]^2 - Z_{q_j^*}^2 > (C-1)\overline{\tau}_1 \right\}. \tag{A.72}$$

Moreover, by (A.62) the first term on the right-hand side of (A.72) can be further bounded by

$$\overline{\mathbb{P}}_{\boldsymbol{\theta}} \left[ \mathbb{E}_{\ \boldsymbol{\theta}}\widetilde{q}_{j^*}(Y,\boldsymbol{X}) - Z_{\widetilde{q}_{j^*}} > \overline{\tau}_1 \right]$$
$$\leq \overline{\mathbb{P}}_{\boldsymbol{\theta}} \left\{ \mathbb{E}_{\mathbb{P}_{\boldsymbol{\theta}}}\widetilde{q}_{j^*}(Y,\boldsymbol{X}) - Z_{\widetilde{q}_{j^*}} \geq \tau_{\widetilde{q}_{j^*}} \right\}$$
$$\leq \overline{\mathbb{P}}_{\boldsymbol{\theta}} \left( \bigcup\nolimits_{j \in [d]} \left\{ |Z_{\widetilde{q}_j} - \mathbb{E}_{\ \boldsymbol{\theta}}\widetilde{q}_j(Y,\boldsymbol{X})| \geq \tau_{\widetilde{q}_j} \right\} \right) \leq \xi. \tag{A.73}$$

Similarly, for the second term on the right-hand side of (A.72), by (A.62) and (A.64) we have

$$\overline{\mathbb{P}}_{\boldsymbol{\theta}} \left\{ [\mathbb{E}_{\boldsymbol{\theta}} q_{j^*}(Y, \boldsymbol{X})]^2 - Z_{q_j^*}^2 > (C-1)\overline{\tau}_1 \right\}$$

$$\le \overline{\mathbb{P}}_{\boldsymbol{\theta}} \left( \bigcup_{j \in [d]} \left\{ |Z_{q_j} - \mathbb{E}_{\boldsymbol{\theta}} \widetilde{q}_j(Y, \boldsymbol{X})| \ge \tau_{q_j} \right\} \right) \le \xi. \tag{A.74}$$

Therefore, combining (A.73) and (A.74), we conclude that the type-II error of $\overline{\phi}_2$ is no more than $2\xi$.

**Case (ii).** Now we assume study the type-II error of $\overline{\phi}_2$ under the assumption that

$$\sup_{j \in [d]} (\Delta\boldsymbol{\mu})_j^2 / \sigma_j = \Omega \left[ \log(d/\xi) / (\alpha^2 \cdot n) \right].$$

Let $j^* = \mathrm{argmax}_{j \in [d]} (\Delta\boldsymbol{\mu})_j^2 / \sigma_j$ and $\mathbf{v}^* = \mathrm{argmax}_{\in \mathcal{B}_2(1)} \mathbb{E}_{\boldsymbol{\theta}} \overline{q}(Y, \boldsymbol{X})$. Then by (A.62) and (A.68), when $C > 4$ we have

$$2\overline{\tau}_2 \le \alpha/2 \cdot \sup_{j \in [d]} |(\Delta\boldsymbol{\mu})_j / \sqrt{\sigma_j}| - 2\overline{\tau}_2 = \mathbb{E}_{\boldsymbol{\theta}} \overline{q}_*(Y, \boldsymbol{X}) - 2\overline{\tau}_2. \tag{A.75}$$

Then by (A.69) and (A.75) the type-II error of $\overline{\phi}_2$ is bounded by

$$\overline{\mathbb{P}}_{\boldsymbol{\theta}} \left( \sup_{\in \mathcal{B}_2(1)} Z_{\overline{q}_{\mathbf{v}}} < 2\overline{\tau}_2 \right) \le \overline{\mathbb{P}}_{\boldsymbol{\theta}} \left[ \sup_{\mathbf{v} \in \mathcal{B}_2(1)} Z_{\overline{q}_{\mathbf{v}}} < \mathbb{E}_{\mathbb{P}_{\boldsymbol{\theta}}} \overline{q}_{\mathbf{v}^*}(Y, \boldsymbol{X}) - 2\overline{\tau}_2 \right]$$

$$\le \overline{\mathbb{P}}_{\boldsymbol{\theta}} \left[ Z_{\overline{q}_{\mathbf{v}^*}} < \mathbb{E}_{\boldsymbol{\theta}} \overline{q}_{\mathbf{v}^*}(Y, \boldsymbol{X}) - 2\overline{\tau}_2 \right]$$

$$\le \overline{\mathbb{P}}_{\boldsymbol{\theta}} \left( \bigcup_{\in \mathcal{B}_2(1)} \left\{ |Z_{\overline{q}_{\mathbf{v}}} - \mathbb{E}_{\boldsymbol{\theta}} [\overline{q}(Y, \boldsymbol{X})]| \ge \tau_{\overline{q}_{\mathbf{v}}} \right\} \right) \le \xi. \tag{A.76}$$

Thus by (A.76), the type-II error of $\overline{\phi}_2$ is no more than $\xi$. Then together with Case (i), we have $\overline{\mathbb{P}}_{\boldsymbol{\theta}}(\overline{\phi}) \le 2\xi$ for all $\boldsymbol{\theta} \in \mathcal{G}_1(\boldsymbol{\Sigma}; \gamma_n)$. Therefore the total risk of $\overline{\phi}$ is bounded by

$$\overline{R}_n(\overline{\phi}) = \sup_{\boldsymbol{\theta} \in \mathcal{G}_0( )} \overline{\mathbb{P}}_{\boldsymbol{\theta}}(\overline{\phi} = 1) + \sup_{\boldsymbol{\theta} \in \mathcal{G}_1( ; \gamma_n)} \overline{\mathbb{P}}_{\boldsymbol{\theta}}(\overline{\phi} = 0) \le 5\xi.$$

# B  Proofs for Technical Lemmas

In this section, we prove the technical lemmas which appear in the proofs of the main results.

## B.1  Proof of Lemma A.1

Under $\mathbb{P}_0$, $\boldsymbol{X}$ and $Y$ are independent with $\boldsymbol{X} \sim \mathcal{N}(\boldsymbol{0}, \mathbf{I})$ and $Y$ is uniform over $\{0, 1\}$. We denote by $f(\mathbf{x}; \boldsymbol{\mu})$ the density of $\mathcal{N}(\boldsymbol{\mu}, \mathbf{I})$ and by $p_0(y, \mathbf{x})$ the density of $\mathbb{P}_0$. Then for any $y \in \{0, 1\}$ and $\mathbf{x} \in \mathbb{R}^d$, we have $p_0(y, \mathbf{x}) = 1/2 \cdot f(\mathbf{x}; \boldsymbol{0})$. In addition, for any $\mathbf{v} \in \mathcal{H}(s)$, we denote the density of $\mathbb{P}_{\mathbf{v}}$ by $p_{\mathbf{v}}(y, \mathbf{x})$. By the definition of the statistical model, we have

$$p_{\mathbf{v}}(1, \mathbf{x}) = (1 + \alpha)/4 \cdot f(\mathbf{x}; \mathbf{v}/2) + (1 - \alpha)/4 \cdot f(\mathbf{x}; -\mathbf{v}/2),$$
$$p_{\mathbf{v}}(0, \mathbf{x}) = (1 - \alpha)/4 \cdot f(\mathbf{x}; \mathbf{v}/2) + (1 + \alpha)/4 \cdot f(\mathbf{x}; -\mathbf{v}/2).$$

Thus for any $y \in \{0, 1\}$ and $\mathbf{x} \in \mathbb{R}^d$, we have

$$\frac{d\mathbb{P}_{\mathbf{v}}}{d\mathbb{P}_0}(y, \mathbf{x}) = \frac{1}{2} \cdot \left[ \frac{f(\mathbf{x}; \mathbf{v}/2)}{f(\mathbf{x}; \boldsymbol{0})} + \frac{f(\mathbf{x}; -\mathbf{v}/2)}{f(\mathbf{x}; \boldsymbol{0})} \right] + \frac{\alpha(2y - 1)}{2} \cdot \left[ \frac{f(\mathbf{x}; \mathbf{v}/2)}{f(\mathbf{x}; \boldsymbol{0})} - \frac{f(\mathbf{x}; -\mathbf{v}/2)}{f(\mathbf{x}; \boldsymbol{0})} \right]. \tag{B.1}$$

Note that by definition, for any $\boldsymbol{\mu} \in \mathbb{R}^d$, we have

$$g(\mathbf{x}; \boldsymbol{\mu}) := f(\mathbf{x}; \boldsymbol{\mu})/f(\mathbf{x}; \boldsymbol{0}) = \exp(\boldsymbol{\mu}^\top \mathbf{x} - 1/2 \cdot \|\boldsymbol{\mu}\|_2^2).$$

Thus (B.1) is reduced to

$$\frac{d\mathbb{P}_{\mathbf{v}}}{d\mathbb{P}_0}(y, \mathbf{x}) = [g(\mathbf{x}, \mathbf{v}/2) + g(\mathbf{x}; -\mathbf{v}/2)]/2 + \alpha(2y - 1) \cdot [g(\mathbf{x}, \mathbf{v}/2) - g(\mathbf{x}; -\mathbf{v}/2)]/2. \tag{B.2}$$

For any $\mathbf{v}_1, \mathbf{v}_2 \in \mathcal{H}(s)$, by (B.2) we have

$$\mathbb{E}_0 \left[ \frac{d\mathbb{P}_{\mathbf{v}_1}}{d\mathbb{P}_0} \frac{d\mathbb{P}_{\mathbf{v}_2}}{d\mathbb{P}_0}(Y, \boldsymbol{X}) \right]$$

$$= \mathbb{E}_0 \left\{ [g(\boldsymbol{X}, \mathbf{v}_1/2) + g(\boldsymbol{X}; -\mathbf{v}_1/2)] \cdot [g(\boldsymbol{X}, \mathbf{v}_2/2) + g(\boldsymbol{X}; -\mathbf{v}_2/2)]/4 \right\}$$

$$+ \alpha^2 \cdot \mathbb{E}_0 \left\{ [g(\boldsymbol{X}, \mathbf{v}_1/2) - g(\boldsymbol{X}; -\mathbf{v}_1/2)] \cdot [g(\boldsymbol{X}, \mathbf{v}_2/2) - g(\boldsymbol{X}; -\mathbf{v}_2/2)]/4 \right\}, \tag{B.3}$$

where we use the independence of $Y$ and $\boldsymbol{X}$ under $\mathbb{P}_{\mathbf{0}}$. In what follows, we calculate the two terms on the right-hand side of (B.3), respectively. Let $\eta_1$ and $\eta_2$ be two independent Rademacher random variables over $\{-1,1\}$. Then for $\ell \in \{1,2\}$, we have

$$\left[g(\boldsymbol{X}, \mathbf{v}_\ell/2) + g(\boldsymbol{X}; -\mathbf{v}_\ell/2)\right]/2 = \mathbb{E}_{\eta_\ell}\left[g(\boldsymbol{X}, \eta_\ell \mathbf{v}_\ell/2)\right], \tag{B.4}$$

$$\left[g(\boldsymbol{X}, \mathbf{v}_\ell/2) - g(\boldsymbol{X}; -\mathbf{v}_\ell/2)\right]/2 = \mathbb{E}_{\eta_\ell}\left[\eta_\ell \cdot g(\boldsymbol{X}, \eta_\ell \mathbf{v}_\ell/2)\right]. \tag{B.5}$$

Then by (B.4) and (B.5) we have

$$\begin{aligned}
&\mathbb{E}_{\mathbf{0}}\left\{\left[g(\boldsymbol{X}, \mathbf{v}_1/2) + g(\boldsymbol{X}; -\mathbf{v}_1/2)\right] \cdot \left[g(\boldsymbol{X}, \mathbf{v}_2/2) + g(\boldsymbol{X}; -\mathbf{v}_2/2)\right]/4\right\} \\
&= \mathbb{E}_{\mathbf{0}}\mathbb{E}_{\eta_1,\eta_2}\left[g(\boldsymbol{X}; \eta_1\mathbf{v}_1/2) \cdot g(\boldsymbol{X}; \eta_2\mathbf{v}_2/2)\right] \\
&= \mathbb{E}_{\eta_1,\eta_2}\mathbb{E}_{\mathbf{0}} \exp\left[\boldsymbol{X}^\top(\eta_1\mathbf{v}_1 + \eta_2\mathbf{v}_2)/2 - 1/8 \cdot (\|\mathbf{v}_1\|_2^2 + \|\mathbf{v}_2\|_2^2)\right].
\end{aligned} \tag{B.6}$$

Using the moment-generating function of $\boldsymbol{X}$, by (B.6) we have

$$\begin{aligned}
&\mathbb{E}_{\mathbf{0}}\left\{\left[g(\boldsymbol{X}, \mathbf{v}_1/2) + g(\boldsymbol{X}; -\mathbf{v}_1/2)\right] \cdot \left[g(\boldsymbol{X}, \mathbf{v}_2/2) + g(\boldsymbol{X}; -\mathbf{v}_2/2)\right]/4\right\} \\
&= \mathbb{E}_{\eta_1,\eta_2}\left[\exp(1/2 \cdot \eta_1\eta_2 \cdot \mathbf{v}_1^\top\mathbf{v}_2) = \cosh(1/2 \cdot \langle\mathbf{v}_1, \mathbf{v}_2\rangle)\right].
\end{aligned}$$

Similarly, for (B.5) we have

$$\begin{aligned}
&\mathbb{E}_{\mathbf{0}}\left\{\left[g(\boldsymbol{X}, \mathbf{v}_1/2) - g(\boldsymbol{X}; -\mathbf{v}_1/2)\right] \cdot \left[g(\boldsymbol{X}, \mathbf{v}_2/2) - g(\boldsymbol{X}; -\mathbf{v}_2/2)\right]/4\right\} \\
&= \mathbb{E}_{\mathbf{0}}\mathbb{E}_{\eta_1,\eta_2}\left[\eta_1\eta_2 \cdot g(\boldsymbol{X}; \eta_1\mathbf{v}_1/2) \cdot g(\boldsymbol{X}; \eta_2\mathbf{v}_2/2)\right] \\
&= \mathbb{E}_{\eta_1,\eta_2}\left[\eta_1\eta_2 \cdot \exp(1/2 \cdot \eta_1\eta_2 \cdot \mathbf{v}_1^\top\mathbf{v}_2)\right] = \sinh(1/2 \cdot \langle\mathbf{v}_1, \mathbf{v}_2\rangle).
\end{aligned}$$

Thus we conclude the proof of Lemma A.1.

### B.2 Proof of Lemma A.2

It is straightforward to verify (A.4) holds when $x = 0$. We focus on region $x > 0$. It is then sufficient to prove the result for these two cases below.

**Case 1:** We consider the case $v \leq 1/(2x) \cdot \log[\cosh(2x)]$. Then we need to prove

$$\cosh(x) + v\sinh(x) \leq \cosh(2x). \tag{B.7}$$

Using the bound of $v$, it remains to show the function

$$f(x) = 1/(2x) \cdot \log[\cosh(2x)] \cdot \sinh(x) + \cosh(x) - \cosh(2x) \leq 0.$$

holds for all $x > 0$. It's easy to verify $f(x)$ is monotonically decreasing over $(0, \infty]$ and $\lim_{x\to 0} f(x) = 0$. We thus finish proving (B.7).

**Case 2:** We consider the case $v \geq 1/(2x) \cdot \log[\cosh(2x)]$. We would like to show

$$\cosh(x) + v\sinh(x) \leq \exp(2vx). \tag{B.8}$$

Let us define $g(v) := \exp(2vx) - \cosh(x) - v\sinh(x)$. We have that for any $x \geq 0$,

$$g'(v) = 2x\exp(2vx) - \sinh(x) \geq 2x\cosh(2x) - \sinh(x) \geq 0.$$

Hence, $g(v)$ is a monotonically increasing function. We thus have

$$\begin{aligned}
g(v) &\geq g\left\{1/(2x) \cdot \log[\cosh(2x)]\right\} \\
&= \cosh(2x) - \cosh(x) - 1/(2x) \cdot \log[\cosh(2x)] \cdot \sinh(x) = -f(x) \geq 0.
\end{aligned}$$

We thus finish proving (B.8).

## C Supporting Lemmas

In this section we list the supporting lemmas that establish two concentration inequalities for Gaussian random variables.

**Lemma C.1** ($\chi^2$-tail bound, [16]). *Let $X_1, \ldots, X_n$ be $n$ i.i.d. standard normal random variables. For all $t \in (0, 1)$,*

$$\mathbb{P}\left(\left|\frac{1}{n}\sum_{i=1}^n X_i^2 - 1\right| \geq t\right) \leq 2\exp(-nt^2/8).$$

**Lemma C.2** (Gaussian covariance estimation, [25]). *Suppose $\{\boldsymbol{X}_i\}_{i=1}^n$ are $n$ i.i.d. Gaussian random vectors in $\mathbb{R}^d$ and $\boldsymbol{X}_1 \sim \mathcal{N}(\mathbf{0}, \boldsymbol{\Sigma})$. For every $\epsilon \in (0, 1)$, and $t \geq 1$, if $n \geq C(t/\epsilon)^2 d$ for some*

*constant C, then with probability at least $1 - 2e^{-t^2 n}$,*

$$\|\widehat{\boldsymbol{\Sigma}} - \boldsymbol{\Sigma}\|_2 \leq \epsilon \|\boldsymbol{\Sigma}\|_2,$$

*where $\widehat{\boldsymbol{\Sigma}} := 1/n \cdot \sum_{i=1}^{n} \boldsymbol{X}_i \boldsymbol{X}_i^{\top}$.*

[Supplementary Material 3]