[Reviews · NeurIPS 2016]

Reviewer 1

Summary

This paper proposes to investigate the theoretical limits, both from a statistical point of view (minimax theory) and from a computational point of view the limits of estimators in the context on weakly supervised learning. In their model, a fix proportion of the labels are draw according to a uniform sampling (the complement are the true labels). The authors provides bound showing that there are regimes where the corruption of the label slows down both the minimax rates and the computationally feasible rate. Their technical tools really on testing theory as well as on a "computational (oracle) model".

Qualitative Assessment

This paper is interesting and deals with new kind of results introducing computational aspects in standard minimax theory. The phenomenon illustrated is new to me, and present some limitation of the computationally tractable algorithm w.r.t. "theoretical" ones that could be considered in the classical minimax theory. However, due to the relative novelty of the framework, it would be important that basic definitions and properties be better presented. specific comments: l18:remove / clarify this part. In the following there is only one model investigated. Please introduce only the one used (no missing values later on...). l24/27/29: "See e.g. [...] for a survey" repeated 3 times in 3 lines... l43: by a standard reduction argument [39]. Please provide details page/results for books. l49: log d/n is ambiguous. Please use adapted parenthesis everywhere. l74: the drawback raised "hinge on planted clique" should be described with more details and explain why the authors do not suffer the same issue here. l86: notations - > notation. Also it could be given earlier since some quantities where used before being introduced (eg max sign, \Omega etc.) l138 (Definition 2.2). This definition could be more commented and written more carefully. For instance, what the index T stands for is not that clear, especially in (2.5) there is no T, though the line below refers to R(\xi,n,T,\eta) as the set of oracles satisfying (2.5)... I believe that this part is for the community the one with the main novelty, so spending more time on describing it, and giving examples could be helpful. For instance the point raised on line 154 could be explained with more details. l167: why is \eta chosen of the order d^\eta? l214: (3.7) could be written with the expression given in the Appendix, l477. That would make it easier to understand where it the test is coming from. l225: "the" is missing before "test function". l234: "not computationally tractable": this term is not defined/precise here. To me it is tractable... in a long long time...So this difficulty should be linked with the framework introduced by the authors. l276: in in - > in l283: the quantity z_q_j and \tilde z_q_j are not clearly introduced. What are they? l296: "the" is missing before "test function", and "an diagonal" should be "a diagonal" l302: q(Y,X,Y',X) should be q(Y,X,Y',X'). l314: I guess if one estimate \Delta_\mu an estimator of this quantity should be written \widehat{\Delta_\mu} rather than \Delta_\hat{\mu} l398: the year is wrong in the citation, it should be 2009 Appendix: l420: Pinsker's inequality could be given a (book) reference for the interested reader. l454: ".." one dot is enough. l466: "we introduce random vector W" - > we introduce the random vector W l482: the lemma cited give the inequality for the operator norm. Please provide the similar version for the l2 norm considered, or the way to derive it from the other. l489: I suspect a typo in the control of the probability. Please double check. l492: "Recall that \phi_1" - > Recall that \phi_2 l585: please use align correctly. l629: the sentence is weird... l640: before you used another sign for the max of two quantities. Please be consistent.

Confidence in this Review

2-Confident (read it all; understood it all reasonably well)


Reviewer 2

Summary

The authors show that under a Gaussian generative model, given asymptotic bounds hold for i) any classifier and ii) any classifier within a certain class of polynomial (i.e., tractable) algorithms. The work is very theoretical, I'd say more theoretical than is common in NIPS, but otherwise solid. The theoretical set-up appears quite natural, although somewhat restricted by the Gaussian assumption (covariance fixed, means differ between classes by a sparse vector).

Qualitative Assessment

My only real concern (beyond the question whether this is too theoretical to NIPS), which I ask the authors to fix, is that the title and the abstract make no mention of the assumed generative model. The model should definitely be mentioned. minor comments: - l. 16: "the n independent data points": omit 'the' - ll. 18-19 (and in the abstract): the definition of the label noise process is ambiguous in the sense that "flipped" can be interpreted as a deterministic reversal of the label (that's what I read in the abstract). Moreover, the definition on lines 18-19 states that with probability 1-alpha, the label is randomly chosen (in which case, with probability 0.5, we have y_i=z_i), which is not compatible with the statement that with probability alpha, we have y_i=z_i. Please clarify the wording. - l. 23: "than sample size n" should be "than the sample size n". - l. 44: the second H_0 should be H_1 - l. 59: "these boundaries ... implies", should be plural 'imply' - ll. 140-141: I have trouble understanding the concept of a query function, and consequently, the interpretation of the event whose probability is bounded by Eq. (2.5). What is the meaning of E[q(Y,X)]? - l. 398, 408, etc.: Use Capital Initials for book titles. - l. 406: Conference details? (How many'th?) - l. 408: mention that this is a technical report, give report number. - l. 347: [12] appeared (in part) in ICML-2014. Would that be a better target of citation? ... (please be more thorough with bibliographical details in general)

Confidence in this Review

1-Less confident (might not have understood significant parts)


Reviewer 3

Summary

This paper studies the weakly supervised binary classification problem with a fraction of the labels randomly corrupted. The phase transition behaviors for asymptotic powerful/powerless test of both information-theoretic boundary and computational boundary are studied, where the dependence on the degree of supervision is discussed for the statistical and computational tradeoffs. The theoretical conclusion matches with the intuition that having less supervision requires more computational effort to match the information-theoretic boundary, while having more supervision can improve both statistical and computational efficiency. In addition, the analysis of the lower bounds and upper bounds on SNR for both statistical and computational boundaries seem to shed great light on capturing the effect of the degree of supervision. The overall paper is well presented, and the analysis is sound based on the reviewer's examination.

Qualitative Assessment

The technical results are well analyzed and the proofs are sound. The authors demonstrate the minimax-optimal statistical rate for weekly supervised binary classification problem, along with the bounds of the computational algorithm that may achieve the optimal statistical rate. The results that simultaneously capture the phase transition behaviors of both statistical and computational boundaries for weekly supervised learning are novel, which may have impact on designing new algorithms that are efficient in both statistical and computational performances. The presentation of the paper is well organized and easy to follow overall. Some suggestions: The model presented in this paper is rather general. The high level result is easy to understand, but it may lack some detailed examination of concreate examples that fit the framework. It may be better motivated and understood if some concrete results of problems can be provided. Besides, it can be more convincing if numerical experiments are carried out as evaluations of the theory. Moreover, it may be better understood and easier to read if some high level idea/summary of the proof for main results can be provided.

Confidence in this Review

2-Confident (read it all; understood it all reasonably well)


Reviewer 4

Summary

The paper studies theoretical perspective of weakly supervised problem where the labels are randomly flipped or missing with probability $1-\alpha$. The paper studies the effect of $\alpha$ by exploring both computational and information-theoretic boundaries of algorithms.

Qualitative Assessment

Binary classification is an important problem. Numerous applications require binary classification. Authors have emphasized on practical classification in the very first line, Introduction section. However, I cannot see any empirical results showing the effect of $\alpha$ on both computational and information-theoretic boundaries. Theoretical analysis is necessary, but the physical meaning of those boundaries from the perspective of binary classification has to be explained and verified empirically. Authors have mentioned in the abstract about the dependence of gap between computational and information-theoretic boundaries with different values of $\alpha$. It would be interesting to some experimental results in this perspective on real-world data.

Confidence in this Review

1-Less confident (might not have understood significant parts)


Reviewer 5

Summary

In this paper, the authors studied the effect of the degree of supervision by exploring both computational and information-theoretic boundaries. They identified three regimes, i.e., efficient, intractable and impossible regimes, when they discussed the joint effect of the degree of supervision and the signal length.

Qualitative Assessment

- Typo: line 193, "as label qualities decreases" should be "as label qualities decrease". - The writing of this paper is very dry and hard to follow. The authors should add a few illustrative figures. - Why do the authors call the boundary discussed in Section 3 "information-theoretic boundary"? The reviewer did not see information-theoretic quantities in Theorems 3.1 and 3.2. The authors should elaborate on this. - This paper looks more like a statistical paper. Can the authors explain how this work could possibly shed light on the future research of machine learning? - Theoretical results in this paper were stated and proved without any intuition, which makes this paper extremely hard to follow.

Confidence in this Review

2-Confident (read it all; understood it all reasonably well)